# Momentum Provably Improves Error Feedback!

**Ilyas Fatkhullin**
ETH AI Center & ETH Zurich

**Alexander Tyurin**
KAUST*

**Peter Richtárik**
KAUST

## Abstract

Due to the high communication overhead when training machine learning models in a distributed environment, modern algorithms invariably rely on lossy communication compression. However, when untreated, the errors caused by compression propagate, and can lead to severely unstable behavior, including exponential divergence. Almost a decade ago, Seide et al. [2014] proposed an error feedback (EF) mechanism, which we refer to as EF14, as an immensely effective heuristic for mitigating this issue. However, despite steady algorithmic and theoretical advances in the EF field in the last decade, our understanding is far from complete. In this work we address one of the most pressing issues. In particular, in the canonical nonconvex setting, all known variants of EF rely on very large batch sizes to converge, which can be prohibitive in practice. We propose a surprisingly simple fix which removes this issue both theoretically, and in practice: the application of Polyak's momentum to the latest incarnation of EF due to Richtárik et al. [2021] known as EF21. Our algorithm, for which we coin the name EF21-SGDM, improves the communication and sample complexities of previous error feedback algorithms under standard smoothness and bounded variance assumptions, and does not require any further strong assumptions such as bounded gradient dissimilarity. Moreover, we propose a double momentum version of our method that improves the complexities even further. Our proof seems to be novel even when compression is removed from the method, and as such, our proof technique is of independent interest in the study of nonconvex stochastic optimization enriched with Polyak's momentum.

## 1  Introduction

Since the practical utility of modern machine learning models crucially depends on our ability to train them on large quantities of training data, it is imperative to perform the training in a distributed storage and compute environment. In federated learning (FL) [Konečný et al., 2016, Kairouz, 2019], for example, data is naturally stored in a distributed fashion across a large number of clients (who capture and own the data in the first place), and the goal is to train a single machine learning model from the wealth of all this distributed data, in a private fashion, directly on their devices.

**1.1 Formalism.** We consider the problem of collaborative training of a single model by several clients in a data-parallel fashion. In particular, we aim to solve the *distributed nonconvex stochastic optimization problem*

$$\min_{x \in \mathbb{R}^d} \left[ f(x) := \tfrac{1}{n} \sum_{i=1}^{n} f_i(x) \right], \qquad f_i(x) := \mathbb{E}_{\xi_i \sim \mathcal{D}_i} \left[ f_i(x, \xi_i) \right], \qquad i = 1, \dots, n, \qquad (1)$$

where $n$ is the number of clients, $x \in \mathbb{R}^d$ represents the parameters of the model we wish to train, and $f_i(x)$ is the (typically nonconvex) loss of model parameterized by the vector $x$ on the data $\mathcal{D}_i$ owned by client $i$. Unlike most works in federated learning, we do not assume the datasets to be similar, i.e., we allow the distributions $\mathcal{D}_1, \dots, \mathcal{D}_n$ to be arbitrarily different.

---

*King Abdullah University of Science and Technology, Thuwal, Saudi Arabia.

37th Conference on Neural Information Processing Systems (NeurIPS 2023).

We are interested in the fundamental problem of finding an approximately stationary point of $f$ in expectation, i.e., we wish to find a (possibly random) vector $\hat{x} \in \mathbb{R}^d$ such that $\mathbb{E}\left[\|\nabla f(\hat{x})\|\right] \leq \varepsilon$. In order to solve this problem, we assume that the $n$ clients communicate via an orchestrating server. Typically, the role of the server is to first perform aggregation of the messages obtained from the workers, and to subsequently broadcast the aggregated information back to the workers. Following an implicit assumption made in virtually all theoretically-focused papers on communication-efficient training, we also assume that the speed of client-to-workers broadcast is so fast (compared to speed of workers-to-client communication) that the cost associated with broadcast can be neglected[2].

**1.2 Aiming for communication and computation efficiency at the same time.** In our work, we pay attention to two key aspects of efficient distributed training—*communication cost* and *computation cost* for finding an approximate stationary point $\hat{x}$. The former refers to the number of bits that need to be communicated by the workers to the server, and the latter refers to the number of stochastic gradients that need to be sampled by each client. The rest of the paper can be summarized as follows: *We pick one of the most popular communication-efficient gradient-type methods (the EF21 method of Richtárik et al. [2021] – the latest variant of error feedback pioneered by Seide et al. [2014]) and modify it in a way which provably preserves its communication complexity, but massively improves its computation/sample complexity, both theoretically and in practice.*

## 2 Communication Compression, Error Feedback, and Sample Complexity

Communication compression techniques such as *quantization* [Alistarh et al., 2017, Horváth et al., 2019a] and *sparsification* [Seide et al., 2014, Beznosikov et al., 2020] are known to be immensely powerful for reducing the communication footprint of gradient-type[3] methods. Arguably the most studied, versatile and practically useful class of compression mappings are *contractive* compressors.

**Definition 1** (Contractive compressors). *We say that a (possibly randomized) mapping $\mathcal{C} : \mathbb{R}^d \to \mathbb{R}^d$ is a contractive compression operator if there exists a constant $0 < \alpha \leq 1$ such that*

$$\mathbb{E}\left[\|\mathcal{C}(x) - x\|^2\right] \leq (1 - \alpha) \|x\|^2, \qquad \forall x \in \mathbb{R}^d. \tag{2}$$

Inequality (2) is satisfied by a vast array of compressors considered in the literature, including numerous variants of sparsification operators [Alistarh et al., 2018, Stich et al., 2018], quantization operators [Alistarh et al., 2017, Horváth et al., 2019a], and low-rank approximation [Vogels et al., 2019, Safaryan et al., 2022] and more [Beznosikov et al., 2020, Safaryan et al., 2021]. The canonical examples are i) the Top$K$ sparsifier, which preserves the $K$ largest components of $x$ in magnitude and sets all remaining coordinates to zero [Stich et al., 2018], and ii) the (scaled) Rand$K$ sparsifier, which preserves a subset of $K$ components of $x$ chosen uniformly at random and sets all remaining coordinates to zero [Beznosikov et al., 2020]. In both cases, (2) is satisfied with $\alpha = K/d$.

### 2.1 Brief history of error-feedback

When greedy contractive compressors, such as Top$K$, are used in a direct way to compress the local gradients in distributed gradient descent (GD), the resulting method may diverge exponentially, even on strongly convex quadratics [Beznosikov et al., 2020]. Empirically, instability caused by such a naive application of greedy compressors was observed much earlier, and a fix was proposed in the form of the *error feedback* (EF) mechanism by Seide et al. [2014], which we henceforth call EF14 or EF14-SGD (in the stochastic case).[4] To the best of our knowledge, the best *sample complexity* of EF14-SGD for finding a stationary point in the distributed nonconvex setting is given by Koloskova et al. [2020]: after $\mathcal{O}(G\alpha^{-1}\varepsilon^{-3} + \sigma^2 n^{-1}\varepsilon^{-4})$ samples[5], EF14-SGD finds a point $x$ such that $\mathbb{E}[\|\nabla f(x)\|] \leq \varepsilon$, where $\alpha$ is the contraction parameter (see Definition 1). However, such

---

[2]While this is a reasonable assumption in many practical situations [Mishchenko et al., 2019, Kairouz, 2019], some works consider the regime when the server-to-workers broadcast cannot be neglected [Horváth et al., 2019a, Tang et al., 2020, Philippenko and Dieuleveut, 2020, Kovalev et al., 2021, Fatkhullin et al., 2021, Gruntkowska et al., 2022].

[3]For Newton-type methods, see [Islamov et al., 2022] and references therein.

[4]In Appendix A, we provide a more detailed discussion on theoretical develepments for this method.

[5]Here $\sigma^2$ is the bound on the variance of stochastic gradients at each node, see Assumption 2. When referring the sample complexity we count the number of stochastic gradients used only at one of the $n$ nodes rather than by all nodes in total. This is a meaningful notion because the computations are done in parallel.

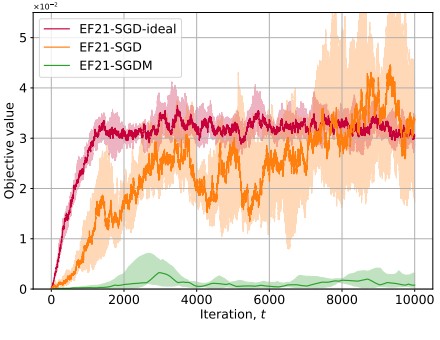 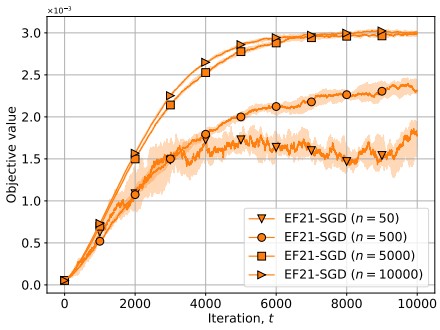

(a) Divergence for $n = 1$.  (b) No improvement with $n$.

Figure 1: Divergence of EF21-SGD on the quadratic function $f(x) = \frac{1}{2}\|x\|^2$, $x \in \mathbb{R}^2$, using the Top1 compressor. See the proof of Theorem 1 for details on the construction of the noise $\xi$; we use $\sigma = 1$, $B = 1$. The starting point is $x^0 = (0, -0.01)^\top$. Unlike EF-SGD, our method EF21-SGDM does not suffer from divergence and is stable near optimum. Figure 1b shows that when increasing the number of nodes $n$, EF21-SGD applied with $B = 1$ does not improve, and, moreover, diverges from the optimum even faster. All experiments use constant parameters $\gamma = \eta = {}^{0.1}/\sqrt{\tau} = 10^{-3}$; see Figure 4 for diminishing parameters. Each method is run 10 times and the plot shows the median performance alongside the 25% and 75% quantiles.

an analysis has two important deficiencies. First, in the deterministic case (when exact gradients are computable by each node), the analysis only gives the suboptimal $\mathcal{O}(\varepsilon^{-3})$ *iteration complexity*, which is suboptimal compared to vanilla (i.e., non-compressed) gradient descent, whose iteration complexity is $\mathcal{O}(\varepsilon^{-2})$. Second, their analysis relies heavily on additional strong assumptions, such as the *bounded gradient* (BG) assumption, $\mathbb{E}[\|\nabla f_i(x, \xi_i)\|^2] \leq G^2$ for all $x \in \mathbb{R}^d$, $i \in [n]$, $\xi_i \sim \mathcal{D}_i$, or the bounded gradient similarity (BGS) assumption, $\frac{1}{n}\sum_{i=1}^{n}\|\nabla f_i(x) - \nabla f(x)\|^2 \leq G^2$ for all $x \in \mathbb{R}^d$. Such assumptions are restrictive and sometimes even unrealistic. In particular, both BG and BGS might not hold even in the case of convex quadratic functions.[6] Moreover, it was recently shown that nonconvex analysis of stochastic gradient methods using a BG assumption may hide an exponential dependence on the smoothness constant in the complexity [Yang et al., 2023].

In 2021, these issues were *partially* resolved by Richtárik et al. [2021], who propose a modification of the EF mechanism, which they call EF21. They address both deficiencies of the original EF14 method: i) they removed the BG/BGS assumptions, and improved the iteration complexity to $\mathcal{O}(\varepsilon^{-2})$ in the full gradient regime. Subsequently, the EF21 method was modified in several directions, e.g., extended to bidirectional compression, variance reduction and proximal setup [Fatkhullin et al., 2021], generalized from contractive to three-point compressors [Richtárik et al., 2022] and adaptive compressors [Makarenko et al., 2022], modified from dual (gradient) to primal (model) compression [Gruntkowska et al., 2022] and from centralized to decentralized setting [Zhao et al., 2022]. For further work, we refer to [Wang et al., 2022, Dorfman et al., 2023, Islamov et al., 2022].

## 2.2 Key issue: error feedback has an unhealthy appetite for samples!

Unfortunately, the current theory of EF21 with *stochastic gradients* has weak sample complexity guarantees. In particular, Fatkhullin et al. [2021] extended the EF21-GD method, which is the basic variant of EF21 using full gradient at the clients, to EF21-SGD, which uses a "large minibatch" of stochastic gradients instead. They obtained $\mathcal{O}(\frac{1}{\alpha\varepsilon^2} + \frac{\sigma^2}{\alpha^3\varepsilon^4})$ sample complexity for their method. Later, Zhao et al. [2022] improved this result slightly[7] to $\mathcal{O}(\frac{1}{\alpha\varepsilon^2} + \frac{\sigma^2}{\alpha^2\varepsilon^4})$, shaving off one $\alpha$ in the stochastic term. However, it is easy to notice several issues in these results, which generally feature the fundamental challenge of combining biased gradient methods with stochastic gradients.

● **Mega-batches.** These works require all clients to sample "mega-batches" of stochastic gradients/datapoints in each iteration, of order $\mathcal{O}(\varepsilon^{-2})$, in order to control the variance coming from stochastic gradients. In Figure 1, we find that, in fact, a batch-free (i.e., with mini-batch size $B = 1$)

---

[6]For example, one can consider $f_i(x) = x^\top A_i x$ with $A_i \in \mathbb{R}^{d \times d}$, for which BG or BGS assumptions hold only in the trivial cases: matrices $A_i$ are all zero or all equal to each other (homogeneous data regime).

[7]The result was obtained under a more general setting of decentralized optimization over a network.

version of EF21-SGD diverges even on a very simple quadratic function. We also observe a similar behavior when a small batch $B > 1$ is applied. This implies that there is a fundamental flaw in the EF21-SGD method itself, rather "just" a problem of the theoretical analysis. While mega-batch methods are common in optimization literature, smaller batches are often preferred whenever they "work". For example, the time/cost required to obtain such a large number of samples at each iteration might be unreasonably large compared to the communication time, which is already reduced using compression. Moreover, when dealing with medical data, large batches might simply be unavailable [Rieke et al., 2020]. In certain applications, such as federated reinforcement learning (RL) or multi-agent RL, it is often intractable to sample more than one trajectory of the environment in order to form a gradient estimator [Mitra et al., 2023, Doan et al., 2019, Jin et al., 2022, Khodadadian et al., 2022]. Further, a method using a mega-batch at each iteration effectively follows the gradient descent (GD) dynamics instead of the dynamics of (mini-batch) SGD, which may hinder the training and generalization performance of such algorithms since it is both empirically [Keskar et al., 2017, Kleinberg et al., 2018] and theoretically [Kale et al., 2021] observed that mini-batch SGD is superior to mega-batch SGD or GD in a number of machine learning tasks.

• **Dependence on $\alpha$.** The total sample complexity results derived by Fatkhullin et al. [2021], Zhao et al. [2022] suffer from poor dependence on the contraction parameter $\alpha$. Typically, EF methods are used with the Top$K$ sparsifier, which only communicates $K$ largest entries in magnitude. In this case, $\alpha = K/d$, and the stochastic part of sample complexity scales quadratically with dimension.

• **No improvement with $n$.** The stochastic term in the sample complexity of EF21-SGD does *not* improve when increasing the number of nodes. However, the opposite behavior is typically desired, and is present in several latest non-EF methods based on *unbiased* compressors, such as MARINA [Gorbunov et al., 2021] and DASHA [Tyurin and Richtárik, 2022]. We are not aware of any distributed algorithms utilizing the Top$K$ compressor achieving linear speedup in $n$ in the stochastic term without relying on restrictive BG or BGS assumptions.

These observations motivate our work with the following central questions:

> *Can we design a batch-free distributed SGD method utilizing contractive communication compression (such as Top$K$) without relying on restrictive BG/BGS assumptions? Is it possible to improve over the current state-of-the-art $\mathcal{O}\left(\alpha^{-1}\varepsilon^{-2} + \sigma^2\alpha^{-2}\varepsilon^{-4}\right)$ sample complexity under the standard smoothness and bounded variance assumptions?*

We answer both questions in the affirmative by incorporating a momentum update into EF21-SGD.

## 2.3 Mysterious effectiveness of momentum in nonconvex optimization

An immensely popular modification of SGD (and its distributed variants) is the use of *momentum*. This technique, initially inspired by the developments in convex optimization [Polyak, 1964], is often applied in machine learning for stabilizing convergence and speeding up the training. In particular, momentum is an important part of an immensely popular and empirically successful line of adaptive methods for deep learning, including ADAM [Kingma and Ba, 2015] and a plethora of variants. The classical SGD method with Polyak (i.e., heavy ball) momentum (SGDM) reads:

$$x^{t+1} = x^t - \gamma v^t, \qquad v^{t+1} = (1-\eta)v^t + \eta \nabla f(x^{t+1}, \xi^{t+1}), \tag{3}$$

where $\gamma > 0$ is a learning rate and $\eta > 0$ is the momentum parameter.

We provide a concise walk through the key theoretical developments in the analysis of SGDM in stochastic nonconvex optimization in Appendix A; and only mention the most relevant works here. The most closely related works to ours are [Mishchenko et al., 2019], [Xie et al., 2020], and [Fatkhullin et al., 2021], which analyze momentum together with communication compression. The analysis in [Mishchenko et al., 2019, Xie et al., 2020] requires BG/BGS assumption, and does not provide any theoretical improvement over the variants without momentum. Finally, the analysis of Fatkhullin et al. [2021] is only established for deterministic case, and it is unclear if its extension to stochastic case can bring any convergence improvement over EF21-SGD. Recently, several other works attempt to explain the benefit of momentum [Plattner, 2022]; some consider structured nonconvex problems [Wang and Abernethy, 2021], and others focus on generalization [Jelassi and Li, 2022].

| Method | Communication complexity | Asymptotic sample complexity | Batch-free? | No extra assumptions? |
|---|---|---|---|---|
| EF14-SGD [Koloskova et al., 2020] | $\frac{KG}{\alpha\varepsilon^3}$ | $\frac{\sigma^2}{n\varepsilon^4}$ | ✔ | ✗ [a] |
| NEOLITHIC [Huang et al., 2022] | $\frac{K}{\alpha\varepsilon^2}\log\left(\frac{G}{\varepsilon}\right)$ [b] | $\frac{\sigma^2}{n\varepsilon^4}$ | ✗ | ✗ [c] |
| EF21-SGD [Fatkhullin et al., 2021] | $\frac{K}{\alpha\varepsilon^2}$ | $\frac{\sigma^2}{\alpha^3\varepsilon^4}$ [d] | ✗ | ✔ |
| BEER [Zhao et al., 2022] | $\frac{K}{\alpha\varepsilon^2}$ | $\frac{\sigma^2}{\alpha^2\varepsilon^4}$ [d] | ✗ | ✔ |
| EF21-SGDM Corollary 2 | $\frac{K}{\alpha\varepsilon^2}$ | $\frac{\sigma^2}{n\varepsilon^4}$ | ✔ | ✔ |

[a] Analysis requires a bound of the second moment of the stochastic gradients, i.e., $\mathbb{E}\left[\|\nabla f_i(x,\xi_i)\|^2\right] \leq G^2$ for all $x \in \mathbb{R}^d$.

[b] This complexity is achieved by using a large mini-batch and communicating $\lceil K/\alpha \rceil$ coordinates per iteration, see Appendix A.

[c] Analysis requires a bounded gradient disimilarity assumption, i.e., $\frac{1}{n}\sum_{i=1}^n \|\nabla f_i(x) - \nabla f(x)\|^2 \leq G^2$ for all $x \in \mathbb{R}^d$.

[d] Analysis requires a batch-size at least $B \geq \frac{\sigma^2}{\alpha^2\varepsilon^2}$ for EF21-SGD and $B \geq \frac{\sigma^2}{\alpha\varepsilon^2}$ for BEER.

Table 1: Summary of related works on distributed error compensated SGD methods using a TopK compressor under Assumptions 1 and 2. The goal is to find an $\varepsilon$-stationary point of a smooth nonconvex function of the form (1), i.e., a point $x$ such that $\mathbb{E}[\|\nabla f(x)\|] \leq \varepsilon$. "**Communication complexity**": the total # of communicated bits if the method is applied with sufficiently large batch-size; see Table 2 for batch-size. "**Asymptotic sample complexity**": the total # of samples required at each node to find an $\varepsilon$-stationary point for batch-size $B = 1$ in the regime $\varepsilon \to 0$. "**No extra assumptions**": ✔ means that no additional assumption is required.

**Summary of contributions.** Despite the vast amount of work trying to explain the benefits of momentum, there is no work obtaining any theoretical improvement over vanilla SGD in the smooth nonconvex setting under the standard assumptions of smoothness and bounded variance.

• First, we establish a *negative result* for a simplified/idealized version of EF21-SGD, which shows that this algorithm does not converge with constant batch-size, and that a mega-batch of order $\Omega(\sigma^2\varepsilon^{-2})$ is required. This provides a strong indication that EF21-SGD method is inherently sensitive to stochastic gradients, which is also confirmed by our numerical simulations.

• We propose a simple fix for this problem by incorporating *momentum* step into EF21-SGD, which leads to our one-batch EF21-SGDM algorithm. By leveraging our *new Lyapunov function construction and new analysis*, we establish $\mathcal{O}\left(\alpha^{-1}\varepsilon^{-2} + \sigma^2\varepsilon^{-4}\right)$ sample complexity in the single node case.

• We extend our algorithm to the distributed setting and derive an improved sample complexity result compared to other methods using the TopK compressor without resorting to the BG/BGS assumptions. In particular, EF21-SGDM achieves *asymptotically optimal $\mathcal{O}\left(\sigma^2 n^{-1}\varepsilon^{-4}\right)$ sample complexity*. Moreover, when EF21-SGDM is applied with large enough batch size, we prove that it reaches the *optimal communication complexity $\mathcal{O}\left(K\alpha^{-1}\varepsilon^{-2}\right)$*; see Tables 1 & 2 for more details.

• Finally, we propose a *double momentum* variant of EF21-SGDM, and find that it further improves the sample complexity of EF21-SGDM in the non-asymptotic regime.

We highlight that, interestingly, we *prove that momentum helps*: EF21-SGDM is theoretically better compared to its non-momentum variant – large-batch EF21-SGD. We believe that our new technique can be extended in many ways, e.g., to dealing with other biased compressors and other (biased) communication saving techniques such as lazy aggregation of gradients, model compression, bidirectional compression, partial participation, decentralized training, adaptive compression; other important optimization techniques involving biased updates such as proximal SGD with momentum, gradient clipping and adaptive step-size schedules. We also hope that our proof techniques can be useful to establish linear speedup for other classes of distributed methods, e.g, algorithms based on local training ProxSkip/Scaffnew [Mishchenko et al., 2022].

Additionally, we extend our results to the class of absolute compressors in Appendix H and study the variance reduced variant of error feedback in Appendix I.

## 3 Main Results

Throughout the paper we work under the following standard assumptions.

**Assumption 1** (Smoothness and lower boundedness)**.** *We assume that $f$ has $L$-Lipschitz gradient, i.e., $\|\nabla f(x) - \nabla f(y)\| \leq L \|x - y\|$ for all $x, y \in \mathbb{R}^d$, and each $f_i$ has $L_i$-Lipschitz gradient,*

*i.e., $\|\nabla f_i(x) - \nabla f_i(y)\| \le L_i \|x - y\|$ for all $i \in [n]$, $x, y \in \mathbb{R}^d$. We denote $\widetilde{L}^2 := \frac{1}{n} \sum_{i=1}^n L_i^2$. Moreover, we assume that $f$ is lower bounded, i.e., $f^* := \inf_{x \in \mathbb{R}^d} f(x) > -\infty$.*

**Assumption 2** (Bounded variance (BV)). *There exists $\sigma > 0$ such that*

$$\mathbb{E}\left[\|\nabla f_i(x, \xi_i) - \nabla f_i(x)\|^2\right] \le \sigma^2, \qquad \forall x \in \mathbb{R}^d, \qquad \forall i \in [n], \tag{4}$$

*where $\xi_i \sim \mathcal{D}_i$ are i.i.d. random samples for each $i \in [n]$.*

### 3.1 A deeper dive into the issues EF21 has with stochastic gradients

As remarked before, the current analysis of EF21 in the stochastic setting requires each client to sample a mega-batch in each iteration, and it is not clear how to avoid this. In order to understand this phenomenon, we propose to step back and examine an "idealized" version of EF21-SGD, which we call EF21-SGD-ideal, defined by the update rules (5a) + (5aa):

$$x^{t+1} = x^t - \gamma g^t, \quad g^t = \frac{1}{n} \sum_{i=1}^n g_i^t \tag{5a}$$

$$\text{EF21-SGD-ideal:} \quad g_i^{t+1} = \nabla f_i(x^{t+1}) \quad + \mathcal{C}\left(\nabla f_i(x^{t+1}, \xi_i^{t+1}) - \nabla f_i(x^{t+1})\right), \tag{5aa}$$

$$\text{EF21-SGD:} \quad g_i^{t+1} = \quad g_i^t \quad + \mathcal{C}\left(\nabla f_i(x^{t+1}, \xi_i^{t+1}) - \quad g_i^t\right). \tag{5ab}$$

Compared to EF21-SGD, given by (5a) + (5ab), we replace the previous state $g_i^t$ by the *exact gradient* at the current iteration. Since EF21-SGD heavily relies on the approximation $g_i^t \approx \nabla f_i(x^{t+1})$, and according to the proof of convergence of EF21-SGD, such discrepancy tends to zero as $t \to \infty$, this change can only improve the method. While we admit this is a conceptual algorithm only (it does not lead to any communication or sample complexity reduction in practice)[8], it serves us well to illustrate the drawbacks of EF21-SGD. We now establish the following negative result for EF21-SGD-ideal.

**Theorem 1.** *Let $L, \sigma > 0$, $0 < \gamma \le 1/L$ and $n = 1$. There exists a convex, L-smooth function $f : \mathbb{R}^2 \to \mathbb{R}$, a contractive compressor $\mathcal{C}(\cdot)$ satisfying Definition 1, and an unbiased stochastic gradient with bounded variance $\sigma^2$ such that if the method EF21-SGD-ideal ((5a) + (5aa)) is run with step-size $\gamma$, then for all $T \ge 0$ and for all $x^0 \in \{(0, x_{(2)}^0)^\top \in \mathbb{R}^2 \mid x_{(2)}^0 < 0\}$, we have*

$$\mathbb{E}\left[\|\nabla f(x^T)\|^2\right] \ge \frac{1}{60} \min\left\{\sigma^2, \|\nabla f(x^0)\|^2\right\}.$$

*Fix $0 < \varepsilon \le L/\sqrt{60}$ and $x^0 = (0, -1)^\top$. Additionally assume that $n \ge 1$ and the variance of unbiased stochastic gradient is controlled by $\sigma^2/B$ for some $B \ge 1$. If $B < \frac{\sigma^2}{60\varepsilon^2}$, then we have $\mathbb{E}\left[\|\nabla f(x^T)\|\right] > \varepsilon$ for all $T \ge 0$.*

The above theorem implies that the method (5a), (5aa), does not converge with small batch-size (e.g., equal to one) for any fixed step-size choice.[9] Moreover, in distributed setting with $n$ nodes, a mini-batch of order $B = \Omega\left(\sigma^2/\varepsilon^2\right)$ is required for convergence. Notice that this batch-size is independent of $n$, which further implies that a linear speedup in the number of nodes $n$ cannot be achieved for this method. While we only prove these negative results for an "idealized" version of EF21-SGD rather than for the method itself, in Figures 1a and 4a, we empirically verify that EF21-SGD also suffers from a similar divergence on the same problem instance provided in the proof of Theorem 1. Additionally, Figures 1b and 4b illustrate that the situation does not improve for EF21-SGD when increasing $n$.

### 3.2 Momentum for avoiding mega-batches

Let us now focus on the single node setting[10] and try to fix the divergence issue shown above. As we can learn from Theorem 1, the key reason for non-convergence of EF21-SGD is that even if the state vector $g^t$ sufficiently approximates the current gradient, i.e., $g^t \approx \nabla f(x^{t+1})$, the design of

---

[8]This is because full gradients would need to be computed and communicated for its implementation. Notice also that if $\sigma = 0$, this method becomes the exact distributed gradient descent.

[9]In fact, the example can be easily extended to the case of polynomially decaying step-size.

[10]In this case, we can drop index $i$ everywhere and write $g_i^t = g^t$, $\xi_i^t = \xi^t$ for all $t \ge 0$.

this method cannot guarantee that the quantity $\|g^t - \nabla f(x^t)\|^2 \approx \|\mathcal{C}(\nabla f(x^t, \xi^t) - \nabla f(x^t))\|^2$ is small enough. Indeed, the last term above can be bounded by $2(2-\alpha)\sigma^2$ in expectation, but it is not sufficient as formally illustrated in Theorem 1. To fix this problem, we propose to modify our "idealized" EF21-SGD-ideal method so that the compressed difference can be controlled and made arbitrarily small, which leads us to another (more advanced) conceptual algorithm,

$$\text{EF21-SGDM-ideal:} \quad \begin{aligned} v^{t+1} &= \nabla f(x^{t+1}) + \eta(\nabla f(x^{t+1}, \xi^{t+1}) - \nabla f(x^{t+1})), \\ g^{t+1} &= \nabla f(x^{t+1}) + \mathcal{C}\left(v^{t+1} - \nabla f(x^{t+1})\right). \end{aligned} \tag{6}$$

In this method, instead of using $v^{t+1} = \nabla f(x^{t+1}, \xi^{t+1})$ as in EF21-SGD-ideal, we introduce a correction, which allows to control variance of the difference $\nabla f(x^{t+1}, \xi^{t+1}) - \nabla f(x^{t+1})$. This allows us to derive the following convergence result. Let $\delta_0 := f(x^0) - f^*$.

**Proposition 1.** *Let Assumptions 1, 2 hold, and let $\mathcal{C}$ satisfy Definition 1. Let $g^0 = 0$ and the step-size in method (5a), (6) be set as $\gamma \le 1/L$. Let $\hat{x}^T$ be sampled uniformly at random from the iterates of the method. Then for any $\eta > 0$ after $T$ iterations, we have $\mathbb{E}\left[\|\nabla f(\hat{x}^T)\|^2\right] \le \frac{2\delta_0}{\gamma T} + 4\eta^2\sigma^2$.*

Notice that if $\eta = 1$, then algorithm EF21-SGDM-ideal (5a), (6) reduces to EF21-SGD-ideal method (5a), (5aa), and this result shows that the lower bound for the batch-size established in Theorem 1 is tight, i.e., $B = \Theta(\sigma^2/\varepsilon^2)$ is necessary and sufficient[11] for convergence. For $\eta < 1$, the above theorem suggests that using a small enough parameter $\eta$, the variance term can be completely eliminated. This observation motivates us to design a practical variant of this method. Similarly to the design of EF21 mechanism (from EF21-SGD-ideal), we propose to do this by replacing the exact gradients $\nabla f(x^{t+1})$ by state vectors $v^t$ and $g^t$ as follows:

$$\text{EF21-SGDM:} \quad \begin{aligned} v^{t+1} &= v^t + \eta(\nabla f(x^{t+1}, \xi^{t+1}) - v^t), \\ g^{t+1} &= g^t + \mathcal{C}\left(v^{t+1} - g^t\right) \end{aligned} \tag{7}$$

**Theorem 2.** *Let Assumptions 1, 2 hold, and let $\mathcal{C}$ satisfy Definition 1. Let method (5a), (7) be run with $g^0 = v^0 = \nabla f(x^0)$, and $\hat{x}^T$ be sampled uniformly at random from the iterates of the method. Then for all $\eta \in (0, 1]$ with $\gamma \le \gamma_0 = \min\left\{\frac{\alpha}{20L}, \frac{\eta}{7L}\right\}$, we have $\mathbb{E}\left[\|\nabla f(\hat{x}^T)\|^2\right] \le \mathcal{O}(\frac{\delta_0}{\gamma T} + \eta\sigma^2)$. The choice $\eta = \min\left\{1, \left(\frac{L\delta_0}{\sigma^2 T}\right)^{1/2}\right\}$, $\gamma = \gamma_0$ results in $\mathbb{E}\left[\|\nabla f(\hat{x}^T)\|^2\right] \le \mathcal{O}(\frac{L\delta_0}{\alpha T} + \left(\frac{L\delta_0\sigma^2}{T}\right)^{1/2})$.*

Compared to Proposition 1, where $\eta$ can be made arbitrarily small, Theorem 2 suggests that there is a trade-off for the choice of $\eta \in (0, 1]$ in algorithm (5a), (7). The above theorem implies that in single node setting EF21-SGDM has $\mathcal{O}(\frac{L}{\alpha\varepsilon^2} + \frac{L\sigma^2}{\varepsilon^4})$ sample complexity. For $\alpha = 1$, this result matches with the sample complexity of SGD and is known to be unimprovable under Assumptions 1 and 2 [Arjevani et al., 2019]. Moreover, when $\alpha = 1$, our sample complexity matches with previous analysis of momentum methods in [Liu et al., 2020] and [Defazio, 2021]. However, even in this single node ($n = 1$), uncompressed ($\alpha = 1$) setting our analysis is different from the previous work, in particular, our choice of momentum parameter and the Lyapunov function are different, see Appendix A and J. For $\alpha < 1$, the above result matches with sample complexity of EF14-SGD (single node setting) [Stich and Karimireddy, 2019], which was recently shown to be optimal [Huang et al., 2022] for biased compressors satisfying Definition 1. However, notice that the extension of the analysis by Stich and Karimireddy [2019] for EF14-SGD to distributed setting meets additional challenges and it is unclear whether it is possible without imposing additional BG or BGS assumptions as in [Koloskova et al., 2020]. We revisit this analysis in Appendix K to showcase the difficulty of removing BG/BGS. In the following we will demonstrate the benefit of our EF21-SGDM method by extending it to distributed setting without imposing any additional assumptions.

### 3.3 Distributed stochastic error feedback with momentum

Now we are ready to present a distributed variant of EF21-SGDM, see Algorithm 1. Letting $\delta_t := f(x^t) - f^*$, our convergence analysis of this method relies on the monotonicity of the following Lyapunov function:

$$\Lambda_t := \delta_t + \frac{\gamma}{\alpha n} \sum_{i=1}^n \|g_i^t - v_i^t\|^2 + \frac{\gamma\eta}{\alpha^2 n} \sum_{i=1}^n \|v_i^t - \nabla f_i(x^t)\|^2 + \frac{\gamma}{\eta}\left\|\sum_{i=1}^n (v_i^t - \nabla f_i(x^t))\right\|^2. \tag{8}$$

---

[11]This follows by replacing $\sigma^2$ in the batch free algorithm by $\sigma^2/B$ if the batch-size of size $B > 1$ is used.

---

**Algorithm 1** EF21-SGDM (Error Feedback 2021 Enhanced with Polyak Momentum)

---

1: **Input:** starting point $x^0$, step-size $\gamma > 0$, momentum $\eta \in (0,1]$, initial batch size $B_{\text{init}}$
2: Initialize $v_i^0 = g_i^0 = \frac{1}{B_{\text{init}}} \sum_{j=1}^{B_{\text{init}}} \nabla f_i(x^0, \xi_{i,j}^0)$ for $i = 1, \ldots, n$; $g^0 = \frac{1}{n} \sum_{i=1}^{n} g_i^0$
3: **for** $t = 0, 1, 2, \ldots, T - 1$ **do**
4:     Master computes $x^{t+1} = x^t - \gamma g^t$ and broadcasts $x^{t+1}$ to all nodes
5:     **for all nodes** $i = 1, \ldots, n$ **in parallel do**
6:         Compute momentum estimator $v_i^{t+1} = (1 - \eta)v_i^t + \eta \nabla f_i(x^{t+1}, \xi_i^{t+1})$
7:         Compress $c_i^{t+1} = \mathcal{C}(v_i^{t+1} - g_i^t)$ and send $c_i^{t+1}$ to the master
8:         Update local state $g_i^{t+1} = g_i^t + c_i^{t+1}$
9:     **end for**
10:    Master computes $g^{t+1} = \frac{1}{n} \sum_{i=1}^{n} g_i^{t+1}$ via $g^{t+1} = g^t + \frac{1}{n} \sum_{i=1}^{n} c_i^{t+1}$
11: **end for**

---

● **Convergence of EF21-SGDM with contractive compressors.** We obtain the following result:

**Theorem 3.** *Let Assumptions 1 and 2 hold. Let $\hat{x}^T$ be sampled uniformly at random from the $T$ iterates of the method. Let EF21-SGDM (Algorithm 1) be run with a contractive compressor. For all $\eta \in (0, 1]$ and $B_{\text{init}} \geq 1$, with $\gamma \leq \min\left\{ \frac{\alpha}{20\widetilde{L}}, \frac{\eta}{7L} \right\}$, we have*

$$\mathbb{E}\left[ \left\| \nabla f(\hat{x}^T) \right\|^2 \right] \leq \mathcal{O}\left( \frac{\Lambda_0}{\gamma T} + \frac{\eta^3 \sigma^2}{\alpha^2} + \frac{\eta^2 \sigma^2}{\alpha} + \frac{\eta \sigma^2}{n} \right), \tag{9}$$

*where $\Lambda_0$ is given by (8). Choosing the batch size $B_{\text{init}} = \left\lceil \frac{\sigma^2}{L\delta_0} \right\rceil$, and stepsize $\gamma = \min\left\{ \frac{\alpha}{20\widetilde{L}}, \frac{\eta}{7L} \right\}$, and momentum $\eta = \min\left\{ 1, \left( \frac{L\delta_0 \alpha^2}{\sigma^2 T} \right)^{1/4}, \left( \frac{L\delta_0 \alpha}{\sigma^2 T} \right)^{1/3}, \left( \frac{L\delta_0 n}{\sigma^2 T} \right)^{1/2}, \frac{\alpha \sqrt{L\delta_0 B_{\text{init}}}}{\sigma} \right\}$, [12] we get*

$$\mathbb{E}\left[ \left\| \nabla f(\hat{x}^T) \right\|^2 \right] \leq \mathcal{O}\left( \frac{\widetilde{L}\delta_0}{\alpha T} + \left( \frac{L\delta_0 \sigma^{2/3}}{\alpha^{2/3} T} \right)^{3/4} + \left( \frac{L\delta_0 \sigma}{\sqrt{\alpha} T} \right)^{2/3} + \left( \frac{L\delta_0 \sigma^2}{n T} \right)^{1/2} \right).$$

**Remark 1.** *Note that using large initial batch size $B_{init} > 1$ is not necessary for convergence of EF21-SGDM. If we set $B_{init} = 1$, the above theorem still holds by replacing $\delta_0$ with $\Lambda_0$.*

**Remark 2.** *In the single node setting ($n = 1$), the above result recovers the statement of Theorem 2 (with the same choice of parameters) since by Young's inequality $\left( \frac{L\delta_0 \sigma^{2/3}}{\alpha^{2/3} T} \right)^{3/4} \leq \frac{1}{2} \frac{L\delta_0}{\alpha T} + \frac{1}{2} \left( \frac{L\delta_0 \sigma^2}{T} \right)^{1/2}, \left( \frac{L\delta_0 \sigma}{\sqrt{\alpha} T} \right)^{2/3} \leq \frac{1}{3} \frac{L\delta_0}{\alpha T} + \frac{2}{3} \left( \frac{L\delta_0 \sigma^2}{T} \right)^{1/2}$, and $\widetilde{L} = L$.*

● **Recovering previous rates in case of full gradients.** Compared to the iteration complexity $\mathcal{O}(\frac{L_{\max} G}{\alpha \varepsilon^3})$ of EF14 [Koloskova et al., 2020], our result, summarized in

**Corollary 1.** *If $\sigma = 0$, then $\mathbb{E}\left[ \left\| \nabla f(\hat{x}^T) \right\| \right] \leq \varepsilon$ after $T = \mathcal{O}\left( \frac{\widetilde{L}}{\alpha \varepsilon^2} \right)$ iterations.*

is better by an order of magnitude, and does not require the BG assumption. The result of Corollary 1 is the same as for EF21 method [Richtárik et al., 2021], and EF21-HB method [Fatkhullin et al., 2021]. Notice, however, that even in this deterministic setting ($\sigma = 0$) EF21-SGDM method is different from EF21 and EF21-HB: while the original EF21 does not use momentum, EF21-HB method incorporates momentum on the server side to update $x^t$, which is different from our Algorithm 1, where momentum is applied by each node. This iteration complexity $\mathcal{O}\left( \frac{1}{\alpha \varepsilon^2} \right)$ is optimal in both $\alpha$ and $\varepsilon$. The matching lower bound was recently established by Huang et al. [2022] for smooth nonconvex optimization in the class of centralized, zero-respecting algorithms with contractive compressors.

● **Comparison to previous work.** Our sample complexity[13] in

**Corollary 2.** $\mathbb{E}\left[ \left\| \nabla f(\hat{x}^T) \right\| \right] \leq \varepsilon$ *after $T = \mathcal{O}\left( \frac{\widetilde{L}}{\alpha \varepsilon^2} + \frac{L\sigma^{2/3}}{\alpha^{2/3} \varepsilon^{8/3}} + \frac{L\sigma}{\alpha^{1/2} \varepsilon^3} + \frac{L\sigma^2}{n \varepsilon^4} \right)$ iterations.*

---

[12]In Appendix J, we show how to deal with time varying $\gamma_t$ and $\eta_t$.

[13]Note that the initial batch size contributes to the sample complexity only an additive constant independent of $\varepsilon$. Moreover, $B_{\text{init}} = \left\lceil \frac{\sigma^2}{L\delta_0} \right\rceil \leq \left\lceil \frac{2\sigma^2}{\varepsilon^2} \right\rceil$ since, otherwise, $\left\| \nabla f(x^0) \right\|^2 \leq 2L\delta_0 \leq \varepsilon^2$, and $x^0$ is a solution. In the following, we ignore the dependece on $B_{\text{init}}$ for a fair comparison with other works.

strictly improves over the complexity $\mathcal{O}(\frac{GL_{\max}}{\alpha\varepsilon^3} + \frac{L_{\max}\sigma^2}{n\varepsilon^4})$ of EF14-SGD by Koloskova et al. [2020], even in case when $G < +\infty$. Notice that it always holds that $\sigma \leq G$. If we assume that $G \approx \sigma$, our three first terms in the complexity improve the first term from Koloskova et al. [2020] by the factor of $\varepsilon/\sigma$, $(\varepsilon\alpha/\sigma)^{1/3}$, or $\alpha^{1/2}$. Compared to the BEER algorithm of Zhao et al. [2022], with sample complexity $\mathcal{O}(\frac{L_{\max}}{\alpha\varepsilon} + \frac{L_{\max}\sigma^2}{\alpha^2\varepsilon^4})$, the result of Corollary 2 is strictly better in terms of $\alpha$, $n$, and the smoothness constants.[14] In addition, we remove the large batch requirement for convergence compared to [Fatkhullin et al., 2021, Zhao et al., 2022]. Moreover, notice that Corollary 2 implies that EF21-SGDM achieves asymptotically optimal sample complexity $\mathcal{O}(\frac{L\sigma^2}{n\varepsilon^4})$ in the regime $\varepsilon \to 0$.

### 3.4 Further improvement using *double* momentum!

Unfortunately, in the non-asymptotic regime, our sample complexity does not match with the lower bound in all problem parameters simultanuously due to the middle term $\frac{L\sigma^{2/3}}{\alpha^{2/3}\varepsilon^{8/3}} + \frac{L\sigma}{\alpha^{1/2}\varepsilon^3}$, which can potentially dominate over $\frac{L\sigma^2}{n\varepsilon^4}$ term for large enough $n$ and $\varepsilon$, and small enough $\alpha$ and $\sigma$. We propose a *double-momentum* method, which can further improve the middle term in the sample complexity of EF21-SGDM. We replace the momentum estimator $v_i^t$ in line 6 of Algorithm 1 by the following two-step momentum update

$$\text{EF21-SGD2M:} \quad v_i^{t+1} = (1-\eta)v_i^t + \eta\nabla f_i(x^{t+1}, \xi_i^{t+1}), \quad u_i^{t+1} = (1-\eta)u_i^t + \eta v_i^{t+1}. \quad (10)$$

We formally present this method in Algorithm 3 in Appendix G. Compared to EF21-SGDM (Algorithm 1), the only change is that instead of compressing $v_i^t - g_i^t$, in EF21-SGD2M, we compress $u_i^t - g_i^t$, where $u_i$ is a two step (double) momentum estimator. The intuition behind this modification is that a double momentum estimator $u_i^t$ has richer "memory" of the past gradients compared to $v_i^t$. Notice that for each node, EF21-SGD2M requires to save 3 vectors ($v_i^t$, $u_i^t$, $g_i^t$) instead of 2 in EF21-SGDM ($v_i^t$, $g_i^t$) and EF14-SGD ($e_i^t$, $g_i^t$).[15] When interacting with biased compression operator $\mathcal{C}(\cdot)$, such effect becomes crucial in improving the sample complexity. For EF21-SGD2M, we derive

**Corollary 3.** *Let $v_i^t$ in Algorithm 1 be replaced by $u_i^t$ given by (10) (Algorithm 3 in Appendix G). Then with appropriate choice of $\gamma$ and $\eta$ (given in Theorem 5), we have $\mathbb{E}\left[\left\|\nabla f(\hat{x}^T)\right\|\right] \leq \varepsilon$ after $T = \mathcal{O}\left(\frac{\widetilde{L}\delta_0}{\alpha\varepsilon^2} + \frac{L\delta_0\sigma^{2/3}}{\alpha^{2/3}\varepsilon^{8/3}} + \frac{L\delta_0\sigma^2}{n\varepsilon^4}\right)$ iterations.*

## 4 Experiments

We consider a nonconvex logistic regression problem: $f_i(x_1, \ldots, x_c) = -\frac{1}{m}\sum_{j=1}^m \log(\exp(a_{ij}^\top x_{y_{ij}})/\sum_{y=1}^c \exp(a_{ij}^\top x_y))$ with a nonconvex regularizer $h(x_1, \ldots, x_c) = \lambda\sum_{y=1}^c\sum_{k=1}^l [x_y]_k^2/(1 + [x_y]_k^2)$ with $\lambda = 10^{-3}$, where $x_1, \ldots, x_c \in \mathbb{R}^l$, $[\cdot]_k$ is an indexing operation of a vector, $c \geq 2$ is the number of classes, $l$ is the number of features, $m$ is the size of a dataset, $a_{ij} \in \mathbb{R}^l$ and $y_{ij} \in \{1, \ldots, c\}$ are features and labels. The datasets used are *MNIST* (with $l = 784$, $m = 60\,000$, $c = 10$) and *real-sim* (with $l = 20\,958$, $m = 72\,309$, $c = 2$) [LeCun et al., 2010, Chang and Lin, 2011]. The dimension of the problem is $d = (l + 1)c$, i.e., $d = 7\,850$ for *MNIST* and $d = 41\,918$ for *real-sim*. In each experiment, we show relations between the total number of transmitted coordinates and gradient/function values. The stochastic gradients in each algorithm are replaced by a mini-batch estimator $\frac{1}{B}\sum_{j=1}^B \nabla f_i(x, \xi_{ij})$ with the same $B \geq 1$ in each plot. Notice that all methods (except for NEOLITHIC)[16] calculate the same number of samples at each communication round, thus the dependence on the number of samples used will be qualitatively the same. In all algorithms, the step sizes are fine-tuned from a set $\{2^k \mid k \in [-20, 20]\}$ and the TopK compressor is used to compress information from the nodes to the master. For EF21-SGDM, we fix momentum parameter $\eta = 0.1$ in all experiments. Prior to that, we tuned $\eta \in \{0.01, 0.1\}$ on the independent dataset *w8a* (with $l = 300$, $m = 49\,749$, $c = 2$). We omit BEER method from the plots since it showed worse performance than EF21-SGD in all runs.

---

[14]$L_{\max} := \max_{i\in[n]} L_i$. Notice that $L \leq \widetilde{L} \leq L_{\max}$ and the inequalities are strict in heterogeneous setting.

[15]See Appendix K for details on EF14-SGD. In contrast, EF21-SGD needs to save only one vector ($g_i^t$).

[16]For NEOLITHIC, we use the parameter $R = \lceil d/K \rceil$ following the requirement in their Theorem 3. Experiments in Huang et al. [2022] use a heuristic choice $R = 4$, and thus can show faster convergence.

## 4.1 Experiment 1: increasing batch-size

In this experiment, we use *MNIST* dataset and fix the number of transmitted coordinates to $K = 10$ (thus $\alpha \geq {}^K/_d \approx 10^{-3}$), and set $n = 10$. Figure 2 shows convergence plots for $B \in \{1, 32, 128\}$. EF21-SGDM and its double momentum version EF21-SGD2M have fast convergence and show a significant improvement when increasing batch-size compared to EF14-SGD. In contrast, EF21-SGD suffers from poor performance for small $B$, which confirms our observations in previous sections. NEOLITHIC has order times slower convergence rate due to the fact that it sends $\lceil {}^d/_K \rceil$ compressed vectors in each iteration, while other methods send only one.

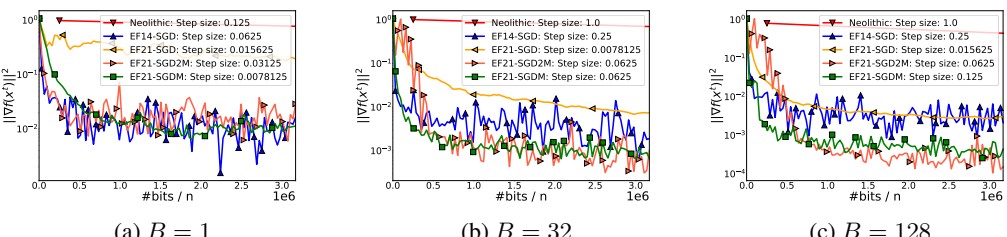

(a) $B = 1$      (b) $B = 32$      (c) $B = 128$

Figure 2: Experiment on *MNIST* dataset with $n = 10$, and Top10 compressor.

## 4.2 Experiment 2: improving convergence with $n$

This experiment uses *real-sim* dataset, $K = 100$ (thus $\alpha \geq {}^K/_d \approx 2 \cdot 10^{-3}$), and with $B = 128 \ll m$. We vary the number of nodes within $n \in \{1, 10, 100\}$, see Figure 3. In this case, EF21-SGDM and EF21-SGD2M have much faster convergence compared to other methods for all $n$. Moreover, the proposed algorithms show a significant improvement when $n$ increases. We also observe that on this task, EF21-SGD2M performs slightly worse than EF21-SGDM for $n = 10, 100$, but it is still much faster than other other methods.

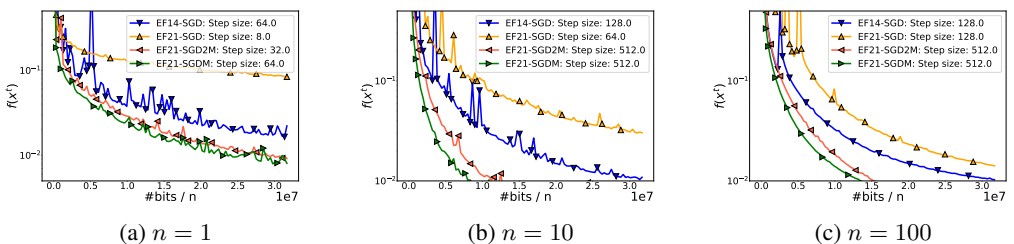

(a) $n = 1$      (b) $n = 10$      (c) $n = 100$

Figure 3: Experiment on *real-sim* dataset with batch-size $B = 128$, and Top100 compressor.

In Section C, we present extra simulations with different parameters for above experiments. Additionally, we inlcude experiemnts on simple quadratic problems and perform training of larger image recognition models. In all cases, EF21-SGDM and EF21-SGD2M outperform other algorithms.

## Acknowledgments and Disclosure of Funding

This work of I. Fatkhullin was supported by ETH AI Center doctoral fellowship. The work of P. Richtárik and A. Tyurin was supported by the KAUST Baseline Research Scheme (KAUST BRF) and the KAUST Extreme Computing Research Center (KAUST ECRC); P. Richtárik was also supported by the SDAIA-KAUST Center of Excellence in Data Science and Artificial Intelligence (SDAIA-KAUST AI).

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

# Contents

| Method | Comm. compl. | Batch-size for comm. compl. | Asymp. sample compl. | Batch free? | No extra assump.? |
|---|---|---|---|---|---|
| EF14-SGD [Koloskova et al., 2020] | $\frac{KGL_{\max}}{\alpha\varepsilon^3}$ | $\frac{\alpha\sigma^2}{n\varepsilon G}$ (*) | $\frac{L_{\max}\sigma^2}{n\varepsilon^4}$ | ✔ | ✗ (a) |
| NEOLITHIC [Huang et al., 2022] | $\frac{KL_{\max}}{\alpha\varepsilon^2}\log\left(\frac{G}{\varepsilon}\right)$ (b) | $\frac{\sigma^2}{n\varepsilon^2}\vee\frac{1}{\alpha}\log\left(\frac{G}{\varepsilon}\right)$ (*) | $\frac{L_{\max}\sigma^2}{n\varepsilon^4}$ | ✗ | ✗ (c) |
| EF21-SGD [Fatkhullin et al., 2021] | $\frac{K\widetilde{L}}{\alpha\varepsilon^2}$ | $\frac{\sigma^2}{\alpha^2\varepsilon^2}$ | $\frac{\widetilde{L}\sigma^2}{\alpha^3\varepsilon^4}$ (d) | ✗ | ✔ |
| BEER [Zhao et al., 2022] | $\frac{KL_{\max}}{\alpha\varepsilon^2}$ | $\frac{\sigma^2}{\alpha\varepsilon^2}$ | $\frac{L_{\max}\sigma^2}{\alpha^2\varepsilon^4}$ (d) | ✗ | ✔ |
| EF21-SGDM Corollary 2 | $\frac{K\widetilde{L}}{\alpha\varepsilon^2}$ | $\frac{\alpha L}{\widetilde{L}}\frac{\sigma^2}{n\varepsilon^2}\vee\frac{\alpha L^2}{\widetilde{L}^2}\frac{\sigma^2}{\varepsilon^2}$ | $\frac{L\sigma^2}{n\varepsilon^4}$ | ✔ | ✔ |
| EF21-SGD2M Corollary 3 | $\frac{K\widetilde{L}}{\alpha\varepsilon^2}$ | $\frac{\alpha L}{\widetilde{L}}\frac{\sigma^2}{n\varepsilon^2}\vee\frac{\alpha L^3}{\widetilde{L}^3}\frac{\sigma^2}{\varepsilon^2}$ | $\frac{L\sigma^2}{n\varepsilon^4}$ | ✔ | ✔ |

(a) Analysis requires a bound of the second moment of the stochastic gradients, i.e., $\mathbb{E}\left[\|\nabla f_i(x,\xi_i)\|^2\right]\leq G^2$ for all $x\in\mathbb{R}^d$.

(b) This complexity is achieved by using a large mini-batch and communicating $\lceil K/\alpha\rceil\approx d$ coordinates per iteration.

(c) Analysis requires a bounded gradient disimilarity assumption, i.e., $\frac{1}{n}\sum_{i=1}^n\|\nabla f_i(x)-\nabla f(x)\|^2\leq G^2$ for all $x\in\mathbb{R}^d$.

(d) Analysis requires a batch-size at least $B\geq\frac{\sigma^2}{\alpha^2\varepsilon^2}$ for EF21-SGD and $B\geq\frac{\sigma^2}{\alpha\varepsilon^2}$ for BEER.

(*) For a fair comparison, we take the (minimal) batch-size for these methods which guarantees the reported communication complexity.

Table 2: Extended summary of related works on distributed error compensated SGD methods using a TopK compressor under Assumptions 1 and 2. The goal is to find an $\varepsilon$-stationary point of a smooth nonconvex function of the form (1), i.e., a point $x$ such that $\mathbb{E}\left[\|\nabla f(x)\|\right]\leq\varepsilon$. "**Comm. compl.**" reports the total number of communicated bits if the method is applied with batch-size equal to "Batch-size" at each node. When TopK compressor is applied, then $\alpha\geq K/d$, and the comm. compl. of error compensated methods can be reduced by a factor of $\alpha d/K$. "**Batch-size for comm. compl.**" means the batch-size for achieving the reported "Comm. compl.". "**Asymp. sample compl.**" reports the asymptotic sample complexity of the algorithm with batch-size $B=1$ in the regime $\varepsilon\to0$, i.e., the total number of samples required at each node to achieve $\varepsilon$-stationary point. "**Batch free**" marks with ✔ if the analysis ensures convergence with batch-size equal to 1. "**No extra assump.**" marks with ✔ if no additional assumption beyond Assumptions 1 and 2 is required for analysis. We denote $L_{\max}:=\max_{i\in[n]}L_i$. Notice that it always holds $L\leq\widetilde{L}\leq L_{\max}$ and these inequalities only become equalities in the homogeneous case. It could be that $\alpha L/\widetilde{L}\ll1$ making the batch-size of EF21-SGDM and EF21-SGD2M much smaller than those of EF21-SGD and BEER. Symbol $\vee$ denotes the maximum of two scalars.

# A    More on Contractive Compressors, Error Feedback and Momentum

**Greedy vs uniform.**    In our work, we specifically focus on the class of contractive compressors satisfying Definition 1, which contains a greedy TopK compressor as a special case. Note that TopK is greedy in that it minimizes the error $\|\text{Top}K(x)-x\|^2$ subject to the sparsity constraint $\|\mathcal{C}(x)\|_0\leq K$, where $\|u\|_0$ counts the number of nonzero entries in $u$. In practice, greediness is almost always[17] very useful, translating into excellent empirical performance, especially when compared to the performance of the RandK sparsifier. On the other hand, it appears to be very hard to formalize these practical gains theoretically[18]. In fact, while greedy compressors such as TopK outperform their randomized cousins such as RandK in practice, and often by huge margins [Lin et al., 2018], the theoretical picture is exactly reversed, and the theoretical communication complexity of gradient-type methods based on randomized compressors [Alistarh et al., 2017, Mishchenko et al., 2019, Horváth et al., 2019b, Li et al., 2020, Gorbunov et al., 2021] is much better than of those based on greedy compressors [Koloskova et al., 2020, Richtárik et al., 2021, Fatkhullin et al., 2021, Richtárik et al., 2022]. The key reason behind this is the fact that popular randomized compressors such as RandK become *unbiased* mappings after appropriate scaling (e.g., $\mathbb{E}[\frac{d}{K}\text{Rand}K(x)]\equiv x$), and that the inherent randomness is typically drawn *independently* by all clients. This leads to several key simplifications in the analysis, and consequently, to theoretical gains over methods that do not compress, and over methods that compress greedily. Further improvements are possible when the randomness is *correlated* in an appropriate way [Szlendak et al., 2021].

Due to the superior empirical properties of greedy contractive compressors, and our desire to push this very potent line of work further, in this paper we work with the general class of compressors

---

[17]Greediness is not useful, for example, when $\mathcal{D}_i=\mathcal{D}_j$ for all $i,j$ and when TopK is applied to the full-batch gradient $\nabla f_i(x)$ by each client. However, situations of this type arise rarely in practice.

[18]No theoretical results of this type exist for $n>1$.

satisfying Definition 1, and do not invoke any additional restrictive assumptions. For example, we do not assume $\mathcal{C}$ can be made unbiased after scaling.

**Error Feedback.** The first theoretical analysis of EF14 was presented in the works of Stich et al. [2018], Alistarh et al. [2018] and further revisited in convex case in [Karimireddy et al., 2019, Beznosikov et al., 2020, Gorbunov et al., 2020, Qian et al., 2020] and analysis was extended to nonconvex setting in [Stich and Karimireddy, 2019]. Later, in nonconvex case, various extensions and combinations of EF14 with other optimization techniques were considered and analyzed, which include bidirectional compression [Tang et al., 2020], decentralized training [Koloskova et al., 2020, Singh et al., 2021], server level momentum [Xie et al., 2020], client level momentum [Zheng et al., 2019], combination with adaptive methods [Li et al., 2022b]. To our knowledge, the best sample complexity for finding a stationary point for this method (including its momentum and adaptive variants) in the distributed nonconvex setting is given by Koloskova et al. [2020], which is $\mathcal{O}(\frac{G}{\alpha\varepsilon^3} + \frac{\sigma^2}{n\varepsilon^4})$. More recently, Huang et al. [2022] propose a modification of EF14 method achieving $\mathcal{O}\left(\frac{1}{\alpha\varepsilon^2}\log(\frac{G}{\varepsilon}) + \frac{\sigma^2}{n\varepsilon^4}\right)$ sample complexity by using the BGS assumption. When applied with TopK compressor, this method requires to communicate $\widetilde{\mathcal{O}}(K/\alpha)$ coordinates at every iteration. This makes it impractical since when the effective $\alpha$ is unknown and is set to $\alpha = K/d$, it means that their method communicates all $d$ coordinates at every iteration, mimicking vanilla (S)GD method. Moreover, their algorithm uses an additional subroutine and applies updates with a large batch-size of samples of order $\mathcal{O}(\frac{1}{\alpha}\log\left(\frac{G}{\varepsilon}\right))$, making the algorithm less practical and difficult to implement. It is worth to mention, that error feedback was also analyzed for other classes of compressors such as absolute (see Definition 2) or additive compressors (i.e., $\mathcal{C}(x + y) = \mathcal{C}(x) + \mathcal{C}(y)$ for all $x, y \in \mathbb{R}^d$) [Tang et al., 2020, Xu and Huang, 2022], which do not include TopK sparsifier.

**Momentum.** The first convergence analysis of gradient descent with momentum was proposed by B.T. Polyak in his seminal work [Polyak, 1964] studying the benefit of multi-step methods. The proof technique proposed in this work is based on the analysis of the spectral norm of a certain matrix arising from the dynamics of a multi-step process on a quadratic function. Unfortunately, such technique is restricted to the case of strongly convex quadratic objective and the setting of full gradients. Later Zavriev and Kostyuk [1993] prove an asymptotic convergence of this method in nonconvex deterministic case without specifying the rate of convergence.

To our knowledge, the first non-asymptotic analysis of SGDM in the smooth nonconvex setting is due to Yu et al. [2019]. Their analysis, however, heavily relies on BG assumption. Later, Liu et al. [2020] provide a refined analysis of SGDM, removing the BG assumption and improving the dependence on the momentum parameter $\eta$. Notice that the analysis of Liu et al. [2020] and the majority of other works relies on some variant of the following Lyapunov function:

$$\Lambda_t := f(z^t) - f^* + \sum_{\tau=0}^{t} c_\tau \left\| x^{t-\tau} - x^{t-1-\tau} \right\|^2, \tag{11}$$

where $\{z^t\}_{t\geq 0}$ is some auxiliary sequence (often) different from the iterates $\{x^t\}_{t\geq 0}$, and $\{c_\tau\}_{\tau\geq 0}$ is a diminishing non-negative sequence. This approach is motivated by the dynamical system point of view at Polyak's heavy ball momentum, where the two terms in (11) are interpreted as the potential and kinetic energy of the system [Sebbouh et al., 2019]. In contrast, the Lyapunov function used in this work is conceptually different even in the single node ($n = 1$), uncompressed ($\alpha = 1$) setting. Later, Defazio [2021] revisit the analysis in [Liu et al., 2020] through the lens of primal averaging and provide insights on why momentum helps in practice. The momentum is also used for stabilizing adaptive algorithms such as normalized SGD [Cutkosky and Mehta, 2020]. In particular, it was shown that by using momentum, one can ensure convergence without large batches for normalized SGD (while keeping the same sample complexity as a large batch normalized SGD). However, their analysis is specific to the normalized method, which allows using the function value as a Lyapunov function. High probability analysis of momentum methods was investigated in [Cutkosky and Mehta, 2021, Li and Orabona, 2020]. In the distributed setting, [Yu et al., 2019, Karimireddy et al., 2021] extend the analysis of SGDM under BGS assumption. Later [Takezawa et al., 2022, Gao et al., 2023] remove this assumption providing a refined analysis based on the techniques developed in [Liu et al., 2020]. However, the algorithms in these works do not apply any bandwidth reduction technique such as communication compression.

We would like to mention that understanding the behavior of SGDM in convex case also remains an active area of research [Ghadimi et al., 2015, Yang et al., 2016, Sebbouh et al., 2021, Li et al., 2022a, Xiao and Yang, 2022] .

# B  Variance Reduction Effect of SGDM and Comparison to STORM

Notice that the choice of our Lyapunov function $\Lambda_t$ (8), which is used in the analysis of EF21-SGDM implies that the gradient estimators $g_i^t$ and $v_i^t$ improve over the iterations, i.e.,

$$g_i^t \to \nabla f_i(x^t), \qquad v_i^t \to \nabla f_i(x^t) \qquad \text{for } t \to \infty.$$

This comes in contrast with the behavior of SGD, for which the gradient estimator $v_i^t = \nabla f_i(x^t, \xi_i^t)$ does not necessarily tend to zero over iterations. Such effect of asymptotic improvement of the estimation error of the gradient estimator is reminiscent to the analogous effect known in the literature on variance reduction (VR) methods. In particular, the classical momentum step 6 of Algorithm 1 may be contrasted with a STORM variance reduced estimator proposed by Cutkosky and Orabona [2019], which updates the gradient estimator via

$$w_i^{t+1} = \nabla f_i(x^{t+1}, \xi_i^{t+1}) + (1 - \eta)(w_i^t - \nabla f_i(x^t, \xi_i^{t+1})), \quad w_i^0 = \nabla f_i(x^0, \xi_i^0) \tag{12}$$

It is known that the class of VR methods (and STORM, in particular) can show faster asymptotic convergence in terms of $T$ (or $\varepsilon$) compared to SGD and SGDM under some additional assumptions. However, we would like to point out the important differences (and limitations) of (12) compared to the classical Polyak's momentum used on line 6 of Algorithm 1. First, the estimator $w_i^{t+1}$ is different from the momentum update rule $v_i^{t+1}$ in that it is unbiased for any $t \geq 0$, i.e., $\mathbb{E}\left[w_i^{t+1} - \nabla f_i(x^{t+1})\right] = 0,$[19] which greatly facilitates the analysis of this method. Notice that, in particular, in the deterministic case ($\sigma = 0$, $\alpha = 1$), the method with estimator (12) reduces to vanilla gradient descent with $w_i^{t+1} = \nabla f_i(x^{t+1})$. Second, the computation of $w_i^{t+1}$ requires access to two stochastic gradients $\nabla f_i(x^{t+1}, \xi_i^{t+1})$ and $\nabla f_i(x^t, \xi_i^{t+1})$ under the same realization of noise $\xi_i^{t+1}$ at each iteration, and requires the additional storage of control variate $x^t$. This is a serious limitation, which can make the method impractical or even not implementable for certain applications such as federated RL [Mitra et al., 2023], multi-agent RL [Doan et al., 2019] or operations research problems [Chen et al., 2022]. Third, the analysis of variance reduced methods such as STORM requires an additional assumptions such as individual smoothness of stochastic functions (or its averaged variants) (Assumption 3), i.e., $\|\nabla f_i(x, \xi_i) - \nabla f_i(y, \xi_i)\| \leq \ell_i \|x - y\|$ for all $x, y \in \mathbb{R}^d$, $\xi_i \sim \mathcal{D}_i$, $i \in [n]$, while our EF21-SGDM only needs smoothness of (deterministic) local functions $f_i(x)$. While this assumption is satisfied for some loss functions in supervised learning, it can also be very limiting. Even if Assumption 3 is satisfied, the constant $\widetilde{\ell}$ (which always satisfies $\widetilde{\ell} \geq \widetilde{L}$) can be much larger than $\widetilde{L}$ canceling the speed-up in terms of $T$ (or $\varepsilon$). For completeness, we provide the sample complexity analysis of our error compensated method combined with estimator (12), which is deferred to Appendix I.

---

[19]Notice that $\mathbb{E}\left[w_i^0 - \nabla f_i(x^0)\right] = 0$. Let $\mathbb{E}\left[w_i^t - \nabla f_i(x^t)\right] = 0$ hold, then

$$\mathbb{E}\left[w_i^{t+1} - \nabla f_i(x^{t+1})\right] = (1 - \eta)\mathbb{E}\left[w_i^t - \nabla f_i(x^t, \xi_i^{t+1})\right] = 0.$$

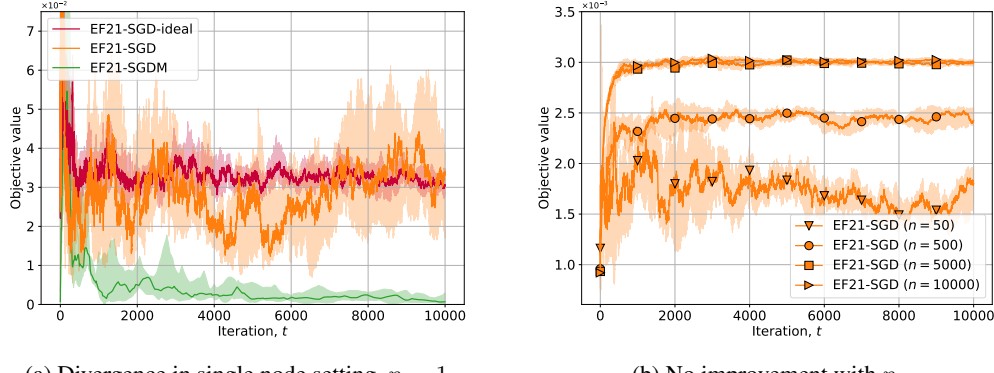

(a) Divergence in single node setting, $n = 1$.      (b) No improvement with $n$.

Figure 4: Divergence of EF21-SGD on a quadratic function $\frac{1}{2}\|x\|^2$ with Top1 compressor. See the proof of Therem 1 for details on the construction of noise $\xi$, we use $\sigma = 1$, $B = 1$. The starting point for all methods is $x^0 = (0, -0.01)^\top$. Unlike Figure 1, these experiments use time varying step-sizes and momentum parameters $\gamma_t = \eta_t = \frac{0.1}{\sqrt{t+1}}$. Each method is run 10 times and the plot shows the median performance alongside the $25\%$ and $75\%$ quantiles.

## C Additional Experiments and Details of Experimental Setup

**Divergence of EF21-SGD with time-varying step-sizes.**  We complement our Figure 1 in the main part of the paper, which shows divergence of EF21-SGD [Fatkhullin et al., 2021] with small (constant) step-size. Here, in Figure 4, we see that the similar divergence is observed when using time varying step-sizes $\gamma_t = \frac{0.1}{\sqrt{t+1}}$. Also, EF21-SGD with time-varying step-size does not improve convergence when $n$ is increased.

**Implementation Details.**  The experiments were implemented in Python 3.7.9. The distributed environment was emulated on machines with Intel(R) Xeon(R) Gold 6248 CPU @ 2.50GHz. In all experiments with *MNIST*, we split the dataset across nodes by labels to simulate the heterogeneous setting.

### C.1 Extra plots for experiments 1 and 2

In Figures 5 and 6, we provide extra experiments for the setup from Section 4.

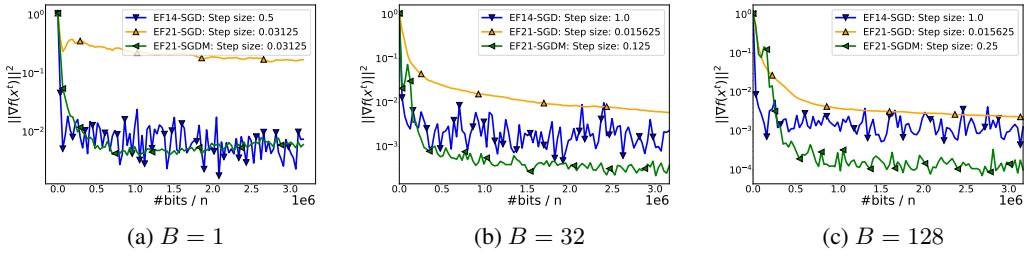

(a) $B = 1$         (b) $B = 32$         (c) $B = 128$

Figure 5: Performance of algorithms on *MNIST* dataset with $n = 100$, and Top10 compressor.

### C.2 Experiment 3: stochastic quadratic optimization

We now consider a synthetic $\lambda$–strongly convex quadratic function problem $f(x) = \frac{1}{n}\sum_{i=1}^{n} f_i(x)$, where the functions $f_i(x) = \frac{1}{2}x^\top \mathbf{Q}_i x - x^\top b_i$ are (not necessarily convex) quadratic functions for all $i \in [n]$ and $x \in \mathbb{R}^d$. The matrices $\mathbf{Q}_1, \cdots, \mathbf{Q}_n$, vectors $b_1, \cdots, b_n$, and a starting point $x^0$ are generated by Algorithm 2 with the number of nodes $n = 100$, dimension $d = 1000$, regularizer $\lambda = 0.01$, and scale $s = 1$. For all $i \in [n]$ and $x \in \mathbb{R}^d$, we consider stochastic gradients $\nabla f_i(x, \xi) = \nabla f_i(x) + \xi_i$, where $\xi_i$ are i.i.d. samples from $\mathcal{N}(0, \sigma)$ with $\sigma \in \{0.001, 0.01\}$. In

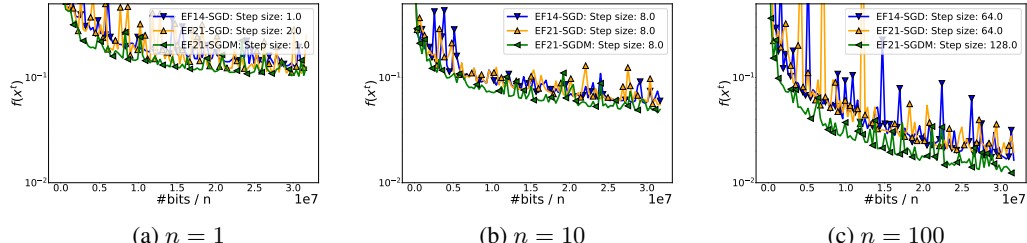

(a) $n = 1$        (b) $n = 10$        (c) $n = 100$

Figure 6: Performance of algorithms on *real-sim* dataset with batch-size $B = 1$, and Top100 compressor.

Figure 7, we present the comparison of EF21-SGDM and EF14-SGD with three different step sizes. The behavior of methods for other step sizes from the set $\{2^k \mid k \in [-20, 20]\}$ follows a similar trend. For every step size, we observe that at the beginning, the methods have almost the same linear rates, but then EF14-SGD gets stuck at high accuracies, while EF21-SGDM continues converging to the lower accuracies.

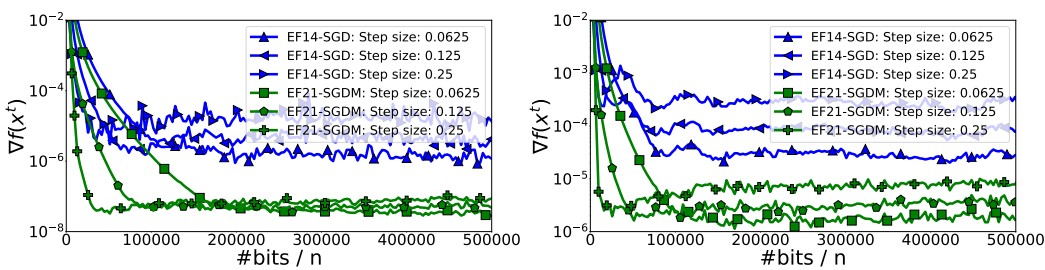

Figure 7: Stochastic Quadratic Optimization Problem with $\sigma = 0.001$ (left figure) and $\sigma = 0.01$ (right figure)

---

**Algorithm 2** Quadratic Optimization Task Generation Procedure

---

1: **Parameters:** number nodes $n$, dimension $d$, regularizer $\lambda$, and scale $s$.
2: **for** $i = 1, \ldots, n$ **do**
3:      Calculate Guassian noises $\mu_i^s = 1 + s\xi_i^s$ and $\mu_i^b = s\xi_i^b$, i.i.d. $\xi_i^s, \xi_i^b \sim \mathcal{N}(0, 1)$
4:      $b_i = \frac{\mu_i^s}{4}(-1 + \mu_i^b, 0, \cdots, 0) \in \mathbb{R}^d$
5:      Scale the predefined tridiagonal matrix

$$\mathbf{Q}_i = \frac{\mu_i^s}{4} \begin{pmatrix} 2 & -1 & & 0 \\ -1 & \ddots & \ddots & \\ & \ddots & \ddots & -1 \\ 0 & & -1 & 2 \end{pmatrix} \in \mathbb{R}^{d \times d}$$

6: **end for**
7: Find the mean of matrices $\mathbf{Q} = \frac{1}{n} \sum_{i=1}^{n} \mathbf{Q}_i$
8: Find the minimum eigenvalue $\lambda_{\min}(\mathbf{Q})$
9: **for** $i = 1, \ldots, n$ **do**
10:      Normalize matrix $\mathbf{Q}_i = \mathbf{Q}_i + (\lambda - \lambda_{\min}(\mathbf{Q}))\mathbf{I}$
11: **end for**
12: Find a starting point $x^0 = (\sqrt{d}, 0, \cdots, 0)$
13: **Output a new problem:** matrices $\mathbf{Q}_1, \cdots, \mathbf{Q}_n$, vectors $b_1, \cdots, b_n$, starting point $x^0$

---

**A procedure to generate stochastic quadratic optimization problems.** In this section, we present an algorithm that generates quadratic optimization tasks. The formal description is provided in Algorithm 2. The idea is to take a predefined tridiagonal matrix and add noises to simulate the heterogeneous setting. Algorithm 2 returns matrices $\mathbf{Q}_1, \cdots, \mathbf{Q}_n$, vectors $b_1, \cdots, b_n$, and a starting

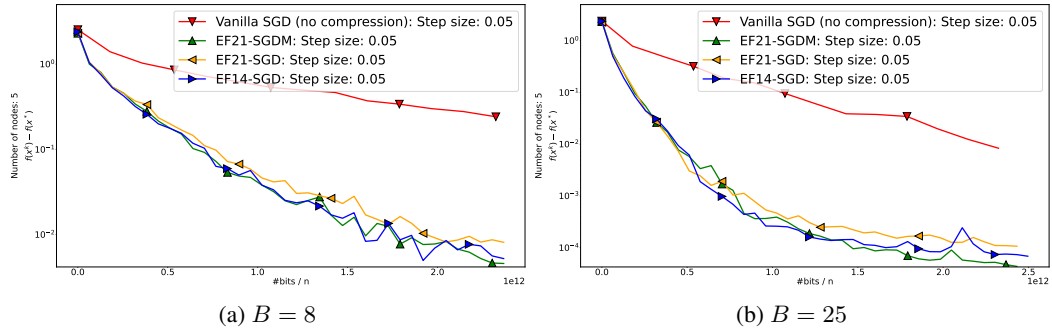

(a) $B = 8$            (b) $B = 25$

Figure 8: *ResNet-18* on *CIFAR10* dataset with $n = 5$.

| Algorithm | Test Accuracy |
|-----------|---------------|
| SGD | 81.5 % |
| EF21-SGD | 82.5 % |
| EF14-SGD | 83.1 % |
| EF21-SGDM | **83.3** % |

Figure 9: Accuracy on the *CIFAR10* test split.

point $x^0$ such that the matrix $\mathbf{Q} = \frac{1}{n} \sum_{i=1}^{n} \mathbf{Q}_i$ has the minimum eigenvalue $\lambda_{\min}(\mathbf{Q}) = \lambda$, where $\lambda \geq 0$ is a parameter. Next, we define the functions $f_i$ and stochastic gradients in the following way:

$$f_i(x) := \frac{1}{2} x^\top \mathbf{Q}_i x - x^\top b_i$$

and

$$\nabla f_i(x, \xi) := \nabla f_i(x) + \xi_i,$$

for all $x \in \mathbb{R}^d$ and $i \in [n]$. The noises $\xi_i$ are i.i.d. samples from $\mathcal{N}(0, \sigma)$, where $\sigma$ is a parameter.

### C.3 Experiment 4: training neural network

We test algorithms on an image recognition task, *CIFAR10* [Krizhevsky et al., 2009], with the *ResNet-18* [He et al., 2016] deep neural network (the number of parameters $d \approx 10^7$). We split *CIFAR10* among 5 nodes, and take $K = 2 \times 10^6$ in Top$K$. In all methods we finetune the step sizes. One can see that our findings in the low-scale experiments translate into large-scale experiments in Figure 8. With different batch sizes, EF21-SGD converges slower than EF21-SGDM and EF14-SGD, and our new method EF21-SGDM improves over EF14-SGD in Figure 8b. We checked the accuracies on the test dataset (see Table 9) and observed the same relations between algorithms (note that accuracies are far from the real SOTA because we turned off all augmentations and regularizations in training).

## D  Descent Lemma

Let us state the following lemma that is used in the analysis of nonconvex optimization methods.

**Lemma 1** ([Li et al., 2021])**.** *Let the function $f(\cdot)$ be $L$-smooth and let $x^{t+1} = x^t - \gamma g^t$ for some vector $g^t \in \mathbb{R}^d$ and a step-size $\gamma > 0$. Then we have*

$$f(x^{t+1}) \leq f(x^t) - \frac{\gamma}{2} \left\| \nabla f(x^t) \right\|^2 - \left( \frac{1}{2\gamma} - \frac{L}{2} \right) \left\| x^{t+1} - x^t \right\|^2 + \frac{\gamma}{2} \left\| g^t - \nabla f(x^t) \right\|^2. \quad (13)$$

## E  EF21-SGDM-ideal (Proof of Theorem 1 and Proposition 1)

We now state a slightly more general result than Theorem 1, which holds for EF21-SGDM-ideal method with any $\eta \in (0, 1]$. The statement of Theorem 1 follows by setting $\eta = 1$, since in that case EF21-SGDM-ideal coincides with EF21-SGD-ideal (5a), (5aa). Recall that EF21-SGDM-ideal (distributed variant) has the following update rule:

$$x^{t+1} = x^t - \gamma g^t, \qquad g^t = \frac{1}{n} \sum_{i=1}^{n} g_i^t, \quad (14)$$

EF21-SGDM-ideal:
$$\begin{aligned} v_i^{t+1} &= \nabla f_i(x^{t+1}) + \eta(\nabla f_i(x^{t+1}, \xi_i^{t+1}) - \nabla f_i(x^{t+1})), \\ g_i^{t+1} &= \nabla f_i(x^{t+1}) + \mathcal{C}\left( v_i^{t+1} - \nabla f_i(x^{t+1}) \right). \end{aligned} \quad (15)$$

**Theorem 4.** *Let $L, \sigma > 0$, $0 < \gamma \leq 1/L$, $0 < \eta \leq 1$ and $n = 1$. There exists a convex, $L$-smooth function $f(\cdot)$, a contractive compressor $\mathcal{C}(\cdot)$ satisfying Definition 1, and an unbiased stochastic gradient with bounded variance $\sigma^2$ such that if the method (14), (15) is run with a step-size $\gamma$, then for all $T \geq 0$ and for all $x^0 \in \{(0, x_{(2)}^0)^\top \in \mathbb{R}^2 \mid x_{(2)}^0 < 0\}$, we have*

$$\mathbb{E}\left[ \left\| \nabla f(x^T) \right\|^2 \right] \geq \frac{1}{60} \min \left\{ \eta^2 \sigma^2, \left\| \nabla f(x^0) \right\|^2 \right\}.$$

*Fix $0 < \varepsilon \leq L/\sqrt{60}$ and $x^0 = (0, -1)^\top$. Additionally assume that $n \geq 1$ and the variance of unbiased stochastic gradient is controlled by $\sigma^2/B$ for some $B \geq 1$. If $B < \frac{\eta^2 \sigma^2}{60 \varepsilon^2}$, then we have $\mathbb{E}\left[ \left\| \nabla f(x^T) \right\| \right] > \varepsilon$ for all $T \geq 0$.*

***Proof of Theorem 1.*** **Part I.** Consider $f(x) = \frac{L}{2} \left\| x \right\|^2$, $x \in \mathbb{R}^2$. For each iteration $t \geq 0$, let the random vector $\xi^{t+1}$ be sampled uniformly at random from the set of vectors:

$$z_1 = \binom{2}{0} \sqrt{\frac{3\sigma^2}{10}}, \quad z_2 = \binom{0}{1} \sqrt{\frac{3\sigma^2}{10}}, \quad z_3 = \binom{-2}{-1} \sqrt{\frac{3\sigma^2}{10}}.$$

Define the stochastic gradient as $\nabla f(x^t, \xi^t) := \nabla f(x^t) + \xi^t = Lx^t + \xi^t$. Notice that $\mathbb{E}\left[ \nabla f(x^t, \xi^t) \right] = \nabla f(x^t)$, and $\mathbb{E}\left[ \left\| \nabla f(x^t, \xi^t) - \nabla f(x^t) \right\|^2 \right] = \sigma^2$. The update rule of method (14), (15) with such estimator is

$$x^{t+1} = x^t - \gamma g^t = x^t - L\gamma x^t - \gamma \mathcal{C}\left( \eta \, \xi^t \right),$$

where we choose $\mathcal{C}(\cdot)$ as a Top1 compressor. Notice that $\mathbb{E}\left[ \xi^t \right] = (0, 0)^\top$, but

$$\mathbb{E}\left[ \mathcal{C}(\xi^t) \right] = \eta \sqrt{\frac{3\sigma^2}{10}} (0, 1/3)^\top \neq (0, 0)^\top.$$

By setting the initial iterate to $x^0 = (0, x_{(2)}^0)^\top$ for any $x_{(2)}^0 < 0$, we can derive

$$\begin{aligned} \mathbb{E}\left[ x^T \right] &= (1 - L\gamma)^T x^0 - \eta \sqrt{\frac{3\sigma^2}{10}} \binom{0}{\frac{1}{3}} \gamma \sum_{t=0}^{T-1} (1 - L\gamma)^t \\ &= (1 - L\gamma)^T \binom{0}{x_{(2)}^0} + \frac{\eta}{L} \sqrt{\frac{\sigma^2}{30}} \binom{0}{-1} (1 - (1 - L\gamma)^T) \neq \binom{0}{0} \quad (16) \end{aligned}$$

for any $0 \leq \gamma \leq 1/L$ and any $x_{(2)}^0 < 0$. The inequality in (16) is because the first vector has strictly negative component $x_{(2)}^0$, and the second vector has non-positive second component when $\gamma > 0$ and $\sigma^2 > 0$. Therefore, since $\|\nabla f(x)\|^2 = \|Lx\|^2$, we have

$$
\begin{aligned}
\mathbb{E}\left[\left\|\nabla f(x^T)\right\|^2\right] &= \mathbb{E}\left[\left\|Lx^T\right\|^2\right] \\
&= \left\|\mathbb{E}\left[Lx^T\right]\right\|^2 + \mathbb{E}\left[\left\|Lx^T - \mathbb{E}\left[Lx^T\right]\right\|^2\right] \\
&\geq \left\|\mathbb{E}\left[Lx^T\right]\right\|^2 \\
&\overset{(i)}{=} \left((1-L\gamma)^T\left\|Lx^0\right\| + \eta\sqrt{\frac{\sigma^2}{30}}(1-(1-L\gamma)^T)\right)^2 \\
&\overset{(ii)}{\geq} (1-L\gamma)^{2T}\left\|\nabla f(x^0)\right\|^2 + \frac{\eta^2\sigma^2}{30}(1-(1-L\gamma)^T)^2 \\
&\geq \frac{\left\|\nabla f(x^0)\right\|^2 \eta^2\sigma^2}{30\left\|\nabla f(x^0)\right\|^2 + \eta^2\sigma^2}
\end{aligned}
$$

for all $T \geq 1$, where in $(i)$ we used the form of vector $\mathbb{E}\left[x^T\right]$ in (16), in $(ii)$ we drop a non-negative cross term, and use $\nabla f(x^0) = Lx^0$. The last inequality follows by lower bounding a univariate quadratic function with respect to $z := (1-L\gamma)^T$ for $0 \leq z \leq 1$, where optimal choice is $z = \eta^2\sigma^2/(30\left\|\nabla f(x^0)\right\|^2 + \eta^2\sigma^2)$. It is left to note that $\frac{xy}{x+y} \geq \frac{1}{2}\min\{x,y\}$ for all $x, y > 0$.

**Part II.** Fix $n \geq 1$ and $B \geq 1$. Let at each node $i = 1, \ldots, n$, the random vectors $\xi_i^t$ be sampled independently and uniformly form the set of vectors:

$$
z_1 = \begin{pmatrix} 2 \\ 0 \end{pmatrix}\sqrt{\frac{3\sigma^2}{10B}}, \quad z_2 = \begin{pmatrix} 0 \\ 1 \end{pmatrix}\sqrt{\frac{3\sigma^2}{10B}}, \quad z_3 = \begin{pmatrix} -2 \\ -1 \end{pmatrix}\sqrt{\frac{3\sigma^2}{10B}}.
$$

Define a random matrix $\xi^t := (\xi_1^t, \ldots, \xi_n^t)^\top$. Then $\mathbb{E}\left[\left\|\nabla f(x^t, \xi^t) - \nabla f(x^t)\right\|^2\right] = \frac{\sigma^2}{B}$. The update of the method (14), (15) on the same function instance will take the form

$$
x^{t+1} = x^t - \gamma\frac{1}{n}\sum_{i=1}^n g_i^t = x^t - L\gamma x^t - \gamma\frac{1}{n}\sum_{i=1}^n \mathcal{C}\left(\eta\,\xi_i^t\right).
$$

Notice that in this case, we still have

$$
\mathbb{E}\left[\frac{1}{n}\sum_{i=1}^n \mathcal{C}(\eta\,\xi_i^t)\right] = \frac{1}{n}\sum_{i=1}^n \mathbb{E}\left[\mathcal{C}(\eta\,\xi_i^t)\right] = \eta\sqrt{\frac{3\sigma^2}{10}}(0, 1/3)^\top \neq (0,0)^\top,
$$

which is independent (!) of $n$. Therefore, by similar derivations, we can conclude that

$$
\mathbb{E}\left[\left\|\nabla f(x^T)\right\|^2\right] \geq \frac{1}{60}\min\left\{\frac{\eta^2\sigma^2}{B}, \left\|\nabla f(x^0)\right\|^2\right\} > \varepsilon^2,
$$

where we use that $B < \frac{\eta^2\sigma^2}{60\varepsilon^2}$, $\varepsilon \leq L/\sqrt{60}$, and $x^0 = (0, -1)^\top$.

***Proof of Proposition 1.*** By smoothness (Assumption 1) of $f(\cdot)$ it follows from Lemma 1 that for $\gamma \leq 1/L$ we have

$$
f(x^{t+1}) \leq f(x^t) - \frac{\gamma}{2}\left\|\nabla f(x^t)\right\|^2 + \frac{\gamma}{2}\left\|g^t - \nabla f(x^t)\right\|^2. \tag{17}
$$

Now it remains to control the last term, which is due to the error introduced by a contractive compressor and stochastic gradients. We have

$$
\begin{aligned}
\mathbb{E}\left[\left\|g^t - \nabla f(x^t)\right\|^2\right] &\overset{(i)}{=} \mathbb{E}\left[\left\|\mathcal{C}\left(v^t - \nabla f(x^t)\right)\right\|^2\right] \overset{(ii)}{=} \mathbb{E}\left[\left\|\mathcal{C}\left(\eta\left(\nabla f(x^t, \xi^t) - \nabla f(x^t)\right)\right)\right\|^2\right] \\
&\overset{(iii)}{\leq} 2\mathbb{E}\left[\left\|\mathcal{C}\left(\eta\left(\nabla f(x^t, \xi^t) - \nabla f(x^t)\right)\right) - \eta(\nabla f(x^t, \xi^t) - \nabla f(x^t))\right\|^2\right]
\end{aligned}
$$

$$+2\eta^2\mathbb{E}\left[\left\|\nabla f(x^t,\xi^t)-\nabla f(x^t)\right\|^2\right]$$

$$\leq \quad 2(2-\alpha)\eta^2\mathbb{E}\left[\left\|\nabla f(x^t,\xi^t)-\nabla f(x^t)\right\|^2\right]$$

$$\leq \quad 4\eta^2\sigma^2,$$

where $(i)$ and $(ii)$ use the update rule (6), $(iii)$ holds by Young's inequality, and the last two steps hold by Definition 1 and Assumption 2.

Subtracting $f^*$ from both sides of (17), taking expectation and defining $\delta_t := \mathbb{E}\left[f(x^t)-f^*\right]$, we derive

$$\mathbb{E}\left[\left\|\nabla f(\hat{x}^T)\right\|^2\right]=\frac{1}{T}\sum_{t=0}^{T-1}\mathbb{E}\left[\left\|\nabla f(x^t)\right\|^2\right]\leq\frac{2\delta_0}{\gamma T}+4\eta^2\sigma^2.$$

# F EF21-SGDM (Proof of Theorems 2 and 3)

The statement of Theorem 2 follows directly from Theorem 3 and Remark 2. Let us prove Theorem 3.

***Proof of Theorem 3.*** In order to control the error between $g^t$ and $\nabla f(x^t)$, we decompose it into two terms

$$\left\| g^t - \nabla f(x^t) \right\|^2 \leq 2 \left\| g^t - v^t \right\|^2 + 2 \left\| v^t - \nabla f(x^t) \right\|^2 \leq 2 \frac{1}{n} \sum_{i=1}^{n} \left\| g_i^t - v_i^t \right\|^2 + 2 \left\| v^t - \nabla f(x^t) \right\|^2,$$

and develop a recursion for each term above separately.

**Part I (a). Controlling the error of momentum estimator for each $v_i^t$.** Recall that by Lemma 2-(24), we have for each $i = 1, \ldots, n$, and any $0 < \eta \leq 1$ and $t \geq 0$

$$\mathbb{E}\left[ \left\| v_i^{t+1} - \nabla f_i(x^{t+1}) \right\|^2 \right] \leq (1-\eta)\mathbb{E}\left[ \left\| v_i^t - \nabla f_i(x^t) \right\|^2 \right] + \frac{3L_i^2}{\eta}\mathbb{E}\left[ \left\| x^{t+1} - x^t \right\|^2 \right] + \eta^2\sigma^2. \quad (18)$$

Averaging inequalities (18) over $i = 1, \ldots, n$ and denoting $\widetilde{P}_t := \frac{1}{n} \sum_{i=1}^{n} \mathbb{E}\left[ \left\| v_i^t - \nabla f_i(x^t) \right\|^2 \right]$, $R_t := \mathbb{E}\left[ \left\| x^t - x^{t+1} \right\|^2 \right]$ we have

$$\widetilde{P}_{t+1} \leq (1-\eta)\widetilde{P}_t + \frac{3\widetilde{L}^2}{\eta}R_t + \eta^2\sigma^2.$$

Summing up the above inequality for $t = 0, \ldots, T-1$, we derive

$$\frac{1}{T}\sum_{t=0}^{T-1}\widetilde{P}_t \leq \frac{3\widetilde{L}^2}{\eta^2}\frac{1}{T}\sum_{t=0}^{T-1}R_t + \eta\sigma^2 + \frac{1}{\eta T}\widetilde{P}_0. \quad (19)$$

**Part I (b). Controlling the error of momentum estimator for $v^t$ (on average).** Similarly by Lemma 2-(25), we have for any $0 < \eta \leq 1$ and $t \geq 0$

$$\mathbb{E}\left[ \left\| v^{t+1} - \nabla f(x^{t+1}) \right\|^2 \right] \leq (1-\eta)\mathbb{E}\left[ \left\| v^t - \nabla f(x^t) \right\|^2 \right] + \frac{3L^2}{\eta}\mathbb{E}\left[ \left\| x^{t+1} - x^t \right\|^2 \right] + \frac{\eta^2\sigma^2}{n},$$

where $v^t := \frac{1}{n} \sum_{i=1}^{n} v_i^t$ is an auxiliary sequence.

Summing up the above inequality for $t = 0, \ldots, T-1$, and denoting $P_t := \mathbb{E}\left[ \left\| v^t - \nabla f(x^t) \right\|^2 \right]$, we derive

$$\frac{1}{T}\sum_{t=0}^{T-1}P_t \leq \frac{3L^2}{\eta^2}\frac{1}{T}\sum_{t=0}^{T-1}R_t + \frac{\eta\sigma^2}{n} + \frac{1}{\eta T}P_0. \quad (20)$$

**Part II. Controlling the error of contractive compressor and momentum estimator.** By Lemma 3 we have for each $i = 1, \ldots, n$, and any $0 < \eta \leq 1$ and $t \geq 0$

$$
\begin{aligned}
\mathbb{E}\left[ \left\| g_i^{t+1} - v_i^{t+1} \right\|^2 \right] &\leq \left(1 - \frac{\alpha}{2}\right)\mathbb{E}\left[ \left\| g_i^t - v_i^t \right\|^2 \right] + \frac{4\eta^2}{\alpha}\mathbb{E}\left[ \left\| v_i^t - \nabla f_i(x^t) \right\|^2 \right] \\
&\quad + \frac{4L_i^2\eta^2}{\alpha}\mathbb{E}\left[ \left\| x^{t+1} - x^t \right\|^2 \right] + \eta^2\sigma^2.
\end{aligned} \quad (21)
$$

Averaging inequalities (21) over $i = 1, \ldots, n$, denoting $\widetilde{V}_t := \frac{1}{n} \sum_{i=1}^{n} \mathbb{E}\left[ \left\| g_i^t - v_i^t \right\|^2 \right]$, and summing up the resulting inequality for $t = 0, \ldots, T-1$, we obtain

$$\frac{1}{T}\sum_{t=0}^{T-1}\widetilde{V}_t \leq \frac{8\eta^2}{\alpha^2}\frac{1}{T}\sum_{t=0}^{T-1}\widetilde{P}_t + \frac{8\widetilde{L}^2\eta^2}{\alpha^2}\frac{1}{T}\sum_{t=0}^{T-1}R_t + \frac{2\eta^2\sigma^2}{\alpha} + \frac{2}{\alpha T}\widetilde{V}_0. \quad (22)$$

**Part III. Combining steps I and II with descent lemma.** By smoothness (Assumption 1) of $f(\cdot)$ it follows from Lemma 1 that for any $\gamma \leq 1/(2L)$ we have

$$
\begin{aligned}
f(x^{t+1}) \;\leq\;& f(x^t) - \frac{\gamma}{2}\left\|\nabla f(x^t)\right\|^2 - \frac{1}{4\gamma}\left\|x^{t+1} - x^t\right\|^2 + \frac{\gamma}{2}\left\|g^t - \nabla f(x^t)\right\|^2 \qquad (23)\\[2mm]
\leq\;& f(x^t) - \frac{\gamma}{2}\left\|\nabla f(x^t)\right\|^2 - \frac{1}{4\gamma}\left\|x^{t+1} - x^t\right\|^2 + \gamma\frac{1}{n}\sum_{i=1}^{n}\left\|g_i^t - v_i^t\right\|^2 + \gamma\left\|v^t - \nabla f(x^t)\right\|^2.
\end{aligned}
$$

Subtracting $f^*$ from both sides of (23), taking expectation and defining $\delta_t := \mathbb{E}\left[f(x^t) - f^*\right]$, we derive

$$
\begin{aligned}
\mathbb{E}\left[\left\|\nabla f(\hat{x}^T)\right\|^2\right] \;=\;& \frac{1}{T}\sum_{t=0}^{T-1}\mathbb{E}\left[\left\|\nabla f(x^t)\right\|^2\right]\\[2mm]
\leq\;& \frac{2\delta_0}{\gamma T} + 2\frac{1}{T}\sum_{t=0}^{T-1}\widetilde{V}_t + 2\frac{1}{T}\sum_{t=0}^{T-1}P_t - \frac{1}{2\gamma^2}\frac{1}{T}\sum_{t=0}^{T-1}R_t\\[2mm]
\overset{(i)}{\leq}\;& \frac{2\delta_0}{\gamma T} + \frac{16\eta^2}{\alpha^2}\frac{1}{T}\sum_{t=0}^{T-1}\widetilde{P}_t + 2\frac{1}{T}\sum_{t=0}^{T-1}P_t + \frac{4\eta^2\sigma^2}{\alpha}\\[2mm]
& - \frac{\frac{1}{2} - \frac{16\gamma^2\widetilde{L}^2\eta^2}{\alpha^2}}{\gamma^2}\frac{1}{T}\sum_{t=0}^{T-1}R_t\\[2mm]
\overset{(ii)}{\leq}\;& \frac{2\delta_0}{\gamma T} + \frac{16\eta^3\sigma^2}{\alpha^2} + \frac{4\eta^2\sigma^2}{\alpha} + \frac{2\eta\sigma^2}{n} + \frac{4}{\alpha T}\widetilde{V}_0\\[2mm]
& - \frac{\frac{1}{2} - \frac{16\gamma^2\widetilde{L}^2\eta^2}{\alpha^2} - \frac{6\gamma^2L^2}{\eta^2} - \frac{48\gamma^2\widetilde{L}^2}{\alpha^2}}{\gamma^2}\frac{1}{T}\sum_{t=0}^{T-1}R_t + \frac{2}{\eta T}P_0 + \frac{16\eta}{\alpha^2 T}\widetilde{P}_0\\[2mm]
\overset{(iii)}{\leq}\;& \frac{2\delta_0}{\gamma T} + \frac{16\eta^3\sigma^2}{\alpha^2} + \frac{4\eta^2\sigma^2}{\alpha} + \frac{2\eta\sigma^2}{n} - \frac{\frac{1}{2} - \frac{6\gamma^2L^2}{\eta^2} - \frac{64\gamma^2\widetilde{L}^2}{\alpha^2}}{\gamma^2}\frac{1}{T}\sum_{t=0}^{T-1}R_t\\[2mm]
& + \frac{2}{\eta T}P_0 + \frac{16\eta}{\alpha^2 T}\widetilde{P}_0 + \frac{4}{\alpha T}\widetilde{V}_0\\[2mm]
\leq\;& \frac{2\delta_0}{\gamma T} + \frac{16\eta^3\sigma^2}{\alpha^2} + \frac{4\eta^2\sigma^2}{\alpha} + \frac{2\eta\sigma^2}{n} + \frac{2}{\eta T}P_0 + \frac{16\eta}{\alpha^2 T}\widetilde{P}_0 + \frac{4}{\alpha T}\widetilde{V}_0.
\end{aligned}
$$

where $(i)$ holds due to (22), $(ii)$ utilizes (20), and $(iii)$ follows by $\eta \leq 1$, and the last step holds due to the assumption on the step-size. We proved (9).

We now find the particular values of parameters. Since $g_i = v_i$ for all $i \in [n]$, we have $\widetilde{V}_0 = 0$. Using $v_i^0 = \frac{1}{B_{\text{init}}}\sum_{j=1}^{B_{\text{init}}}\nabla f_i(x^0, \xi_{i,j}^0)$ for all $i = 1, \ldots, n$, we have

$$
P_0 = \mathbb{E}\left[\left\|v^0 - \nabla f(x^0)\right\|^2\right] \leq \frac{\sigma^2}{nB_{\text{init}}} \quad \text{and} \quad \widetilde{P}_0 = \frac{1}{n}\sum_{i=1}^{n}\mathbb{E}\left[\left\|v_i^0 - \nabla f_i(x^0)\right\|^2\right] \leq \frac{\sigma^2}{B_{\text{init}}}.
$$

We can substitute the choice of $\gamma$ and obtain

$$
\mathbb{E}\left[\left\|\nabla f(\hat{x}^T)\right\|^2\right] \;=\; \mathcal{O}\left(\frac{\widetilde{L}\delta_0}{\alpha T} + \frac{L\delta_0}{\eta T} + \frac{\eta^3\sigma^2}{\alpha^2} + \frac{\eta^2\sigma^2}{\alpha} + \frac{\eta\sigma^2}{n} + \frac{\sigma^2}{\eta n B_{\text{init}} T} + \frac{\eta\sigma^2}{\alpha^2 B_{\text{init}} T}\right).
$$

Since $B_{\text{init}} \geq \frac{\sigma^2}{L\delta_0 n}$, we have

$$
\mathbb{E}\left[\left\|\nabla f(\hat{x}^T)\right\|^2\right] \;=\; \mathcal{O}\left(\frac{\widetilde{L}\delta_0}{\alpha T} + \frac{L\delta_0}{\eta T} + \frac{\eta^3\sigma^2}{\alpha^2} + \frac{\eta^2\sigma^2}{\alpha} + \frac{\eta\sigma^2}{n} + \frac{\eta\sigma^2}{\alpha^2 B_{\text{init}} T}\right).
$$

Notice that the choice of the momentum parameter such that $\eta \leq \left(\frac{L\delta_0\alpha^2}{\sigma^2 T}\right)^{1/4}$, $\eta \leq \left(\frac{L\delta_0\alpha}{\sigma^2 T}\right)^{1/3}$, $\eta \leq \left(\frac{L\delta_0 n}{\sigma^2 T}\right)^{1/2}$ and $\eta \leq \frac{\alpha\sqrt{L\delta_0 B_{\text{init}}}}{\sigma}$ ensures that $\frac{\eta^3\sigma^2}{\alpha^2} \leq \frac{L\delta_0}{\eta T}$, $\frac{\eta^2\sigma^2}{\alpha} \leq \frac{L\delta_0}{\eta T}$, $\frac{\eta\sigma^2}{n} \leq \frac{L\delta_0}{\eta T}$, and

$\frac{\eta\sigma^2}{\alpha^2 B_{\text{init}} T} \leq \frac{L\delta_0}{\eta T}$. Therefore, we have

$$\mathbb{E}\left[\left\|\nabla f(\hat{x}^T)\right\|^2\right] \leq \mathcal{O}\left(\frac{\widetilde{L}\delta_0}{\alpha T} + \left(\frac{L\delta_0\sigma^{2/3}}{\alpha^{2/3}T}\right)^{3/4} + \left(\frac{L\delta_0\sigma}{\sqrt{\alpha}T}\right)^{2/3} + \left(\frac{L\delta_0\sigma^2}{nT}\right)^{1/2} + \frac{\sigma\sqrt{L\delta_0}}{\alpha\sqrt{B_{\text{init}}}T}\right).$$

Using $B_{\text{init}} \geq \min\left\{\frac{\sigma^2 L}{\widetilde{L}^2\delta_0}, \frac{\sigma}{\alpha\sqrt{L\delta_0 T}}, \frac{\sigma^{2/3}}{\alpha^{4/3}T^{2/3}(L\delta_0)^{1/3}}, \frac{n}{\alpha^2 T}\right\}$, we obtain

$$\mathbb{E}\left[\left\|\nabla f(\hat{x}^T)\right\|^2\right] \leq \mathcal{O}\left(\frac{\widetilde{L}\delta_0}{\alpha T} + \left(\frac{L\delta_0\sigma^{2/3}}{\alpha^{2/3}T}\right)^{3/4} + \left(\frac{L\delta_0\sigma}{\sqrt{\alpha}T}\right)^{2/3} + \left(\frac{L\delta_0\sigma^2}{nT}\right)^{1/2}\right).$$

It remains to notice that $\left\lceil \max\left\{\min\left\{\frac{\sigma^2 L}{\widetilde{L}^2\delta_0}, \frac{\sigma}{\alpha\sqrt{L\delta_0 T}}, \frac{\sigma^{2/3}}{\alpha^{4/3}T^{2/3}(L\delta_0)^{1/3}}, \frac{n}{\alpha^2 T}\right\}, \frac{\sigma^2}{L\delta_0 n}\right\}\right\rceil \leq \left\lceil\frac{\sigma^2}{L\delta_0}\right\rceil$.

### F.1 Controlling the error of momentum estimator

**Lemma 2.** *Let Assumption 1 be satisfied, and suppose $0 < \eta \leq 1$. For every $i = 1, \ldots, n$, let the sequence $\{v_i^t\}_{t\geq 0}$ be updated via*

$$v_i^t = v_i^{t-1} + \eta\left(\nabla f_i(x^t, \xi_i^t) - v_i^{t-1}\right),$$

*Define the sequence $v^t := \frac{1}{n}\sum_{i=1}^n v_i^t$. Then for every $i = 1, \ldots, n$ and $t \geq 0$ it holds*

$$\mathbb{E}\left[\left\|v_i^t - \nabla f_i(x^t)\right\|^2\right] \leq (1-\eta)\mathbb{E}\left[\left\|v_i^{t-1} - \nabla f_i(x^{t-1})\right\|^2\right] + \frac{3L_i^2}{\eta}\mathbb{E}\left[\left\|x^t - x^{t-1}\right\|^2\right] + \eta^2\sigma^2, \quad (24)$$

$$\mathbb{E}\left[\left\|v^t - \nabla f(x^t)\right\|^2\right] \leq (1-\eta)\mathbb{E}\left[\left\|v^{t-1} - \nabla f(x^{t-1})\right\|^2\right] + \frac{3L^2}{\eta}\mathbb{E}\left[\left\|x^t - x^{t-1}\right\|^2\right] + \frac{\eta^2\sigma^2}{n}. \quad (25)$$

*Proof.* By the update rule of $v_i^t$, we have

$$\begin{aligned}
\mathbb{E}\left[\left\|v_i^t - \nabla f_i(x^t)\right\|^2\right] &= \mathbb{E}\left[\left\|v_i^{t-1} - \nabla f_i(x^t) + \eta(\nabla f_i(x^t, \xi_i^t) - v_i^{t-1})\right\|^2\right] \\
&= \mathbb{E}\left[\mathbb{E}_{\xi_i^t}\left[\left\|(1-\eta)(v_i^{t-1} - \nabla f_i(x^t)) + \eta(\nabla f_i(x^t, \xi_i^t) - \nabla f_i(x^t))\right\|^2\right]\right] \\
&= (1-\eta)^2\mathbb{E}\left[\left\|v_i^{t-1} - \nabla f_i(x^t)\right\|^2\right] + \eta^2\mathbb{E}\left[\left\|\nabla f_i(x^t, \xi_i^t) - \nabla f_i(x^t)\right\|^2\right] \\
&\leq (1-\eta)^2\left(1 + \frac{\eta}{2}\right)\mathbb{E}\left[\left\|v_i^{t-1} - \nabla f_i(x^{t-1})\right\|^2\right] \\
&\quad + \left(1 + \frac{2}{\eta}\right)\mathbb{E}\left[\left\|\nabla f_i(x^{t-1}) - \nabla f_i(x^t)\right\|^2\right] + \eta^2\sigma^2 \\
&\leq (1-\eta)\mathbb{E}\left[\left\|v_i^{t-1} - \nabla f_i(x^{t-1})\right\|^2\right] + \frac{3L_i^2}{\eta}\mathbb{E}\left[\left\|x^t - x^{t+1}\right\|^2\right] + \eta^2\sigma^2,
\end{aligned}$$

where the first inequality holds by Young's inequality, and the last step uses smoothness of $f_i(\cdot)$ (Assumption 1), which concludes the proof of (24).

For each $t = 0, \ldots, T-1$, define a random vector $\xi^t := (\xi_1^t, \ldots, \xi_n^t)$ and denote by $\nabla f(x^t, \xi^t) := \frac{1}{n}\sum_{i=1}^n \nabla f_i(x^t, \xi_i^t)$. Note that the entries of the random vector $\xi^t$ are independent and $\mathbb{E}_{\xi^t}\left[\nabla f(x^t, \xi^t)\right] = \nabla f(x^t)$, then we have

$$v^t = v^{t-1} + \eta\left(\nabla f(x^t, \xi^t) - v^{t-1}\right),$$

where $v^t := \frac{1}{n}\sum_{i=1}^n v_i^t$ is an auxiliary sequence. Therefore, we can similarly derive

$$\begin{aligned}
\mathbb{E}\left[\left\|v^t - \nabla f(x^t)\right\|^2\right] &= \mathbb{E}\left[\left\|v^{t-1} - \nabla f(x^t) + \eta\left(\nabla f(x^t, \xi^t) - v^{t-1}\right)\right\|^2\right] \\
&= \mathbb{E}\left[\mathbb{E}_{\xi^t}\left[\left\|(1-\eta)(v^{t-1} - \nabla f(x^t)) + \eta\left(\nabla f(x^t, \xi^t) - \nabla f(x^t)\right)\right\|^2\right]\right]
\end{aligned}$$

$$
\begin{aligned}
&= \quad (1-\eta)^2 \mathbb{E}\left[\left\|v^{t-1} - \nabla f(x^t)\right\|^2\right] + \eta^2 \mathbb{E}\left[\left\|\nabla f(x^t, \xi^t) - \nabla f(x^t)\right\|^2\right] \\
&\leq \quad (1-\eta)^2 \left(1 + \frac{\eta}{2}\right) \mathbb{E}\left[\left\|v^{t-1} - \nabla f(x^{t-1})\right\|^2\right] \\
&\qquad + \left(1 + \frac{2}{\eta}\right) \mathbb{E}\left[\left\|\nabla f(x^{t-1}) - \nabla f(x^t)\right\|^2\right] + \frac{\eta^2 \sigma^2}{n} \\
&\leq \quad (1-\eta)\mathbb{E}\left[\left\|v^{t-1} - \nabla f(x^{t-1})\right\|^2\right] + \frac{3L^2}{\eta}\mathbb{E}\left[\left\|x^t - x^{t-1}\right\|^2\right] + \frac{\eta^2 \sigma^2}{n},
\end{aligned}
$$

where the last step uses smoothness of $f(\cdot)$ (Assumption 1), which concludes the proof of (25). $\quad\square$

## F.2 Controlling the error of contractive compression and momentum estimator

**Lemma 3.** *Let Assumption 1 be satisfied, and suppose $\mathcal{C}$ is a contractive compressor with $\alpha \leq \frac{1}{2}$.*
*For every $i = 1, \ldots, n$, let the sequences $\{v_i^t\}_{t \geq 0}$ and $\{g_i^t\}_{t \geq 0}$ be updated via*

$$
v_i^t = v_i^{t-1} + \eta\left(\nabla f_i(x^t, \xi_i^t) - v_i^{t-1}\right),
$$

$$
g_i^t = g_i^{t-1} + \mathcal{C}\left(v_i^t - g_i^{t-1}\right),
$$

*Then for every $i = 1, \ldots, n$ and $t \geq 0$ it holds*

$$
\begin{aligned}
\mathbb{E}\left[\left\|g_i^t - v_i^t\right\|^2\right] &\leq \quad \left(1 - \frac{\alpha}{2}\right)\mathbb{E}\left[\left\|g_i^{t-1} - v_i^{t-1}\right\|^2\right] + \frac{4\eta^2}{\alpha}\mathbb{E}\left[\left\|v_i^{t-1} - \nabla f_i(x^{t-1})\right\|^2\right] \\
&\qquad + \frac{4L_i^2 \eta^2}{\alpha}\mathbb{E}\left[\left\|x^t - x^{t-1}\right\|^2\right] + \eta^2 \sigma^2. \qquad (26)
\end{aligned}
$$

*Proof.* By the update rules of $g_i^t$ and $v_i^t$, we derive

$$
\begin{aligned}
\mathbb{E}\left[\left\|g_i^t - v_i^t\right\|^2\right] &= \quad \mathbb{E}\left[\left\|g_i^{t-1} - v_i^t + \mathcal{C}(v_i^t - g_i^{t-1})\right\|^2\right] \\
&= \quad \mathbb{E}\left[\mathbb{E}_{\mathcal{C}}\left[\left\|\mathcal{C}(v_i^t - g_i^{t-1}) - (v_i^t - g_i^{t-1})\right\|^2\right]\right] \\
&\overset{(i)}{\leq} \quad (1-\alpha)\mathbb{E}\left[\left\|v_i^t - g_i^{t-1}\right\|^2\right] \\
&\overset{(ii)}{=} \quad (1-\alpha)\mathbb{E}\left[\left\|v_i^{t-1} - g_i^{t-1} + \eta(\nabla f_i(x^t, \xi_i^t) - v_i^{t-1})\right\|^2\right] \\
&= \quad (1-\alpha)\mathbb{E}\left[\mathbb{E}_{\xi_i^t}\left[\left\|v_i^{t-1} - g_i^{t-1} + \eta(\nabla f_i(x^t, \xi_i^t) - v_i^{t-1})\right\|^2\right]\right] \\
&= \quad (1-\alpha)\mathbb{E}\left[\left\|v_i^{t-1} - g_i^{t-1} + \eta(\nabla f_i(x^t) - v_i^{t-1})\right\|^2\right] \\
&\qquad + (1-\alpha)\eta^2\mathbb{E}\left[\left\|\nabla f_i(x^t, \xi_i^t) - \nabla f_i(x^t)\right\|^2\right] \\
&\overset{(iii)}{\leq} \quad (1-\alpha)(1+\rho)\mathbb{E}\left[\left\|v_i^{t-1} - g_i^{t-1}\right\|^2\right] + (1-\alpha)(1+\rho^{-1})\eta^2\mathbb{E}\left[\left\|v_i^{t-1} - \nabla f_i(x^t)\right\|^2\right] \\
&\qquad + (1-\alpha)\eta^2\sigma^2 \\
&\overset{(iv)}{=} \quad (1-\theta)\mathbb{E}\left[\left\|g_i^{t-1} - v_i^{t-1}\right\|^2\right] + \beta\eta^2\mathbb{E}\left[\left\|v_i^{t-1} - \nabla f_i(x^t)\right\|^2\right] + (1-\alpha)\eta^2\sigma^2 \\
&\overset{(v)}{\leq} \quad (1-\theta)\mathbb{E}\left[\left\|g_i^{t-1} - v_i^{t-1}\right\|^2\right] + 2\beta\eta^2\mathbb{E}\left[\left\|v_i^{t-1} - \nabla f_i(x^{t-1})\right\|^2\right] \\
&\qquad + 2\beta\eta^2\mathbb{E}\left[\left\|\nabla f_i(x^t) - \nabla f_i(x^{t-1})\right\|^2\right] + \eta^2\sigma^2 \\
&\leq \quad (1-\theta)\mathbb{E}\left[\left\|g_i^{t-1} - v_i^{t-1}\right\|^2\right] + 2\beta\eta^2\mathbb{E}\left[\left\|v_i^{t-1} - \nabla f_i(x^{t-1})\right\|^2\right] \\
&\qquad + 2\beta L_i^2\eta^2\mathbb{E}\left[\left\|x^t - x^{t-1}\right\|^2\right] + \eta^2\sigma^2,
\end{aligned}
$$

where $(i)$ is due to the definition of a contractive compressor (Definition 1), $(ii)$ follows by the update rule of $v_i^t$, $(iii)$ and $(v)$ hold by Young's inequality for any $\rho > 0$. In $(iv)$, we introduced the notation

$\theta := 1 - (1 - \alpha)(1 + \rho)$, and $\beta := (1 - \alpha)(1 + \rho^{-1})$. The last step follows by smoothness of $f_i(\cdot)$ (Assumption 1). The proof is complete by the choice $\rho = \alpha/2$, which guarantees $1 - \theta \le 1 - \alpha/2$, and $2\beta \le 4/\alpha$. $\qquad\square$

# G Further Improvement Using Double Momentum (Proof of Corollary 3)

---

**Algorithm 3** EF21-SGD2M (Error Feedback 2021 Enhanced with *Double Momentum*)

---

1: **Input:** starting point $x^0$, step-size $\gamma > 0$, parameter $\eta \in (0, 1]$, initial batch size $B_{\text{init}}$
2: Initialize $u_i^0 = v_i^0 = g_i^0 = \frac{1}{B_{\text{init}}} \sum_{j=1}^{B_{\text{init}}} \nabla f_i(x^0, \xi_{i,j})$ for $i = 1, \ldots, n$; $g^0 = \frac{1}{n} \sum_{i=1}^n g_i^0$
3: **for** $t = 0, 1, 2, \ldots, T-1$ **do**
4:     Master computes $x^{t+1} = x^t - \gamma g^t$ and broadcasts $x^{t+1}$ to all nodes
5:     **for all nodes** $i = 1, \ldots, n$ **in parallel do**
6:         Compute the first momentum estimator $v_i^{t+1} = (1 - \eta)v_i^t + \eta \nabla f_i(x^{t+1}, \xi_i^{t+1})$
7:         Compute the second momentum estimator $u_i^{t+1} = (1 - \eta)u_i^t + \eta v_i^{t+1}$
8:         Compress $c_i^{t+1} = \mathcal{C}(u_i^{t+1} - g_i^t)$ and send $c_i^{t+1}$ to the master
9:         Update local state $g_i^{t+1} = g_i^t + c_i^{t+1}$
10:     **end for**
11:     Master computes $g^{t+1} = \frac{1}{n} \sum_{i=1}^n g_i^{t+1}$ via $g^{t+1} = g^t + \frac{1}{n} \sum_{i=1}^n c_i^{t+1}$
12: **end for**

---

In this section, we state the detailed version of Corollary 3 in Theorem 5, followed by its formal proof. Notice that the key reason for the sample complexity improvement of the double momentum variant compared to EF21-SGDM (Theorem 3) is that in (27), one of the terms has better dependence on $\eta$ compared to (9) in Theorem 3, i.e., $\eta^4 \sigma^2 / \alpha$ instead of $\eta^2 \sigma^2 / \alpha$. As a result, this term is dominated by other terms and vanishes in Corollary 3.

**Theorem 5.** *Let Assumptions 1 and 2 hold. Let $\hat{x}^T$ be sampled uniformly at random from the iterates of the method. Let Algorithm 3 run with a contractive compressor. For all $\eta \in (0, 1]$ and $B_{\text{init}} \geq 1$, with $\gamma \leq \min\left\{\frac{\alpha}{60\widetilde{L}}, \frac{\eta}{16L}\right\}$, we have*

$$\mathbb{E}\left[\left\|\nabla f(\hat{x}^T)\right\|^2\right] \leq \mathcal{O}\left(\frac{\Psi_0}{\gamma T} + \frac{\eta^3 \sigma^2}{\alpha^2} + \frac{\eta^4 \sigma^2}{\alpha} + \frac{\eta \sigma^2}{n}\right), \tag{27}$$

*where $\Psi_0 := \delta_0 + \frac{\gamma}{\eta}\mathbb{E}\left[\left\|v^0 - \nabla f(x^0)\right\|^2\right] + \frac{\gamma \eta^4}{\alpha^2} \frac{1}{n} \sum_{i=1}^n \mathbb{E}\left[\left\|v_i^0 - \nabla f_i(x^0)\right\|^2\right]$. Setting initial batch size $B_{\text{init}} = \left\lceil \frac{\sigma^2}{L\delta_0}\right\rceil$, step-size and momentum parameters*

$$\gamma = \min\left\{\frac{\alpha}{60\widetilde{L}}, \frac{\eta}{16L}\right\}, \qquad \eta = \min\left\{1, \left(\frac{L\delta_0 \alpha^2}{\sigma^2 T}\right)^{1/4}, \left(\frac{L\delta_0 n}{\sigma^2 T}\right)^{1/2}, \frac{\alpha\sqrt{L\delta_0 B_{\text{init}}}}{\sigma}\right\}, \tag{28}$$

*we get*

$$\frac{1}{T}\sum_{t=0}^{T-1} \mathbb{E}\left[\left\|\nabla f(x^t)\right\|^2\right] \leq \mathcal{O}\left(\frac{\widetilde{L}\delta_0}{\alpha T} + \left(\frac{L\delta_0 \sigma^{2/3}}{\alpha^{2/3} T}\right)^{3/4} + \left(\frac{L\delta_0 \sigma^2}{nT}\right)^{1/2}\right).$$

*Proof.* In order to control the error between $g^t$ and $\nabla f(x^t)$, we decompose it into three terms

$$\begin{aligned}\left\|g^t - \nabla f(x^t)\right\|^2 &\leq 3\left\|g^t - u^t\right\|^2 + 3\left\|u^t - v^t\right\|^2 + 3\left\|v^t - \nabla f(x^t)\right\|^2 \\ &\leq 3\frac{1}{n}\sum_{i=1}^n \left\|g_i^t - u_i^t\right\|^2 + 3\left\|u^t - v^t\right\|^2 + 3\left\|v^t - \nabla f(x^t)\right\|^2,\end{aligned}$$

where we define the sequences $v^t := \frac{1}{n}\sum_{i=1}^n v_i^t$ and $u^t := \frac{1}{n}\sum_{i=1}^n u_i^t$. In the following, we develop a recursion for each term above separately.

**Part I. Controlling the error of momentum estimator for each $v_i^t$ and on average for $v^t$.** Denote $P_t := \mathbb{E}\left[\left\|v^t - \nabla f(x^t)\right\|^2\right]$, $\widetilde{P}_t := \frac{1}{n}\sum_{i=1}^n \mathbb{E}\left[\left\|v_i^t - \nabla f_i(x^t)\right\|^2\right]$, $R_t := \mathbb{E}\left[\left\|x^t - x^{t+1}\right\|^2\right]$. Similarly to Part I of the proof of Theorem 3, we have

$$\frac{1}{T}\sum_{t=0}^{T-1} \widetilde{P}_t \leq \frac{3\widetilde{L}^2}{\eta^2} \frac{1}{T}\sum_{t=0}^{T-1} R_t + \eta\sigma^2 + \frac{1}{\eta T}\widetilde{P}_0, \tag{29}$$

$$\frac{1}{T}\sum_{t=0}^{T-1} P_t \le \frac{3L^2}{\eta^2}\frac{1}{T}\sum_{t=0}^{T-1} R_t + \frac{\eta\sigma^2}{n} + \frac{1}{\eta T}P_0. \tag{30}$$

**Part II (a). Controlling the error of the second momentum estimator for each $u_i^t$.** Recall that by Lemma 4-(37), we have for each $i = 1, \ldots, n$, and any $0 < \eta \le 1$ and $t \ge 0$

$$\mathbb{E}\left[\left\|u_i^{t+1} - v_i^{t+1}\right\|^2\right] \le (1-\eta)\mathbb{E}\left[\left\|u_i^t - v_i^t\right\|^2\right] + 6\eta\mathbb{E}\left[\left\|v_i^t - \nabla f_i(x^t)\right\|^2\right]$$
$$+ 6L_i^2\eta\mathbb{E}\left[\left\|x^{t+1} - x^t\right\|^2\right] + \eta^2\sigma^2, \tag{31}$$

Averaging inequalities (31) over $i = 1, \ldots, n$ and denoting $\widetilde{Q}_t := \frac{1}{n}\sum_{i=1}^n \mathbb{E}\left[\left\|u_i^t - v_i^t\right\|^2\right]$, we have

$$\widetilde{Q}_{t+1} \le (1-\eta)\widetilde{Q}_t + 6\eta\widetilde{P}_t + 6\widetilde{L}^2\eta R_t + \eta^2\sigma^2.$$

Summing up the above inequalities for $t = 0, \ldots, T-1$, we derive

$$\frac{1}{T}\sum_{t=0}^{T-1}\widetilde{Q}_t \le \frac{6}{T}\sum_{t=0}^{T-1}\widetilde{P}_t + 6\widetilde{L}^2\frac{1}{T}\sum_{t=0}^{T-1} R_t + \eta\sigma^2 + \frac{1}{\eta T}\widetilde{Q}_0$$

$$\le \left(\frac{6\cdot 3\widetilde{L}^2}{\eta^2} + 6\widetilde{L}^2\right)\frac{1}{T}\sum_{t=0}^{T-1} R_t + 7\eta\sigma^2 + \frac{1}{\eta T}\widetilde{Q}_0 + \frac{6}{\eta T}\widetilde{P}_0$$

$$\le \frac{19\widetilde{L}^2}{\eta^2}\frac{1}{T}\sum_{t=0}^{T-1} R_t + 7\eta\sigma^2 + \frac{6}{\eta T}\widetilde{P}_0, \tag{32}$$

where we used (29), the bound $\eta \le 1$, and $u_i^0 = v_i^0$ for $i = 1, \ldots, n$.

**Part II (b). Controlling the error of the second momentum estimator $u^t$ (on average).** Similarly by Lemma 4-(38), we have for any $0 < \eta \le 1$ and $t \ge 0$

$$\mathbb{E}\left[\left\|u^{t+1} - v^{t+1}\right\|^2\right] \le (1-\eta)\mathbb{E}\left[\left\|u^t - v^t\right\|^2\right] + 6\eta\mathbb{E}\left[\left\|v^t - \nabla f(x^t)\right\|^2\right]$$
$$+ 6L^2\eta\mathbb{E}\left[\left\|x^{t+1} - x^t\right\|^2\right] + \frac{\eta^2\sigma^2}{n},$$

Summing up the above inequalities for $t = 0, \ldots, T-1$, and denoting $Q_t := \mathbb{E}\left[\left\|u^t - v^t\right\|^2\right]$, we derive

$$\frac{1}{T}\sum_{t=0}^{T-1} Q_t \le \frac{6}{T}\sum_{t=0}^{T-1} P_t + 6L^2\frac{1}{T}\sum_{t=0}^{T-1} R_t + \eta\sigma^2 + \frac{1}{\eta T}Q_0$$

$$\le \left(\frac{6\cdot 3L^2}{\eta^2} + 6L^2\right)\frac{1}{T}\sum_{t=0}^{T-1} R_t + \frac{7\eta\sigma^2}{n} + \frac{1}{\eta T}Q_0 + \frac{6}{\eta T}P_0$$

$$\le \frac{19L^2}{\eta^2}\frac{1}{T}\sum_{t=0}^{T-1} R_t + \frac{7\eta\sigma^2}{n} + \frac{6}{\eta T}P_0, \tag{33}$$

where we used (30), the bound $\eta \le 1$, and $u^0 = v^0$.

**Part III. Controlling the error of contractive compressor and the double momentum estimator.** By Lemma 5 we have for each $i = 1, \ldots, n$, and any $0 < \eta \le 1$ and $t \ge 0$

$$\mathbb{E}\left[\left\|g_i^{t+1} - u_i^{t+1}\right\|^2\right] \le \left(1 - \frac{\alpha}{2}\right)\mathbb{E}\left[\left\|g_i^t - u_i^t\right\|^2\right] + \frac{6\eta^2}{\alpha}\mathbb{E}\left[\left\|u_i^t - v_i^t\right\|^2\right] \tag{34}$$
$$+ \frac{6\eta^4}{\alpha}\mathbb{E}\left[\left\|v_i^t - \nabla f_i(x^t, \xi_i^t)\right\|^2\right] + \frac{6L_i^2\eta^4}{\alpha}\mathbb{E}\left[\left\|x^t - x^{t+1}\right\|^2\right] + \eta^4\sigma^2.$$

Averaging inequalities (34) over $i = 1, \ldots, n$, denoting $\widetilde{V}_t := \frac{1}{n}\sum_{i=1}^n \mathbb{E}\left[\left\|g_i^t - u_i^t\right\|^2\right]$, and summing up the resulting inequality for $t = 0, \ldots, T-1$, we obtain

$$\frac{1}{T}\sum_{t=0}^{T-1}\widetilde{V}_t \le \frac{12\eta^2}{\alpha^2}\frac{1}{T}\sum_{t=0}^{T-1}\widetilde{Q}_t + \frac{12\eta^4}{\alpha^2}\frac{1}{T}\sum_{t=0}^{T-1}\widetilde{P}_t + \frac{12\widetilde{L}^2\eta^4}{\alpha^2}\frac{1}{T}\sum_{t=0}^{T-1} R_t + \frac{2\eta^4\sigma^2}{\alpha}$$

$$
\begin{aligned}
\leq \quad & \frac{12\eta^2}{\alpha^2}\left(\frac{19\widetilde{L}^2}{\eta^2}\frac{1}{T}\sum_{t=0}^{T-1}R_t + 7\eta\sigma^2\right) + \frac{12\eta^4}{\alpha^2}\left(\frac{3\widetilde{L}^2}{\eta^2}\frac{1}{T}\sum_{t=0}^{T-1}R_t + \eta\sigma^2\right) \\
& + \frac{12\widetilde{L}^2\eta^4}{\alpha^2}\frac{1}{T}\sum_{t=0}^{T-1}R_t + \frac{2\eta^4\sigma^2}{\alpha} + \frac{12\eta^4}{\alpha^2 T}\widetilde{P}_0 \\
\leq \quad & \frac{12\cdot 19\widetilde{L}^2}{\alpha^2}\frac{1}{T}\sum_{t=0}^{T-1}R_t + \frac{12\cdot 7\eta^3\sigma^2}{\alpha^2} + \frac{12\cdot 3\widetilde{L}^2\eta^2}{\alpha^2}\frac{1}{T}\sum_{t=0}^{T-1}R_t + \frac{12\eta^5\sigma^2}{\alpha^2} \\
& + \frac{12\widetilde{L}^2\eta^4}{\alpha^2}\frac{1}{T}\sum_{t=0}^{T-1}R_t + \frac{2\eta^4\sigma^2}{\alpha} + \frac{12\eta^4}{\alpha^2 T}\widetilde{P}_0 \\
\leq \quad & \frac{276\widetilde{L}^2}{\alpha^2}\frac{1}{T}\sum_{t=0}^{T-1}R_t + \frac{84\eta^3\sigma^2}{\alpha^2} + \frac{12\eta^5\sigma^2}{\alpha^2} + \frac{2\eta^4\sigma^2}{\alpha} + \frac{12\eta^4}{\alpha^2 T}\widetilde{P}_0
\end{aligned}
$$

$$(35)$$

**Part IV. Combining steps I, II and III with descent lemma.** By smoothness (Assumption 1) of $f(\cdot)$ it follows from Lemma 1 that for any $\gamma \leq 1/(2L)$ we have

$$
\begin{aligned}
f(x^{t+1}) \quad \leq \quad & f(x^t) - \frac{\gamma}{2}\left\|\nabla f(x^t)\right\|^2 - \frac{1}{4\gamma}\left\|x^{t+1}-x^t\right\|^2 + \frac{\gamma}{2}\left\|g^t - \nabla f(x^t)\right\|^2 \qquad (36) \\
\leq \quad & f(x^t) - \frac{\gamma}{2}\left\|\nabla f(x^t)\right\|^2 - \frac{1}{4\gamma}\left\|x^{t+1}-x^t\right\|^2 \\
& + \frac{3\gamma}{2}\frac{1}{n}\sum_{i=1}^{n}\left\|g_i^t - u_i^t\right\|^2 + \frac{3\gamma}{2}\left\|u^t - v^t\right\|^2 + \frac{3\gamma}{2}\left\|v^t - \nabla f(x^t)\right\|^2.
\end{aligned}
$$

Subtracting $f^*$ from both sides of (36), taking expectation and defining $\delta_t := \mathbb{E}\left[f(x^t) - f^*\right]$, we derive

$$
\begin{aligned}
\mathbb{E}\left[\left\|\nabla f(\hat{x}^T)\right\|^2\right] \quad = \quad & \frac{1}{T}\sum_{t=0}^{T-1}\mathbb{E}\left[\left\|\nabla f(x^t)\right\|^2\right] \\
\leq \quad & \frac{2\delta_0}{\gamma T} + 3\frac{1}{T}\sum_{t=0}^{T-1}\widetilde{V}_t + 3\frac{1}{T}\sum_{t=0}^{T-1}Q_t + 3\frac{1}{T}\sum_{t=0}^{T-1}P_t - \frac{1}{2\gamma^2}\frac{1}{T}\sum_{t=0}^{T-1}R_t \\
\overset{(i)}{\leq} \quad & \frac{2\delta_0}{\gamma T} + \frac{3\cdot 276\widetilde{L}^2}{\alpha^2}\frac{1}{T}\sum_{t=0}^{T-1}R_t + \frac{3\cdot 84\eta^3\sigma^2}{\alpha^2} + \frac{3\cdot 12\eta^5\sigma^2}{\alpha^2} + \frac{3\cdot 2\eta^4\sigma^2}{\alpha} \\
& + \frac{3\cdot 19L^2}{\eta^2}\frac{1}{T}\sum_{t=0}^{T-1}R_t + \frac{3\cdot 7\eta\sigma^2}{n} \\
& + \frac{3L^2}{\eta^2}\frac{1}{T}\sum_{t=0}^{T-1}R_t + \frac{\eta\sigma^2}{n} - \frac{1}{2\gamma^2}\frac{1}{T}\sum_{t=0}^{T-1}R_t \\
& + \frac{36\eta^4}{\alpha^2 T}\widetilde{P}_0 + \frac{18}{\eta T}P_0 + \frac{3}{\eta T}P_0 \\
= \quad & \frac{2\delta_0}{\gamma T} + \frac{3\cdot 84\eta^3\sigma^2}{\alpha^2} + \frac{3\cdot 12\eta^5\sigma^2}{\alpha^2} + \frac{3\cdot 2\eta^4\sigma^2}{\alpha} + \frac{22\eta\sigma^2}{n} \\
& + \left(\frac{60L^2}{\eta^2} + \frac{3\cdot 276\widetilde{L}^2}{\alpha^2} - \frac{1}{2\gamma^2}\right)\frac{1}{T}\sum_{t=0}^{T-1}R_t \\
& + \frac{36\eta^4}{\alpha^2 T}\widetilde{P}_0 + \frac{21}{\eta T}P_0 \\
= \quad & \frac{2\delta_0}{\gamma T} + \frac{288\eta^3\sigma^2}{\alpha^2} + \frac{6\eta^4\sigma^2}{\alpha} + \frac{22\eta\sigma^2}{n} + \frac{36\eta^4}{\alpha^2 T}\widetilde{P}_0 + \frac{21}{\eta T}P_0.
\end{aligned}
$$

where $(i)$ holds due to (30), (35) and (33), the last two steps hold because of the assumption on the step-size, and $\eta \leq 1$, which completes the proof of the first part of Theorem.

Notice that it suffices to take the same initial batch-size as in the proof of the Theorem 3 in order to "remove" $\widetilde{P}_0$ and $P_0$ terms, since the power of $\eta$ in front of $\widetilde{P}_0$ is larger here compared to the proof of Theorem 3. The choice of the momentum parameter such that $\eta \leq \left(\frac{L\delta_0\alpha^2}{\sigma^2 T}\right)^{1/4}$, $\eta \leq \left(\frac{L\delta_0 n}{\sigma^2 T}\right)^{1/2}$ ensures that $\frac{\eta^3 \sigma^2}{\alpha^2} \leq \frac{L\delta_0}{\eta T}$, and $\frac{\eta\sigma^2}{n} \leq \frac{L\delta_0}{\eta T}$. Therefore, we can guarantee that the choice $\eta = \min\left\{\frac{\alpha\sqrt{L\delta_0 B_{\mathrm{init}}}}{\sigma}, \left(\frac{L\delta_0\alpha^2}{\sigma^2 T}\right)^{1/4}, \left(\frac{L\delta_0 n}{\sigma^2 T}\right)^{1/2}\right\}$ ensures that

$$\mathbb{E}\left[\|\nabla f(\hat{x}^T)\|^2\right] \leq \mathcal{O}\left(\frac{\widetilde{L}\delta_0}{\alpha T} + \left(\frac{L\delta_0\sigma^{2/3}}{\alpha^{2/3}T}\right)^{3/4} + \left(\frac{L\delta_0\sigma^2}{nT}\right)^{1/2}\right).$$

$\square$

### G.1 Controlling the error of second momentum estimator

**Lemma 4.** *Let Assumption 1 be satisfied, and suppose $0 < \eta \leq 1$. For every $i = 1, \ldots, n$, let the sequences $\{v_i^t\}_{t\geq 0}$ and $\{u_i^t\}_{t\geq 0}$ be updated via*

$$v_i^t = v_i^{t-1} + \eta\left(\nabla f_i(x^t, \xi_i^t) - v_i^{t-1}\right),$$

$$u_i^t = u_i^{t-1} + \eta\left(v_i^t - u_i^{t-1}\right).$$

*Define the sequences $v^t := \frac{1}{n}\sum_{i=1}^n v_i^t$ and $u^t := \frac{1}{n}\sum_{i=1}^n u_i^t$. Then for every $i = 1, \ldots, n$ and $t \geq 0$ it holds*

$$\mathbb{E}\left[\|u_i^t - v_i^t\|^2\right] \leq (1-\eta)\mathbb{E}\left[\|u_i^{t-1} - v_i^{t-1}\|^2\right] + 6\eta\mathbb{E}\left[\|v_i^{t-1} - \nabla f_i(x^{t-1})\|^2\right]$$
$$+6L_i^2\eta\mathbb{E}\left[\|x^t - x^{t-1}\|^2\right] + \eta^2\sigma^2, \qquad (37)$$

$$\mathbb{E}\left[\|u^t - v^t\|^2\right] \leq (1-\eta)\mathbb{E}\left[\|u^{t-1} - v^{t-1}\|^2\right] + 6\eta\mathbb{E}\left[\|v^{t-1} - \nabla f(x^{t-1})\|^2\right]$$
$$+6L^2\eta\mathbb{E}\left[\|x^t - x^{t-1}\|^2\right] + \frac{\eta^2\sigma^2}{n}. \qquad (38)$$

*Proof.* By the update rule of $v_i^t$, we have

$$\mathbb{E}\left[\|u_i^t - v_i^t\|^2\right] = \mathbb{E}\left[\|u_i^{t-1} - v_i^t + \eta(v_i^t - u_i^{t-1})\|^2\right]$$
$$= (1-\eta)^2\mathbb{E}\left[\|v_i^t - u_i^{t-1}\|^2\right]$$
$$= (1-\eta)^2\mathbb{E}\left[\|(1-\eta)v_i^{t-1} + \eta\nabla f_i(x^t, \xi_i^t) - u_i^{t-1}\|^2\right]$$
$$= (1-\eta)^2\mathbb{E}\left[\|(u_i^{t-1} - v_i^{t-1}) + \eta(v_i^{t-1} - \nabla f_i(x^t, \xi_i^t))\|^2\right]$$
$$= (1-\eta)^2\mathbb{E}\left[\|(u_i^{t-1} - v_i^{t-1}) + \eta(v_i^{t-1} - \nabla f_i(x^t)) + \eta(\nabla f_i(x^t) - \nabla f_i(x^t, \xi_i^t))\|^2\right]$$
$$= (1-\eta)^2\mathbb{E}\left[\mathbb{E}_{\xi_i^t}\left[\|(u_i^{t-1} - v_i^{t-1}) + \eta(v_i^{t-1} - \nabla f_i(x^t)) + \eta(\nabla f_i(x^t) - \nabla f_i(x^t, \xi_i^t))\|^2\right]\right]$$
$$= (1-\eta)^2\left(\mathbb{E}\left[\|u_i^{t-1} - v_i^{t-1} + \eta(v_i^{t-1} - \nabla f_i(x^t))\|^2\right] + \eta^2\mathbb{E}\left[\|\nabla f_i(x^t, \xi_i^t) - \nabla f_i(x^t)\|^2\right]\right)$$
$$\leq (1-\eta)^2\mathbb{E}\left[\|u_i^{t-1} - v_i^{t-1} + \eta(v_i^{t-1} - \nabla f_i(x^t))\|^2\right] + \eta^2\sigma^2$$
$$\leq (1-\eta)^2\left(1 + \frac{\eta}{2}\right)\mathbb{E}\left[\|u_i^{t-1} - v_i^{t-1}\|^2\right]$$
$$+ \left(1 + \frac{2}{\eta}\right)\eta^2\mathbb{E}\left[\|v_i^{t-1} - \nabla f_i(x^t)\|^2\right] + \eta^2\sigma^2$$

$$\leq (1-\eta)\mathbb{E}\left[\left\|u_i^{t-1} - v_i^{t-1}\right\|^2\right] + 3\eta\mathbb{E}\left[\left\|v_i^{t-1} - \nabla f_i(x^t)\right\|^2\right] + \eta^2\sigma^2$$

$$\leq (1-\eta)\mathbb{E}\left[\left\|u_i^{t-1} - v_i^{t-1}\right\|^2\right] + 6\eta\mathbb{E}\left[\left\|v_i^{t-1} - \nabla f_i(x^{t-1})\right\|^2\right]$$
$$+ 6\eta\mathbb{E}\left[\left\|\nabla f_i(x^t) - \nabla f_i(x^{t-1})\right\|^2\right] + \eta^2\sigma^2$$

$$\leq (1-\eta)\mathbb{E}\left[\left\|u_i^{t-1} - v_i^{t-1}\right\|^2\right] + 6\eta\mathbb{E}\left[\left\|v_i^{t-1} - \nabla f_i(x^{t-1})\right\|^2\right]$$
$$+ 6L_i^2\eta\mathbb{E}\left[\left\|x^t - x^{t-1}\right\|^2\right] + \eta^2\sigma^2,$$

where the first inequality holds Assumption 2, the second inequality holds by Young's inequality, and the last step uses smoothness of $f_i(\cdot)$ (Assumption 1), which concludes the proof of (37).

For each $t = 0, \ldots, T-1$, define a random vector $\xi^t := (\xi_1^t, \ldots, \xi_n^t)$ and denote by $\nabla f(x^t, \xi^{t+1}) := \frac{1}{n}\sum_{i=1}^n \nabla f_i(x^t, \xi_i^t)$. Note that the entries of the random vector $\xi^t$ are independent and $\mathbb{E}_{\xi^t}\left[\nabla f(x^t, \xi^t)\right] = \nabla f(x^t)$, then we have

$$v^t = v^{t-1} + \eta\left(\nabla f(x^t, \xi^t) - v^{t-1}\right),$$

$$u_i^t = u_i^{t-1} + \eta\left(v_i^t - u_i^{t-1}\right),$$

where $v^t := \frac{1}{n}\sum_{i=1}^n v_i^t$, $u^t := \frac{1}{n}\sum_{i=1}^n u_i^t$ are auxiliary sequences. Therefore, we can similarly derive

$$\mathbb{E}\left[\left\|u^t - v^t\right\|^2\right] = \mathbb{E}\left[\left\|u^{t-1} - v^t + \eta(v^t - u^{t-1})\right\|^2\right]$$

$$= (1-\eta)^2\mathbb{E}\left[\left\|v^t - u^{t-1}\right\|^2\right]$$

$$= (1-\eta)^2\mathbb{E}\left[\left\|(1-\eta)v^{t-1} + \eta\nabla f(x^t, \xi^t) - u^{t-1}\right\|^2\right]$$

$$= (1-\eta)^2\mathbb{E}\left[\left\|(u^{t-1} - v^{t-1}) + \eta(v^{t-1} - \nabla f(x^t, \xi^t))\right\|^2\right]$$

$$= (1-\eta)^2\mathbb{E}\left[\left\|(u^{t-1} - v^{t-1}) + \eta(v^{t-1} - \nabla f(x^t)) + \eta(\nabla f(x^t) - \nabla f(x^t, \xi^t))\right\|^2\right]$$

$$= (1-\eta)^2\mathbb{E}\left[\mathbb{E}_{\xi^t}\left[\left\|(u^{t-1} - v^{t-1}) + \eta(v^{t-1} - \nabla f(x^t)) + \eta(\nabla f(x^t) - \nabla f(x^t, \xi^t))\right\|^2\right]\right]$$

$$= (1-\eta)^2\left(\mathbb{E}\left[\left\|u^{t-1} - v^{t-1} + \eta(v^{t-1} - \nabla f(x^t))\right\|^2\right] + \eta^2\mathbb{E}\left[\left\|\nabla f(x^t, \xi^t) - \nabla f(x^t)\right\|^2\right]\right)$$

$$\leq (1-\eta)^2\mathbb{E}\left[\left\|u^{t-1} - v^{t-1} + \eta(v^{t-1} - \nabla f(x^t))\right\|^2\right] + \frac{\eta^2\sigma^2}{n}$$

$$\leq (1-\eta)^2\left(1 + \frac{\eta}{2}\right)\mathbb{E}\left[\left\|u^{t-1} - v^{t-1}\right\|^2\right]$$
$$+ \left(1 + \frac{2}{\eta}\right)\eta^2\mathbb{E}\left[\left\|v^{t-1} - \nabla f(x^t)\right\|^2\right] + \frac{\eta^2\sigma^2}{n}$$

$$\leq (1-\eta)\mathbb{E}\left[\left\|u^{t-1} - v^{t-1}\right\|^2\right] + 3\eta\mathbb{E}\left[\left\|v^{t-1} - \nabla f(x^t)\right\|^2\right] + \frac{\eta^2\sigma^2}{n}$$

$$\leq (1-\eta)\mathbb{E}\left[\left\|u^{t-1} - v^{t-1}\right\|^2\right] + 6\eta\mathbb{E}\left[\left\|v^{t-1} - \nabla f(x^{t-1})\right\|^2\right]$$
$$+ 6\eta\mathbb{E}\left[\left\|\nabla f(x^t) - \nabla f(x^{t-1})\right\|^2\right] + \frac{\eta^2\sigma^2}{n}$$

$$\leq (1-\eta)\mathbb{E}\left[\left\|u^{t-1} - v^{t-1}\right\|^2\right] + 6\eta\mathbb{E}\left[\left\|v^{t-1} - \nabla f(x^{t-1})\right\|^2\right]$$
$$+ 6L^2\eta\mathbb{E}\left[\left\|x^t - x^{t-1}\right\|^2\right] + \frac{\eta^2\sigma^2}{n},$$

where the first inequality holds Assumption 2, the second inequality holds by Young's inequality, and the last step uses smoothness of $f(\cdot)$ (Assumption 1), which concludes the proof of (38). $\qquad\square$

## G.2 Controlling the error of contractive compression and double momentum estimator

**Lemma 5.** *Let Assumption 1 be satisfied, and suppose $\mathcal{C}$ is a contractive compressor. For every $i = 1, \ldots, n$, let the sequences $\{v_i^t\}_{t \geq 0}$, $\{u_i^t\}_{t \geq 0}$, and $\{g_i^t\}_{t \geq 0}$ be updated via*

$$
\begin{aligned}
v_i^t &= v_i^{t-1} + \eta \left( \nabla f_i(x^t, \xi_i^t) - v_i^{t-1} \right), \\
u_i^t &= u_i^{t-1} + \eta \left( v_i^t - u_i^{t-1} \right), \\
g_i^t &= g_i^{t-1} + \mathcal{C} \left( u_i^t - g_i^{t-1} \right).
\end{aligned}
$$

*Then for every $i = 1, \ldots, n$ and $t \geq 0$ it holds*

$$
\begin{aligned}
\mathbb{E}\left[ \left\| g_i^t - u_i^t \right\|^2 \right] &\leq \left( 1 - \frac{\alpha}{2} \right) \mathbb{E}\left[ \left\| g_i^{t-1} - u_i^{t-1} \right\|^2 \right] + \frac{6\eta^2}{\alpha} \mathbb{E}\left[ \left\| u_i^{t-1} - v_i^{t-1} \right\|^2 \right] \\
&\quad + \frac{6\eta^4}{\alpha} \mathbb{E}\left[ \left\| v_i^{t-1} - \nabla f_i(x^{t-1}, \xi_i^{t-1}) \right\|^2 \right] + \frac{6L_i^2 \eta^4}{\alpha} \mathbb{E}\left[ \left\| x^t - x^{t-1} \right\|^2 \right] + \eta^4 \sigma^2.
\end{aligned}
\tag{39}
$$

*Proof.* By the update rules of $g_i^t$, $u_i^t$ and $v_i^t$, we derive

$$
\begin{aligned}
\mathbb{E}\left[ \left\| g_i^t - u_i^t \right\|^2 \right] &= \mathbb{E}\left[ \left\| g_i^{t-1} - u_i^t + \mathcal{C}(u_i^t - g_i^{t-1}) \right\|^2 \right] \\
&\overset{(i)}{\leq} (1 - \alpha) \mathbb{E}\left[ \left\| u_i^t - g_i^{t-1} \right\|^2 \right] \\
&\overset{(ii)}{=} (1 - \alpha) \mathbb{E}\left[ \left\| u_i^{t-1} - g_i^{t-1} + \eta(v_i^{t-1} - u_i^{t-1}) + \eta^2(\nabla f_i(x^t, \xi_i^t) - v_i^{t-1}) \right\|^2 \right] \\
&= (1 - \alpha) \mathbb{E}\big[ \| u_i^{t-1} - g_i^{t-1} + \eta(v_i^{t-1} - u_i^{t-1}) + \eta^2(\nabla f_i(x^t) - v_i^{t-1}) \\
&\qquad + \eta^2(\nabla f_i(x^t, \xi_i^t) - \nabla f_i(x^t)) \|^2 \big] \\
&= (1 - \alpha) \mathbb{E}\big[ \| \mathbb{E}_{\xi_i^t}[ \| u_i^{t-1} - g_i^{t-1} + \eta(v_i^{t-1} - u_i^{t-1}) + \eta^2(\nabla f_i(x^t) - v_i^{t-1}) \\
&\qquad + \eta^2(\nabla f_i(x^t, \xi_i^t) - \nabla f_i(x^t)) \|^2] \big] \\
&= (1 - \alpha) \mathbb{E}\left[ \left\| u_i^{t-1} - g_i^{t-1} + \eta(v_i^{t-1} - u_i^{t-1}) + \eta^2(\nabla f_i(x^t) - v_i^{t-1}) \right\|^2 \right] \\
&\quad + (1 - \alpha)\eta^4 \mathbb{E}\left[ \left\| \nabla f_i(x^t, \xi_i^t) - \nabla f_i(x^t) \right\|^2 \right] \\
&\overset{(iii)}{\leq} (1 - \alpha)(1 + \rho) \mathbb{E}\left[ \left\| u_i^{t-1} - g_i^{t-1} \right\|^2 \right] \\
&\quad + (1 - \alpha)(1 + \rho^{-1}) \mathbb{E}\left[ \left\| \eta(v_i^{t-1} - u_i^{t-1}) + \eta^2(\nabla f_i(x^t) - v_i^{t-1}) \right\|^2 \right] \\
&\quad + \eta^4 \sigma^2 \\
&\overset{(iv)}{=} (1 - \theta) \mathbb{E}\left[ \left\| u_i^{t-1} - g_i^{t-1} \right\|^2 \right] + \eta^4 \sigma^2 \\
&\quad + \beta \mathbb{E}\left[ \left\| \eta(v_i^{t-1} - u_i^{t-1}) + \eta^2(\nabla f_i(x^{t-1}) - v_i^{t-1}) + \eta^2(\nabla f_i(x^t) - \nabla f_i(x^{t-1})) \right\|^2 \right] \\
&\overset{(v)}{\leq} (1 - \theta) \mathbb{E}\left[ \left\| u_i^{t-1} - g_i^{t-1} \right\|^2 \right] + 3\beta\eta^2 \mathbb{E}\left[ \left\| v_i^{t-1} - u_i^{t-1} \right\|^2 \right] \\
&\quad + 3\beta\eta^4 \mathbb{E}\left[ \left\| v_i^{t-1} - \nabla f_i(x^{t-1}) \right\|^2 \right] \\
&\quad + 3\beta\eta^4 \mathbb{E}\left[ \left\| \nabla f_i(x^t) - \nabla f_i(x^{t-1}) \right\|^2 \right] + \eta^4 \sigma^2 \\
&\leq (1 - \theta) \mathbb{E}\left[ \left\| u_i^{t-1} - g_i^{t-1} \right\|^2 \right] + 3\beta\eta^2 \mathbb{E}\left[ \left\| v_i^{t-1} - u_i^{t-1} \right\|^2 \right] \\
&\quad + 3\beta\eta^4 \mathbb{E}\left[ \left\| v_i^{t-1} - \nabla f_i(x^{t-1}) \right\|^2 \right] \\
&\quad + 3\beta L_i^2 \eta^4 \mathbb{E}\left[ \left\| x^t - x^{t-1} \right\|^2 \right] + \eta^4 \sigma^2
\end{aligned}
$$

where $(i)$ is due to definition of a contractive compressor (Definition 1), $(ii)$ follows by the update rule of $v_i^t$ and $u_i^t$, $(iii)$ and $(v)$ hold by Young's inequality for any $\rho > 0$. In $(iv)$, we introduced

the notation $\theta := 1 - (1-\alpha)(1+\rho)$, and $\beta := (1-\alpha)(1+\rho^{-1})$. The last step follows by smoothness of $f_i(\cdot)$ (Assumption 1). The proof is complete by the choice $\rho = \alpha/2$, which guarantees $1 - \theta \le 1 - \alpha/2$, and $3\beta \le 6/\alpha$. $\qquad\square$

# H  EF21-SGDM with Absolute Compressor

In this section, we complement our theory by analyzing EF21-SGDM under a different class of widely used biased compressors, namely, absolute compressors, which are defined as follows.

**Definition 2** (Absolute compressors). *We say that a (possibly randomized) map $\mathcal{C} : \mathbb{R}^d \to \mathbb{R}^d$ is an absolute compression operator if there exists a constant $\Delta > 0$ such that*

$$\mathbb{E}\left[\|\mathcal{C}(x) - x\|^2\right] \leq \Delta^2, \qquad \forall x \in \mathbb{R}^d. \tag{40}$$

This class includes important examples of compressors such as hard-threshold sparsifier [Sahu et al., 2021], (stochatsic) rounding schemes with bounded error [Gupta et al., 2015] and scaled integer rounding [Sapio et al., 2021].

---

**Algorithm 4** EF21-SGDM (abs)

---

1: **Input:** starting point $x^0$, step-size $\gamma > 0$, momentum $\eta \in (0, 1]$, initial batch size $B_{\text{init}}$
2: Initialize $v_i^0 = g_i^0 = \frac{1}{B_{\text{init}}} \sum_{j=1}^{B_{\text{init}}} \nabla f_i(x^0, \xi_{i,j}^0)$ for $i = 1, \ldots, n$; $g^0 = \frac{1}{n} \sum_{i=1}^{n} g_i^0$
3: **for** $t = 0, 1, 2, \ldots, T - 1$ **do**
4:     Master computes $x^{t+1} = x^t - \gamma g^t$ and broadcasts $x^{t+1}$ to all nodes
5:     **for all nodes** $i = 1, \ldots, n$ **in parallel do**
6:         Compute momentum estimator $v_i^{t+1} = (1 - \eta)v_i^t + \eta \nabla f_i(x^{t+1}, \xi_i^{t+1})$
7:         Compress $c_i^{t+1} = \mathcal{C}\left(\frac{v_i^{t+1} - g_i^t}{\gamma}\right)$ and send $c_i^{t+1}$ to the master
8:         Update local state $g_i^{t+1} = g_i^t + \gamma c_i^{t+1}$
9:     **end for**
10:    Master computes $g^{t+1} = \frac{1}{n} \sum_{i=1}^{n} g_i^{t+1}$ via $g^{t+1} = g^t + \frac{1}{n} \sum_{i=1}^{n} \gamma c_i^{t+1}$
11: **end for**

---

To accomodate absolute compressors into our EF21-SGDM method, we need to make a slight modification to our algorithm, see Algorithm 4. At each iteration, before compressing the difference $v_i^{t+1} - g_i^t$, we divide it by the step-size $\gamma$. Later, we multiply the compressed vector $c_i^{t+1}$ by $\gamma$, i.e., have

$$g_i^{t+1} = g_i^t + \gamma \mathcal{C}\left(\frac{v_i^{t+1} - g_i^t}{\gamma}\right).$$

Such modification is necessary for absolute compressors because by Definition 2 the compression error is not proportional to $\|x\|^2$, but merely an absolute constant $\Delta^2$. In fact, Algorithm 4 is somewhat more universal in the sense that it can be also applied for contractive compressors.[20] We derive the following result for EF21-SGDM (abs).

**Theorem 6.** *Let Assumptions 1 and 2 hold. Let $\hat{x}^T$ be sampled uniformly at random from the iterates of the method. Let Algorithm 4 run with an absolute compressor (Definition 2). For all $\eta \in (0, 1]$ and $B_{\text{init}} \geq 1$, with $\gamma \leq \frac{\eta}{4L}$, we have*

$$\mathbb{E}\left[\|\nabla f(\hat{x}^T)\|^2\right] \leq \mathcal{O}\left(\frac{\Psi_0}{\gamma T} + \gamma^2 \Delta^2 + \frac{\eta \sigma^2}{n}\right), \tag{41}$$

*where $\Psi_0 := \delta_0 + \frac{\gamma}{\eta}\mathbb{E}\left[\|v^0 - \nabla f(x^0)\|^2\right]$ is a Lyapunov function. With the following step-size, momentum parameter, and initial batch size*

$$\gamma = \frac{\eta}{4L}, \qquad \eta = \min\left\{1, \left(\frac{L^3 \delta_0}{\Delta^2 T}\right)^{1/3}, \left(\frac{L\delta_0 n}{\sigma^2 T}\right)^{1/2}\right\}, \qquad B_{init} = \frac{\sigma^2}{L\delta_0 n} \tag{42}$$

*we have*

$$\mathbb{E}\left[\|\nabla f(\hat{x}^T)\|^2\right] \leq \mathcal{O}\left(\frac{L\delta_0}{T} + \left(\frac{\delta_0 \Delta}{T}\right)^{2/3} + \left(\frac{L\delta_0 \sigma^2}{nT}\right)^{1/2}\right).$$

---

[20]It is straightforward to modify the proof of our Theorem 3 for the case when Algorithm 4 is applied with a contractive compressor.

**Corollary 4.** *Under the setting of Theorem 6, we have $\mathbb{E}\left[\left\|\nabla f(\hat{x}^T)\right\|\right] \leq \varepsilon$ after $T = \mathcal{O}\left(\frac{L\delta_0}{\varepsilon^2} + \frac{\Delta\delta_0}{\varepsilon^3} + \frac{\sigma^2 L\delta_0}{n\varepsilon^4}\right)$ iterations.*

**Remark 3.** *The sample complexity result in Corollary 4 matches the one derived for* DoubleSqueeze *algorithm [Tang et al., 2020], which is different from Algorithm 4.*

*Proof.* Similarly to the proof of Theorem 3, we control the error between $g^t$ and $\nabla f(x^t)$ by decomposing it into two terms

$$\left\|g^t - \nabla f(x^t)\right\|^2 \leq 2\left\|g^t - v^t\right\|^2 + 2\left\|v^t - \nabla f(x^t)\right\|^2 \leq 2\frac{1}{n}\sum_{i=1}^{n}\left\|g_i^t - v_i^t\right\|^2 + 2\left\|v^t - \nabla f(x^t)\right\|^2.$$

Again, for the second term above we can use the recursion developed for momentum estimator Lemma 2. However, since we use a different compressor here, we need to bound $\left\|g_i^t - v_i^t\right\|^2$ term differently, thus we invoke Lemma 6 for absolute compressor.

**Part I. Controlling the error of momentum estimator on average for $v^t$.** Denote $P_t := \mathbb{E}\left[\left\|v^t - \nabla f(x^t)\right\|^2\right]$, $R_t := \mathbb{E}\left[\left\|x^t - x^{t+1}\right\|^2\right]$. Similarly to Part I of the proof of Theorem 3, we have by Lemma 2

$$\frac{1}{T}\sum_{t=0}^{T-1} P_t \leq \frac{3L^2}{\eta^2}\frac{1}{T}\sum_{t=0}^{T-1} R_t + \frac{\eta\sigma^2}{n} + \frac{1}{\eta T}P_0. \tag{43}$$

**Part II. Controlling the error of absolute compressor and momentum estimator.** By Lemma 6 we have for any $0 < \eta \leq 1$ and $t \geq 0$

$$\widetilde{V}_t := \frac{1}{n}\sum_{i=1}^{n}\mathbb{E}\left[\left\|g_i^t - v_i^t\right\|^2\right] \leq \gamma^2\Delta^2. \tag{44}$$

**Part III. Combining steps I and II with descent lemma.** By smoothness (Assumption 1) of $f(\cdot)$ it follows from Lemma 1 that for any $\gamma \leq 1/(2L)$ we have

$$
\begin{aligned}
f(x^{t+1}) &\leq f(x^t) - \frac{\gamma}{2}\left\|\nabla f(x^t)\right\|^2 - \frac{1}{4\gamma}\left\|x^{t+1} - x^t\right\|^2 + \frac{\gamma}{2}\left\|g^t - \nabla f(x^t)\right\|^2 \quad (45)\\
&\leq f(x^t) - \frac{\gamma}{2}\left\|\nabla f(x^t)\right\|^2 - \frac{1}{4\gamma}\left\|x^{t+1} - x^t\right\|^2 + \gamma\widetilde{V}_t + \gamma P_t.
\end{aligned}
$$

Subtracting $f^*$ from both sides of (45), taking expectation and defining $\delta_t := \mathbb{E}\left[f(x^t) - f^*\right]$, we derive

$$
\begin{aligned}
\mathbb{E}\left[\left\|\nabla f(\hat{x}^T)\right\|^2\right] &= \frac{1}{T}\sum_{t=0}^{T-1}\mathbb{E}\left[\left\|\nabla f(x^t)\right\|^2\right]\\
&\leq \frac{2\delta_0}{\gamma T} + 2\frac{1}{T}\sum_{t=0}^{T-1}\widetilde{V}_t + 2\frac{1}{T}\sum_{t=0}^{T-1}P_t - \frac{1}{2\gamma^2}\frac{1}{T}\sum_{t=0}^{T-1}R_t\\
&\overset{(i)}{\leq} \frac{2\delta_0}{\gamma T} + 2\gamma^2\Delta^2 + 2\frac{1}{T}\sum_{t=0}^{T-1}P_t - \frac{1}{2\gamma^2}\frac{1}{T}\sum_{t=0}^{T-1}R_t\\
&\overset{(ii)}{\leq} \frac{2\delta_0}{\gamma T} + 2\gamma^2\Delta^2 + \left(\frac{6L^2}{\eta^2} - \frac{1}{2\gamma^2}\right)\frac{1}{T}\sum_{t=0}^{T-1}R_t + \frac{2\eta\sigma^2}{n} + \frac{1}{\eta T}P_0\\
&\leq \frac{2\delta_0}{\gamma T} + 2\gamma^2\Delta^2 + \frac{2\eta\sigma^2}{n} + \frac{1}{\eta T}P_0
\end{aligned}
$$
$$\tag{46}$$

where in $(i)$ and $(ii)$ we apply (43), (44), and in the last step we use the assumption on the step-size $\gamma \leq \eta/(4L)$.

Setting $\gamma = \frac{\eta}{4L}$, and taking $\eta \leq \left(\frac{L^3 \delta_0}{\Delta^2 T}\right)^{1/3}$ we can ensure that $\frac{\eta^2 \Delta^2}{L^2} \leq \frac{L \delta_0}{\eta T}$, since $\eta \leq \left(\frac{L \delta_0 n}{\sigma^2 T}\right)^{1/2}$ we have $\frac{\eta \sigma^2}{n} \leq \frac{L \delta_0}{\eta T}$. Finally, by setting the initial batch-size to $B_{init} = \frac{\sigma^2}{L \delta_0 n}$, we have $\frac{1}{\eta T} P_0 = \frac{\sigma^2}{\eta T n B_{init}} \leq \frac{L \delta_0}{\eta T}$. Therefore, we derive

$$
\begin{aligned}
\frac{1}{T} \sum_{t=0}^{T-1} \mathbb{E}\left[\|\nabla f(x^t)\|^2\right] &\leq \frac{2\delta_0}{\gamma T} + 2\gamma^2 \Delta^2 + \frac{2\eta\sigma^2}{n} + \frac{1}{\eta T} P_0 \\
&= \frac{8L\delta_0}{\eta T} + \frac{\eta^2 \Delta^2}{8L^2} + \frac{2\eta\sigma^2}{n} + \frac{\sigma^2}{\eta T B_{init}} \\
&= \mathcal{O}\left(\frac{L\delta_0}{T} + \frac{\delta_0^{2/3} \Delta^{2/3}}{T^{2/3}} + \frac{\sigma(L\delta_0)^{1/2}}{(nT)^{1/2}}\right).
\end{aligned}
$$

(47)

$\square$

## H.1 Controlling the error of absolute compression

**Lemma 6.** *Let $\mathcal{C}$ be an absolute compressor and $g_i^{t+1}$ be updated according to Algorithm 4, then for $t \geq 0$, we have $\frac{1}{n} \sum_{i=1}^n \mathbb{E}\left[\|g_i^t - v_i^t\|^2\right] \leq \gamma^2 \Delta^2$.*

*Proof.* By the update rule for $g_i^{t+1}$ in Algorithm 4 and Definition 2, we can bound

$$
\begin{aligned}
\mathbb{E}\left[\|g_i^{t+1} - v_i^{t+1}\|^2\right] &= \mathbb{E}\left[\left\|\gamma \mathcal{C}\left(\frac{v_i^{t+1} - g_i^t}{\gamma}\right) - (v_i^{t+1} - g_i^t)\right\|^2\right] \\
&= \gamma^2 \mathbb{E}\left[\left\|\mathcal{C}\left(\frac{v_i^{t+1} - g_i^t}{\gamma}\right) - \frac{v_i^{t+1} - g_i^t}{\gamma}\right\|^2\right] \leq \gamma^2 \Delta^2.
\end{aligned}
$$

$\square$

# I   EF21-STORM/MVR

---

**Algorithm 5** EF21-STORM/MVR

---

1: **Input:** $x^0$, step-size $\gamma > 0$, parameter $\eta \in (0,1]$, $B_{\text{init}} \geq 1$
2: Initialize $w_i^0 = g_i^0 = \frac{1}{B_{\text{init}}} \sum_{j=1}^{B_{\text{init}}} \nabla f_i(x^0, \xi_{i,j}^0)$ for $i = 1, \ldots, n$; $g^0 = \frac{1}{n} \sum_{i=1}^{n} g_i^0$
3: **for** $t = 0,1,2,\ldots,T-1$ **do**
4:     Master computes $x^{t+1} = x^t - \gamma g^t$ and broadcasts $x^{t+1}$ to all nodes
5:     **for all nodes** $i = 1, \ldots, n$ **in parallel do**
6:         Draw $\xi_i^{t+1}$ and compute two (stochastic) gradients $\nabla f_i(x^t, \xi_i^{t+1})$ and $\nabla f_i(x^{t+1}, \xi_i^{t+1})$
7:         Compute variance reduced STORM/MVR estimator
8:         $w_i^{t+1} = \nabla f_i(x^{t+1}, \xi_i^{t+1}) + (1-\eta)(w_i^t - \nabla f_i(x^t, \xi_i^{t+1}))$
9:         Compress $c_i^{t+1} = \mathcal{C}(w_i^{t+1} - g_i^t)$ and send $c_i^{t+1}$ to the master
10:         Update local state $g_i^{t+1} = g_i^t + c_i^{t+1}$
11:     **end for**
12:     Master computes $g^{t+1} = \frac{1}{n} \sum_{i=1}^{n} g_i^{t+1}$ via $g^{t+1} = g^t + \frac{1}{n} \sum_{i=1}^{n} c_i^{t+1}$
13: **end for**

---

**Assumption 3** (Individual smoothness[21]). *For each $i = 1, \ldots, n$, every realization of $\xi_i \sim \mathcal{D}_i$, the stochastic gradient $\nabla f_i(x, \xi_i)$ is $\ell_i$-Lipschitz, i.e., for all $x, y \in \mathbb{R}^d$*

$$\|\nabla f_i(x, \xi_i) - \nabla f_i(y, \xi_i)\| \leq \ell_i \|x - y\|.$$

*We denote $\widetilde{\ell}^2 := \frac{1}{n} \sum_{i=1}^{n} \ell_i^2$*

**Theorem 7.** *Let Assumptions 1, 2 and 3 hold. Let $\hat{x}^T$ be sampled uniformly at random from the iterates of the method. Let Algorithm 5 run with a contractive compressor. For all $\eta \in (0,1]$ and $B_{\text{init}} \geq 1$, with $\gamma \leq \min\left\{\frac{\alpha}{8\widetilde{L}}, \frac{\sqrt{\alpha}}{6\widetilde{\ell}}, \frac{\sqrt{n\eta}}{8\widetilde{\ell}}\right\}$, we have*

$$\mathbb{E}\left[\|\nabla f(\hat{x}^T)\|^2\right] \leq \mathcal{O}\left(\frac{\Psi_0}{\gamma T} + \frac{\eta^3 \sigma^2}{\alpha^2} + \frac{\eta^2 \sigma^2}{\alpha} + \frac{\eta \sigma^2}{n}\right), \tag{48}$$

*where $\Psi_0 := \delta_0 + \frac{\gamma}{\eta} \mathbb{E}\left[\|v^0 - \nabla f(x^0)\|^2\right] + \frac{\gamma\eta}{\alpha^2} \frac{1}{n} \sum_{i=1}^{n} \mathbb{E}\left[\|v_i^0 - \nabla f_i(x^0)\|^2\right]$. With the following step-size, momentum parameter, and initial batch size*

$$\gamma = \min\left\{\frac{\alpha}{8\widetilde{L}}, \frac{\sqrt{\alpha}}{6\widetilde{\ell}}, \frac{\sqrt{n\eta}}{8\widetilde{\ell}}\right\}, \quad \eta = \min\left\{\alpha, \left(\frac{\widetilde{\ell}\delta_0 \alpha^2}{\sigma^2 \sqrt{n}T}\right)^{2/7}, \left(\frac{\widetilde{\ell}\delta_0 \alpha}{\sigma^2 \sqrt{n}T}\right)^{2/5}, \left(\frac{\widetilde{\ell}\delta_0 \sqrt{n}}{\sigma^2 T}\right)^{2/3}\right\},$$

*and $B_{\text{init}} = \max\left\{\frac{\sigma^2}{L\delta_0 n}, \frac{\alpha n}{T}\right\}$, we have*

$$\mathbb{E}\left[\|\nabla f(\hat{x}^T)\|^2\right] \leq \mathcal{O}\left(\frac{\widetilde{L}\delta_0}{\alpha T} + \frac{\widetilde{\ell}\delta_0}{\sqrt{\alpha}T} + \left(\frac{\widetilde{\ell}\delta_0 \sigma^{1/3}}{\alpha^{1/3}\sqrt{n}T}\right)^{6/7} + \left(\frac{\widetilde{\ell}\delta_0 \sigma^{1/2}}{\alpha^{1/4}\sqrt{n}T}\right)^{4/5} + \left(\frac{\widetilde{\ell}\delta_0 \sigma}{nT}\right)^{2/3}\right).$$

**Corollary 5.** *Under the setting of Theorem 7. we have $\mathbb{E}\left[\|\nabla f(\hat{x}^T)\|\right] \leq \varepsilon$ after $T = \mathcal{O}\left(\frac{\widetilde{\ell}\delta_0}{\alpha\varepsilon^2} + \frac{\widetilde{\ell}\delta_0 \sigma^{1/3}}{\alpha^{1/3}\sqrt{n}\varepsilon^{7/3}} + \frac{\widetilde{\ell}\delta_0 \sigma^{1/2}}{\alpha^{1/4}\sqrt{n}\varepsilon^{5/2}} + \frac{\widetilde{\ell}\delta_0 \sigma}{n\varepsilon^3}\right)$ iterations.*

Recently, Yau and Wai [2022] propose and analyze a DoCoM-SGT algorithm for decentralized optimization with contractive compressor under the above Assumption 3. When their method is specialized to centralized setting (with mixing constant $\rho = 1$), their total sample complexity becomes $\mathcal{O}\left(\frac{\widetilde{\ell}}{\alpha\varepsilon^2} + \frac{n^{4/5}\sigma^{3/2}}{\alpha^{9/4}\varepsilon^{3/2}} + \frac{\sigma^3}{n\varepsilon^3}\right)$ (see Table 1 or Theorem 4.1 in [Yau and Wai, 2022]). In contrast, the sample complexity given in our Corollary 5 improves the dependence on $\sigma$ in the last term and, moreover, achieves the linear speedup in terms of $n$ for all stochastic terms in the sample complexity.

---

[21]This assumption can be also relaxed to so-called expected smoothness.

*Proof.* Similarly to the proof of Theorem 3, we control the error between $g^t$ and $\nabla f(x^t)$ by decomposing it into two terms

$$\left\|g^t - \nabla f(x^t)\right\|^2 \leq 2\left\|g^t - w^t\right\|^2 + 2\left\|w^t - \nabla f(x^t)\right\|^2 \leq 2\frac{1}{n}\sum_{i=1}^n \left\|g_i^t - w_i^t\right\|^2 + 2\left\|w^t - \nabla f(x^t)\right\|^2.$$

In the following, we develop a recursive bound for each term above separately.

**Part I. Controlling the variance of STORM/MVR estimator for each $w_i^t$ and on average $w^t$.**
Recall that by Lemma 7-(55), we have for each $i = 1, \ldots, n$, and any $0 < \eta \leq 1$ and $t \geq 0$

$$\mathbb{E}\left[\left\|w_i^{t+1} - \nabla f_i(x^{t+1})\right\|^2\right] \leq (1-\eta)\mathbb{E}\left[\left\|w_i^t - \nabla f_i(x^t)\right\|^2\right] + 2\ell_i^2 \mathbb{E}\left[\left\|x^t - x^{t+1}\right\|^2\right] + 2\eta^2\sigma^2. \quad (49)$$

Averaging inequalities (49) over $i = 1, \ldots, n$ and denoting $\widetilde{P}_t := \frac{1}{n}\sum_{i=1}^n \mathbb{E}\left[\left\|w_i^t - \nabla f_i(x^t)\right\|^2\right]$,
$R_t := \mathbb{E}\left[\left\|x^t - x^{t+1}\right\|^2\right]$ we have

$$\widetilde{P}_{t+1} \leq (1-\eta)\widetilde{P}_t + 2\widetilde{\ell}^2 R_t + 2\eta^2\sigma^2.$$

Summing up the above inequality for $t = 0, \ldots, T-1$, we derive

$$\frac{1}{T}\sum_{t=0}^{T-1} \widetilde{P}_t \leq \frac{2\widetilde{\ell}^2}{\eta}\frac{1}{T}\sum_{t=0}^{T-1} R_t + 2\eta\sigma^2 + \frac{1}{\eta T}\widetilde{P}_0. \quad (50)$$

Similarly by Lemma 7-(56) denoting $P_t := \mathbb{E}\left[\left\|w^t - \nabla f(x^t)\right\|^2\right]$, we have

$$\frac{1}{T}\sum_{t=0}^{T-1} P_t \leq \frac{2\widetilde{\ell}^2}{\eta n}\frac{1}{T}\sum_{t=0}^{T-1} R_t + \frac{2\eta\sigma^2}{n} + \frac{1}{\eta T}P_0. \quad (51)$$

**Part II. Controlling the variance of contractive compressor and STORM/MVR estimator.** By
Lemma 8 we have for each $i = 1, \ldots, n$, and any $0 < \eta \leq 1$ and $t \geq 0$

$$\begin{aligned}
\mathbb{E}\left[\left\|g_i^{t+1} - w_i^{t+1}\right\|^2\right] &\leq \left(1 - \frac{\alpha}{2}\right)\mathbb{E}\left[\left\|g_i^t - w_i^t\right\|^2\right] + \frac{4\eta^2}{\alpha}\mathbb{E}\left[\left\|w_i^t - \nabla f_i(x^t)\right\|^2\right] \\
&\quad + \left(\frac{4L_i^2}{\alpha} + \ell_i^2\right)\mathbb{E}\left[\left\|x^{t+1} - x^t\right\|^2\right] + 2\eta^2\sigma^2.
\end{aligned} \quad (52)$$

Averaging inequalities (52) over $i = 1, \ldots, n$, denoting $\widetilde{V}_t := \frac{1}{n}\sum_{i=1}^n \mathbb{E}\left[\left\|g_i^t - w_i^t\right\|^2\right]$, and summing
up the resulting inequality for $t = 0, \ldots, T-1$, we obtain

$$\begin{aligned}
\frac{1}{T}\sum_{t=0}^{T-1} \widetilde{V}_t &\leq \frac{8\eta^2}{\alpha^2}\frac{1}{T}\sum_{t=0}^{T-1} \widetilde{P}_t + \left(\frac{8\widetilde{L}^2}{\alpha^2} + \frac{2\widetilde{\ell}^2}{\alpha}\right)\frac{1}{T}\sum_{t=0}^{T-1} R_t + \frac{2\eta^2\sigma^2}{\alpha} \\
&\leq \left(\frac{8\widetilde{L}^2}{\alpha^2} + \frac{2\widetilde{\ell}^2}{\alpha} + \frac{16\eta\widetilde{\ell}^2}{\alpha^2}\right)\frac{1}{T}\sum_{t=0}^{T-1} R_t \\
&\quad + \frac{16\eta^3\sigma^2}{\alpha^2} + \frac{2\eta^2\sigma^2}{\alpha} + \frac{8\eta}{\alpha^2 T}\widetilde{P}_0.
\end{aligned} \quad (53)$$

**Part III. Combining steps I and II with descent lemma.** By smoothness (Assumption 1) of $f(\cdot)$ it
follows from Lemma 1 that for any $\gamma \leq 1/(2L)$ we have

$$\begin{aligned}
f(x^{t+1}) &\leq f(x^t) - \frac{\gamma}{2}\left\|\nabla f(x^t)\right\|^2 - \frac{1}{4\gamma}\left\|x^{t+1} - x^t\right\|^2 + \frac{\gamma}{2}\left\|g^t - \nabla f(x^t)\right\|^2 \quad (54) \\
&\leq f(x^t) - \frac{\gamma}{2}\left\|\nabla f(x^t)\right\|^2 - \frac{1}{4\gamma}\left\|x^{t+1} - x^t\right\|^2 + \gamma\frac{1}{n}\sum_{i=1}^n \left\|g_i^t - w_i^t\right\|^2 + \gamma\left\|w^t - \nabla f(x^t)\right\|^2.
\end{aligned}$$

Subtracting $f^*$ from both sides of (54), taking expectation and defining $\delta_t := \mathbb{E}\left[f(x^t) - f^*\right]$, we derive

$$
\begin{aligned}
\mathbb{E}\left[\left\|\nabla f(\hat{x}^T)\right\|^2\right] &= \frac{1}{T}\sum_{t=0}^{T-1}\mathbb{E}\left[\left\|\nabla f(x^t)\right\|^2\right] \\
&\leq \frac{2\delta_0}{\gamma T} + 2\frac{1}{T}\sum_{t=0}^{T-1}\widetilde{V}_t + 2\frac{1}{T}\sum_{t=0}^{T-1}P_t - \frac{1}{2\gamma^2}\frac{1}{T}\sum_{t=0}^{T-1}R_t \\
&\overset{(i)}{\leq} \frac{2\delta_0}{\gamma T} + \left(\frac{16\widetilde{L}^2}{\alpha^2} + \frac{4\widetilde{\ell}^2}{\alpha} + \frac{32\eta\widetilde{\ell}^2}{\alpha^2}\right)\frac{1}{T}\sum_{t=0}^{T-1}R_t + 2\frac{1}{T}\sum_{t=0}^{T-1}P_t - \frac{1}{2\gamma^2}\frac{1}{T}\sum_{t=0}^{T-1}R_t \\
&\quad + \frac{32\eta^3\sigma^2}{\alpha^2} + \frac{4\eta^2\sigma^2}{\alpha} + \frac{16\eta}{\alpha^2 T}\widetilde{P}_0 \\
&\overset{(ii)}{\leq} \frac{2\delta_0}{\gamma T} + \left(\frac{16\widetilde{L}^2}{\alpha^2} + \frac{4\widetilde{\ell}^2}{\alpha} + \frac{32\eta\widetilde{\ell}^2}{\alpha^2} + \frac{4\widetilde{\ell}^2}{\eta n} - \frac{1}{2\gamma^2}\right)\frac{1}{T}\sum_{t=0}^{T-1}R_t \\
&\quad + \frac{32\eta^3\sigma^2}{\alpha^2} + \frac{4\eta^2\sigma^2}{\alpha} + \frac{4\eta\sigma^2}{n} + \frac{16\eta}{\alpha^2 T}\widetilde{P}_0 + \frac{2}{\eta T}P_0 \\
&\leq \frac{2\delta_0}{\gamma T} + \frac{32\eta^3\sigma^2}{\alpha^2} + \frac{4\eta^2\sigma^2}{\alpha} + \frac{4\eta\sigma^2}{n} + \frac{16\eta}{\alpha^2 T}\widetilde{P}_0 + \frac{2}{\eta T}P_0,
\end{aligned}
$$

where in $(i)$ we apply (53), in $(ii)$ we use (51), and the last step follows by assumption on the step-size, which proves (48).

We now find the particular values of parameters. Using $w_i^0 = \frac{1}{B_{\text{init}}}\sum_{j=1}^{B_{\text{init}}}\nabla f_i(x^0, \xi_{i,j}^0)$ for all $i = 1, \dots, n$, we have

$$
P_0 = \mathbb{E}\left[\left\|w^0 - \nabla f(x^0)\right\|^2\right] \leq \frac{\sigma^2}{nB_{\text{init}}} \text{ and } \widetilde{P}_0 = \frac{1}{n}\sum_{i=1}^{n}\mathbb{E}\left[\left\|w_i^0 - \nabla f_i(x^0)\right\|^2\right] \leq \frac{\sigma^2}{B_{\text{init}}}.
$$

We can substitute the choice of $\gamma$ and obtain

$$
\begin{aligned}
\mathbb{E}\left[\left\|\nabla f(\hat{x}^T)\right\|^2\right] &= \mathcal{O}\left(\frac{\delta_0}{\gamma T} + \frac{\eta^3\sigma^2}{\alpha^2} + \frac{\eta^2\sigma^2}{\alpha} + \frac{\eta\sigma^2}{n} + \frac{\sigma^2}{\eta n B_{\text{init}} T} + \frac{\eta\sigma^2}{\alpha^2 B_{\text{init}} T}\right) \\
&= \mathcal{O}\left(\frac{\widetilde{L}\delta_0}{\alpha T} + \frac{\widetilde{\ell}\delta_0}{\sqrt{\alpha}T} + \frac{\widetilde{\ell}\delta_0}{\sqrt{n\eta}T} + \frac{\eta^3\sigma^2}{\alpha^2} + \frac{\eta^2\sigma^2}{\alpha} + \frac{\eta\sigma^2}{n} + \frac{\sigma^2}{\eta n B_{\text{init}} T} + \frac{\eta\sigma^2}{\alpha^2 B_{\text{init}} T}\right).
\end{aligned}
$$

Since $B_{\text{init}} \geq \frac{\sigma^2}{L\delta_0 n}$, we have

$$
\mathbb{E}\left[\left\|\nabla f(\hat{x}^T)\right\|^2\right] = \mathcal{O}\left(\frac{\widetilde{L}\delta_0}{\alpha T} + \frac{\widetilde{\ell}\delta_0}{\sqrt{\alpha}T} + \frac{\widetilde{\ell}\delta_0}{\sqrt{n\eta}T} + \frac{\eta^3\sigma^2}{\alpha^2} + \frac{\eta^2\sigma^2}{\alpha} + \frac{\eta\sigma^2}{n} + \frac{\eta\sigma^2}{\alpha^2 B_{\text{init}} T}\right).
$$

Notice that the choice of the momentum parameter such that $\eta \leq \left(\frac{\widetilde{\ell}\delta_0\alpha^2}{\sigma^2\sqrt{n}T}\right)^{2/7}$, $\eta \leq \left(\frac{\widetilde{\ell}\delta_0\alpha}{\sigma^2\sqrt{n}T}\right)^{2/5}$, $\eta \leq \left(\frac{\widetilde{\ell}\delta_0\sqrt{n}}{\sigma^2 T}\right)^{2/3}$, and $\eta \leq \left(\frac{\widetilde{\ell}\delta_0\alpha^2 B_{\text{init}}}{\sigma^2\sqrt{n}}\right)^{2/3}$ ensures that $\frac{\eta^3\sigma^2}{\alpha^2} \leq \frac{\widetilde{\ell}\delta_0}{\sqrt{n\eta}T}$, $\frac{\eta^2\sigma^2}{\alpha} \leq \frac{\widetilde{\ell}\delta_0}{\sqrt{n\eta}T}$, $\frac{\eta\sigma^2}{n} \leq \frac{\widetilde{\ell}\delta_0}{\sqrt{n\eta}T}$, and $\frac{\eta\sigma^2}{\alpha^2 B_{\text{init}} T} \leq \frac{\widetilde{\ell}\delta_0}{\sqrt{n\eta}T}$. Therefore, we have

$$
\mathbb{E}\left[\left\|\nabla f(\hat{x}^T)\right\|^2\right] = \mathcal{O}\left(\frac{\widetilde{L}\delta_0}{\alpha T} + \frac{\widetilde{\ell}\delta_0}{\sqrt{\alpha}T} + \left(\frac{\widetilde{\ell}\delta_0\sigma^{1/3}}{\alpha^{1/3}\sqrt{n}T}\right)^{6/7} + \left(\frac{\widetilde{\ell}\delta_0\sigma^{1/2}}{\alpha^{1/4}\sqrt{n}T}\right)^{4/5} + \left(\frac{\widetilde{\ell}\delta_0\sigma}{nT}\right)^{2/3} + \left(\frac{\widetilde{\ell}\delta_0\sigma}{\sqrt{n}}\right)^{2/3}\frac{\alpha^{1/3}}{B_{\text{init}}^{1/3}T}\right).
$$

Using $B_{\text{init}} \geq \frac{\alpha n}{T}$, we obtain

$$
\mathbb{E}\left[\left\|\nabla f(\hat{x}^T)\right\|^2\right] = \mathcal{O}\left(\frac{\widetilde{L}\delta_0}{\alpha T} + \frac{\widetilde{\ell}\delta_0}{\sqrt{\alpha}T} + \left(\frac{\widetilde{\ell}\delta_0\sigma^{1/3}}{\alpha^{1/3}\sqrt{n}T}\right)^{6/7} + \left(\frac{\widetilde{\ell}\delta_0\sigma^{1/2}}{\alpha^{1/4}\sqrt{n}T}\right)^{4/5} + \left(\frac{\widetilde{\ell}\delta_0\sigma}{nT}\right)^{2/3}\right).
$$

$\square$

## I.1  Controlling the variance of STORM/MVR estimator

**Lemma 7.** *Let Assumptions 2 and 3 be satisfied, and suppose $0 < \eta \leq 1$. For every $i = 1, \ldots, n$, let the sequence $\{w_i^t\}_{t \geq 0}$ be updated via $w_i^{t+1} = \nabla f_i(x^{t+1}, \xi_i^{t+1}) + (1 - \eta)(w_i^t - \nabla f_i(x^t, \xi_i^{t+1}))$. Define the sequence $w^t := \frac{1}{n} \sum_{i=1}^n w_i^t$. Then for every $i = 1, \ldots, n$ and $t \geq 0$ it holds*

$$\mathbb{E}\left[\left\|w_i^{t+1} - \nabla f_i(x^{t+1})\right\|^2\right] \leq (1 - \eta)\mathbb{E}\left[\left\|w_i^t - \nabla f_i(x^t)\right\|^2\right] + 2\ell_i^2\mathbb{E}\left[\left\|x^t - x^{t+1}\right\|^2\right] + 2\eta^2\sigma^2, \quad (55)$$

$$\mathbb{E}\left[\left\|w^{t+1} - \nabla f(x^{t+1})\right\|^2\right] \leq (1 - \eta)\mathbb{E}\left[\left\|w^t - \nabla f(x^t)\right\|^2\right] + \frac{2\widetilde{\ell}^2}{n}\mathbb{E}\left[\left\|x^t - x^{t+1}\right\|^2\right] + \frac{2\eta^2\sigma^2}{n}. \quad (56)$$

*Proof.* For each $t = 0, \ldots, T - 1$, define a random vector $\xi^t := (\xi_1^t, \ldots, \xi_n^t)$ and denote by $\nabla f(x^t, \xi^t) := \frac{1}{n} \sum_{i=1}^n \nabla f_i(x^t, \xi_i^t)$. Note that the entries of the random vector $\xi^t$ are independent and $\mathbb{E}_{\xi^t}[\nabla f(x^t, \xi^t)] = \nabla f(x^t)$, then we have

$$w^{t+1} = \nabla f(x^{t+1}, \xi^{t+1}) + (1 - \eta)\left(w^t - \nabla f(x^t, \xi^{t+1})\right),$$

where $w^t = \frac{1}{n} \sum_{i=1}^n w_i^t$ is an auxiliary sequence.

We define

$$\mathcal{V}_i^t := \nabla f_i(x^t, \xi_i^t) - \nabla f_i(x^t), \qquad \mathcal{V}^t := \frac{1}{n} \sum_{i=1}^n \mathcal{V}_i^t,$$

$$\mathcal{W}_i^t := \nabla f_i(x^t) - \nabla f_i(x^t, \xi_i^{t+1}) + \nabla f_i(x^{t+1}, \xi_i^{t+1}) - \nabla f_i(x^{t+1}), \qquad \mathcal{W}^t := \frac{1}{n} \sum_{i=1}^n \mathcal{W}_i^t.$$

Then by Assumptions 2, we have

$$\mathbb{E}\left[\mathcal{V}_i^t\right] = \mathbb{E}\left[\mathcal{W}_i^t\right] = \mathbb{E}\left[\mathcal{V}^t\right] = \mathbb{E}\left[\mathcal{W}^t\right] = 0, \quad (57)$$

$$\mathbb{E}\left[\left\|\mathcal{V}_i^t\right\|^2\right] \leq \sigma^2, \qquad \mathbb{E}\left[\left\|\mathcal{V}^t\right\|^2\right] \leq \frac{\sigma^2}{n}. \quad (58)$$

Furthermore, we can derive

$$
\begin{aligned}
\mathbb{E}\left[\left\|\mathcal{W}^t\right\|^2\right] &= \mathbb{E}\left[\left\|\frac{1}{n} \sum_{i=1}^n \mathcal{W}_i^t\right\|^2\right] \\
&= \frac{1}{n^2}\mathbb{E}\left[\left\|\sum_{i=1}^n \mathcal{W}_i^t\right\|^2\right] \\
&= \frac{1}{n^2} \sum_{i=1}^n \mathbb{E}\left[\left\|\mathcal{W}_i^t\right\|^2\right] + \frac{1}{n^2} \sum_{i \neq j} \mathbb{E}\left[\langle\mathcal{W}_i^t, \mathcal{W}_j^t\rangle\right] \\
&\overset{(i)}{=} \frac{1}{n^2} \sum_{i=1}^n \mathbb{E}\left[\left\|\mathcal{W}_i^t\right\|^2\right] + \frac{1}{n^2} \sum_{i \neq j} \langle\mathbb{E}\left[\mathcal{W}_i^t\right], \mathbb{E}\left[\mathcal{W}_j^t\right]\rangle \\
&= \frac{1}{n^2} \sum_{i=1}^n \mathbb{E}\left[\left\|\mathcal{W}_i^t\right\|^2\right] \\
&\leq \frac{1}{n^2} \sum_{i=1}^n \mathbb{E}\left[\left\|\nabla f_i(x^{t+1}, \xi_i^{t+1}) - \nabla f_i(x^t, \xi_i^{t+1})\right\|^2\right] \\
&\leq \frac{1}{n^2} \sum_{i=1}^n \ell_i^2\mathbb{E}\left[\left\|x^{t+1} - x^t\right\|^2\right] = \frac{\widetilde{\ell}^2}{n}\mathbb{E}\left[\left\|x^{t+1} - x^t\right\|^2\right],
\end{aligned}
$$

where $(i)$ holds by the conditional independence of $\mathcal{W}_i^t$ and $\mathcal{W}_j^t$, and the last inequality follows by the individual smoothness of stochastic functions (Assumption 3). Therefore, we have

$$\mathbb{E}\left[\left\|\mathcal{W}_i^t\right\|^2\right] \leq \ell_i^2 \mathbb{E}\left[\left\|x^{t+1} - x^t\right\|^2\right], \qquad \mathbb{E}\left[\left\|\mathcal{W}^t\right\|^2\right] \leq \frac{\widetilde{\ell}^2}{n}\mathbb{E}\left[\left\|x^{t+1} - x^t\right\|^2\right], \qquad (59)$$

where the first inequality is obtained by using a similar derivation.

By the update rule for $w^t$, we can also derive

$$
\begin{aligned}
w^{t+1} - \nabla f(x^{t+1}) &= (1-\eta)\left(w^t - \nabla f(x^t, \xi^{t+1})\right) + \left(\nabla f(x^{t+1}, \xi^{t+1}) - \nabla f(x^{t+1})\right) \\
&= (1-\eta)\left(w^t - \nabla f(x^t)\right) + \eta\left(\nabla f(x^{t+1}, \xi^{t+1}) - \nabla f(x^{t+1})\right) \\
&\quad + (1-\eta)\left(\left(\nabla f(x^t) - \nabla f(x^t, \xi^{t+1}) + \nabla f(x^{t+1}, \xi^{t+1}) - \nabla f(x^{t+1})\right)\right) \\
&= (1-\eta)\left(w^t - \nabla f(x^t)\right) + \eta\mathcal{V}^{t+1} + (1-\eta)\mathcal{W}^t.
\end{aligned}
$$

Therefore, we have

$$
\begin{aligned}
\mathbb{E}\left[\left\|w^{t+1} - \nabla f(x^{t+1})\right\|^2\right] &\leq \mathbb{E}\left[\mathbb{E}_{\xi^{t+1}}\left[\left\|(1-\eta)\left(w^t - \nabla f(x^t)\right) + \eta\mathcal{V}_{t+1} + (1-\eta)\mathcal{W}_t\right\|^2\right]\right] \\
&= (1-\eta)^2 \mathbb{E}\left[\left\|w^t - \nabla f(x^t)\right\|^2\right] + \mathbb{E}\left[\left\|\eta\mathcal{V}^{t+1} + (1-\eta)\mathcal{W}^t\right\|^2\right] \\
&\leq (1-\eta)\left\|w^t - \nabla f(x^t)\right\|^2 + 2\eta^2 \mathbb{E}\left[\left\|\mathcal{V}^{t+1}\right\|^2\right] + 2\mathbb{E}\left[\left\|\mathcal{W}^t\right\|^2\right] \\
&\leq (1-\eta)\mathbb{E}\left[\left\|w^t - \nabla f(x^t)\right\|^2\right] + \frac{2\sigma^2\eta^2}{n} + \frac{2\widetilde{\ell}^2}{n}\mathbb{E}\left[\left\|x^{t+1} - x^t\right\|^2\right],
\end{aligned}
$$

where the last inequality holds by (58) and (59). Similarly for each $i = 1, \ldots, n$, we have

$$w_i^{t+1} - \nabla f_i(x^{t+1}) = (1-\eta)\left(w_i^t - \nabla f_i(x^t)\right) + \eta\mathcal{V}_i^{t+1} + (1-\eta)\mathcal{W}_i^t. \qquad (60)$$

Thus,

$$\mathbb{E}\left[\left\|w_i^{t+1} - \nabla f_i(x^{t+1})\right\|^2\right] \leq (1-\eta)\mathbb{E}\left[\left\|w_i^t - \nabla f_i(x^t)\right\|^2\right] + 2\sigma^2\eta^2 + 2\ell_i^2 R_t.$$

$\square$

## I.2 Controlling the variance of contractive compression and STORM/MVR estimator

**Lemma 8.** *Let Assumptions 1, 2 and 3 be satisfied, and suppose $0 < \eta \leq 1$. For every $i = 1, \ldots, n$, let the sequences $\{w_i^t\}_{t\geq 0}$ and $\{g_i^t\}_{t\geq 0}$ be updated via*

$$
\begin{aligned}
w_i^{t+1} &= \nabla f_i(x^{t+1}, \xi_i^{t+1}) + (1-\eta)(w_i^t - \nabla f_i(x^t, \xi_i^{t+1})), \\
g_i^{t+1} &= g_i^t + \mathcal{C}\left(w_i^{t+1} - g_i^t\right).
\end{aligned}
$$

*Then for every $i = 1, \ldots, n$ and $t \geq 0$ it holds*

$$
\begin{aligned}
\mathbb{E}\left[\left\|g_i^{t+1} - w_i^{t+1}\right\|^2\right] &\leq \left(1 - \frac{\alpha}{2}\right)\mathbb{E}\left[\left\|g_i^t - w_i^t\right\|^2\right] + \frac{4\eta^2}{\alpha}\mathbb{E}\left[\left\|w_i^t - \nabla f_i(x^t)\right\|^2\right] \\
&\quad + \left(\frac{4L_i^2}{\alpha} + \ell_i^2\right)\mathbb{E}\left[\left\|x^{t+1} - x^t\right\|^2\right] + 2\eta^2\sigma^2. \qquad (61)
\end{aligned}
$$

*Proof.* By the update rule of $w_i^t$, $g_i^t$, and definition of $\mathcal{V}_i^t$, $\mathcal{W}_i^t$ given in the proof of Lemma 7, we can derive

$$
\begin{aligned}
\mathbb{E}\left[\left\|g_i^{t+1} - w_i^{t+1}\right\|^2\right] &= \mathbb{E}\left[\left\|\mathcal{C}(w_i^{t+1} - g_i^t) - (w_i^{t+1} - g_i^t)\right\|^2\right] \\
&\overset{(i)}{\leq} (1-\alpha)\mathbb{E}\left[\left\|w_i^{t+1} - g_i^t\right\|^2\right] \\
&\overset{(ii)}{=} (1-\alpha)\mathbb{E}\left[\left\|(1-\eta)\left(w_i^t - \nabla f_i(x^t)\right) + \eta\mathcal{V}_i^{t+1} + (1-\eta)\mathcal{W}_i^t + \nabla f_i(x^{t+1}) - g_i^t\right\|^2\right] \\
&= (1-\alpha)\mathbb{E}\left[\mathbb{E}_{\xi_i^{t+1}}\left[\left\|(1-\eta)\left(w_i^t - \nabla f_i(x^t)\right) + \eta\mathcal{V}_i^{t+1} + (1-\eta)\mathcal{W}_i^t + \nabla f_i(x^{t+1}) - g_i^t\right\|^2\right]\right]
\end{aligned}
$$

$$\overset{(iii)}{=} \quad (1-\alpha)\mathbb{E}\left[\left\|(1-\eta)\left(w_i^t - \nabla f_i(x^t)\right) + \nabla f_i(x^{t+1}) - g_i^t\right\|^2\right]$$

$$+(1-\alpha)\mathbb{E}\left[\left\|\eta\mathcal{V}_i^{t+1} + (1-\eta)\mathcal{W}_i^t\right\|^2\right]$$

$$= \quad (1-\alpha)\mathbb{E}\left[\left\|\left(w_i^t - g_i^t\right) + \left(\nabla f_i(x^{t+1}) - \nabla f_i(x^t)\right) - \eta\left(w_i^t - \nabla f_i(x^t)\right)\right\|^2\right]$$

$$+(1-\alpha)\mathbb{E}\left[\left\|\eta\mathcal{V}_i^{t+1} + (1-\eta)\mathcal{W}_i^t\right\|^2\right]$$

$$\overset{(iv)}{\leq} \quad (1-\alpha)(1+\rho)\,\mathbb{E}\left[\left\|w_i^t - g_i^t\right\|^2\right]$$

$$+(1-\alpha)\left(1+\rho^{-1}\right)\mathbb{E}\left[\left\|\left(\nabla f_i(x^{t+1}) - \nabla f_i(x^t)\right) - \eta\left(w_i^t - \nabla f_i(x^t)\right)\right\|^2\right]$$

$$+2(1-\alpha)\eta^2\mathbb{E}\left[\left\|\mathcal{V}_i^{t+1}\right\|^2\right] + 2(1-\alpha)(1-\eta)^2\mathbb{E}\left[\left\|\mathcal{W}_i^t\right\|^2\right]$$

$$\overset{(v)}{=} \quad (1-\theta)\mathbb{E}\left[\left\|w_i^t - g_i^t\right\|^2\right]$$

$$+\beta\mathbb{E}\left[\left\|\left(\nabla f_i(x^{t+1}) - \nabla f_i(x^t)\right) - \eta\left(w_i^t - \nabla f_i(x^t)\right)\right\|^2\right]$$

$$+2(1-\alpha)\eta^2\mathbb{E}\left[\left\|\mathcal{V}_i^{t+1}\right\|^2\right] + 2(1-\alpha)(1-\eta)^2\mathbb{E}\left[\left\|\mathcal{W}_i^t\right\|^2\right]$$

$$\overset{(vi)}{\leq} \quad (1-\theta)\mathbb{E}\left[\left\|w_i^t - g_i^t\right\|^2\right] + 2\beta\eta^2\mathbb{E}\left[\left\|w_i^t - \nabla f_i(x^t)\right\|^2\right]$$

$$+2\beta\mathbb{E}\left[\left\|\nabla f_i(x^{t+1}) - \nabla f_i(x^t)\right\|^2\right] + 2\ell_i^2\mathbb{E}\left[\left\|x^{t+1} - x^t\right\|^2\right] + 2\eta^2\sigma^2$$

$$\leq \quad (1-\theta)\mathbb{E}\left[\left\|w_i^t - g_i^t\right\|^2\right] + 2\beta\eta^2\mathbb{E}\left[\left\|w_i^t - \nabla f_i(x^t)\right\|^2\right]$$

$$+\left(2\beta L_i^2 + \ell_i^2\right)\mathbb{E}\left[\left\|x^{t+1} - x^t\right\|^2\right] + 2\eta^2\sigma^2,$$

where $(i)$ holds by Definition 1, $(ii)$ follows from (60), $(iii)$ holds by unbiasedness of $\mathcal{V}_i^{t+1}$ and $\mathcal{W}_i^t$ (57). In $(iv)$ we use Young's inequality twice, in $(v)$ we introduce the notation $\theta := 1-(1-\alpha)(1+\rho)$ and $\beta := (1-\alpha)(1+\rho^{-1})$, in $(vi)$ we again use Young's inequality and the bound (58) and (59). The last step holds by smoothness of $f_i(\cdot)$ (Assumption 1). The proof is complete by the choice $\rho = \alpha/2$, which guarantees $1-\theta \leq 1-\alpha/2$, and $2\beta \leq 4/\alpha$. $\qquad\square$

## J Simplified Proof of SGDM: Time Varying Parameters and No Tuning for Momentum Sequence

In this section, we give a simplified proof of SGDM in the single node setting ($n = 1$) without compression ($\alpha = 1$). The following theorem shows that the momentum parameter can be chosen in a parameter agnostic[22] way as $\eta_t = 1/\sqrt{t+1}$ (or $\eta_t = 1/\sqrt{T+1}$), instead of being a constant depending on problem parameters as it is suggested in our main Theorem 3. In other words, using SGDM with time varying momentum does not introduce any additional tuning of hyper-parameters.

**Theorem 8.** *Let Assumptions 1, 2 hold. Let $n = 1$ and Algorithm 1 run with identity compressor $\mathcal{C}$, i.e., $\alpha = 1$, and (possibly) time varying momentum $\eta_t \in (0, 1]$ and step-size paramters $\gamma_t = \gamma\eta_t$ with $\gamma \in (0, 1/(3L)]$. Let $\hat{x}^T$ be sampled from the iterates of the algorithm with probabilities $p_t = \eta_t/(\sum_{t=0}^{T-1} \eta_t)$, then*

$$\mathbb{E}\left[\left\|\nabla f(\hat{x}^T)\right\|^2\right] \leq \frac{2\Lambda_0\gamma^{-1} + 2\sigma^2\sum_{t=0}^{T-1}\eta_t^2}{\sum_{t=0}^{T-1}\eta_t},$$

*where $\Lambda_0 := f(x^0) - f^* + \gamma\mathbb{E}\left[\left\|v^0 - \nabla f(x^0)\right\|^2\right]$ is the Lyapunov function.*

*Proof.* By Lemma 2 denoting $P_t := \mathbb{E}\left[\left\|v^t - \nabla f(x^t)\right\|^2\right]$, $R_t := \mathbb{E}\left[\left\|x^t - x^{t+1}\right\|^2\right]$, we have

$$P_{t+1} \leq P_t - \eta_t P_t + \frac{3L^2}{\eta_t}R_t + \eta_t^2\sigma^2. \tag{62}$$

By descent Lemma 1, we have for any $\gamma_t > 0$

$$\delta_{t+1} \leq \delta_t - \frac{\gamma_t}{2}\mathbb{E}\left[\left\|\nabla f(x^t)\right\|^2\right] - \frac{1}{2\gamma_t}\left(1 - \gamma_t L\right)R_t + \frac{\gamma_t}{2}P_t, \tag{63}$$

where $\delta_0 := \mathbb{E}\left[f(x^t) - f^*\right]$. Define the Lyapunov function as $\Lambda_t = \delta_0 + \gamma P_t$. Then summing up (63) with a $\gamma$ multiple of (62) and noticing that $\gamma_t \leq \gamma$, we get

$$\Lambda_{t+1} \leq \Lambda_t - \frac{\gamma_t}{2}\mathbb{E}\left[\left\|\nabla f(x^t)\right\|^2\right] - \frac{1}{2\gamma_t}\left(1 - \gamma L - 6\gamma^2 L^2\right)R_t + \gamma\eta_t^2\sigma^2.$$

Since $\gamma \leq 1/(3L)$, we have $1 - \gamma L - 6\gamma^2 L^2 \leq 0$, and, therefore, by telescoping we can derive

$$\begin{aligned}
\mathbb{E}\left[\left\|\nabla f(\hat{x}^T)\right\|^2\right] &= \left(\sum_{t=0}^{T-1}\eta_t\right)^{-1}\sum_{t=0}^{T-1}\eta_t\mathbb{E}\left[\left\|\nabla f(x^t)\right\|^2\right] \\
&\leq \frac{2\Lambda_0\gamma^{-1} + 2\sigma^2\sum_{t=0}^{T-1}\eta_t^2}{\sum_{t=0}^{T-1}\eta_t}.
\end{aligned}$$

$\square$

The above theorem suggests that to ensure convergence, we can select any momentum sequence such that $\sigma^2\sum_{t=0}^{\infty}\eta_t^2 < \infty$, and $\sum_{t=0}^{\infty}\eta_t^2 \to \infty$ for $t \to \infty$. The parameter $\gamma$, which determines the step-size $\gamma_t = \gamma\eta_t$, should be set to $\gamma = 1/(3L)$ (to minimize the upper bound). Let us now consider some special cases.

**Deterministic case.** If $\sigma = 0$, we can set it to be any constant $\eta_t = \eta \in (0, 1]$ and derive

$$\mathbb{E}\left[\left\|\nabla f(\hat{x}^T)\right\|^2\right] \leq \frac{2\delta_0}{\gamma\eta T} = \mathcal{O}\left(\frac{L\delta_0}{\eta T}\right).$$

---

[22]That is, independently of the problem specific parameters

**Stochastic case.** For $\sigma^2 > 0$, we can select time-varying $\eta_t = \frac{1}{\sqrt{t+1}}$ or constant $\eta_t = \frac{1}{\sqrt{T+1}}$, which gives $\sum_{t=0}^{T-1} \eta_t^2 = \mathcal{O}\left(\log(T)\right)$, and $\sum_{t=0}^{T-1} \eta_t = \Omega\left(\sqrt{T}\right)$. Thus

$$\mathbb{E}\left[\left\|\nabla f(\hat{x}^T)\right\|^2\right] = \tilde{\mathcal{O}}\left(\frac{L\Lambda_0 + \sigma^2}{\sqrt{T}}\right).$$

Notice that if we set $\eta_t$ as above, we do not need any tuning of momentum parameter. Only tuning of paramter $\gamma$ is required to ensure convergence with optimal dependence on $T$, as in SGD without momentum. Of course, this rate is not yet optimal in other parameters, e.g., $\sigma^2$ and $L$. To make it optimal in all problem parameters, we can set $\eta = \max\left\{1, \left(\frac{L\Lambda_0}{\sigma^2 T}\right)^{1/2}\right\}$ similarly to the statement of Theorem 2.

# K   Revisiting EF14-SGD Analysis under BG and BGS Assumptions

In this section, we revisit the analysis of the original variant of error feedback (EF14-SGD) to showcase the difficulty in avoiding BG/BGS assumptions commonly used in the nonconvex analysis of this variant. In summary, the key reason for BG/BGS assumption is to bound the second term in (67) or (68).

Recall that EF14-SGD has the update rule [Stich et al., 2018]

$$x^{t+1} = x^t - g^t, \qquad g^t = \frac{1}{n} \sum_{i=1}^{n} g_i^t, \tag{64}$$

$$\text{EF14-SGD:} \qquad \begin{aligned} e_i^{t+1} &= e_i^t + \gamma \nabla f_i(x^t, \xi_i^t) - g_i^t, \\ g_i^{t+1} &= \mathcal{C}\left(e_i^{t+1} + \gamma \nabla f_i(x^{t+1}, \xi_i^{t+1})\right), \end{aligned} \tag{65}$$

where $\{e_i^t\}_{t \geq 0}$ are error/memory sequences with $e_i^0 = 0$ for each $i = 1, \ldots, n$. Let $e^t := \frac{1}{n} \sum_{i=1}^{n} e_i^t$. The proof of this method relies on so called perturbed iterate analysis, for which one defines a "virtual sequence": $\tilde{x}^t := x^t - e^t$. Then it is verified by direct substitution that for any $t \geq 0$

$$\tilde{x}^{t+1} = \tilde{x}^t - \gamma \frac{1}{n} \sum_{i=1}^{n} \nabla f_i(x^t, \xi_i^t).$$

If follows from Lemma 9 in [Stich and Karimireddy, 2021] that for any $\gamma \leq 1/2L$ and $t \geq 0$

$$\mathbb{E}\left[f(\tilde{x}^{t+1})\right] \leq \mathbb{E}\left[f(\tilde{x}^t)\right] - \frac{\gamma}{4} \mathbb{E}\left[\|\nabla f(x^t)\|^2\right] + \frac{\gamma L \sigma^2}{2n} + \frac{L^2}{2} \mathbb{E}\left[\|e^t\|^2\right].$$

Telescoping the recursion above and setting $\delta_0 := f(x^0) - f^*$, we have

$$\frac{1}{T} \sum_{t=0}^{T-1} \mathbb{E}\left[\|\nabla f(x^t)\|^2\right] \leq \frac{4\delta_0}{\gamma T} + \frac{2\gamma L \sigma^2}{n} + 2L^2 \frac{1}{T} \sum_{t=0}^{T-1} \mathbb{E}\left[\|e^t\|^2\right]. \tag{66}$$

Now it remains to bound efficiently the average error term $\mathbb{E}\left[\|e^t\|^2\right] = \mathbb{E}\left[\|\frac{1}{n} \sum_{i=1}^{n} e_i^t\|^2\right]$. By Jensen's inequality, we have

$$\mathbb{E}\left[\left\|\frac{1}{n} \sum_{i=1}^{n} e_i^t\right\|^2\right] \leq \frac{1}{n} \sum_{i=1}^{n} \mathbb{E}\left[\|e_i^t\|^2\right],$$

and develop a bound for each $\mathbb{E}\left[\|e_i^t\|^2\right]$ individually. Denote by $z := e_i^t + \gamma \nabla f_i(x^t, \xi_i^t)$, then

$$\begin{aligned}
\mathbb{E}\left[\|e_i^{t+1}\|^2\right] &\leq \mathbb{E}\left[\|\mathcal{C}(z) - z\|^2\right] \\
&\leq (1 - \alpha)\mathbb{E}\left[\|e_i^t + \gamma \nabla f_i(x^t, \xi_i^t)\|^2\right] \\
&\leq (1 - \alpha)\left(1 + \frac{\alpha}{2}\right) \mathbb{E}\left[\|e_i^t\|^2\right] + \left(1 + \frac{2}{\alpha}\right) \mathbb{E}\left[\|\gamma \nabla f_i(x^t, \xi_i^t)\|^2\right] \\
&\leq \left(1 - \frac{\alpha}{2}\right) \mathbb{E}\left[\|e_i^t\|^2\right] + \frac{3\gamma^2}{\alpha} \mathbb{E}\left[\|\nabla f_i(x^t, \xi_i^t)\|^2\right],
\end{aligned} \tag{67}$$

where we used Definition 1 and Young's inequality.

**BG asssumption.**   If we assume bounded (stochastic) gradients (BG), i.e., $\mathbb{E}\left[\|\nabla f_i(x, \xi_i)\|^2\right] \leq G^2$ for all $i = 1, \ldots, n$, then using (67) we can derive

$$\frac{1}{T} \sum_{t=0}^{T-1} \mathbb{E}\left[\|e^t\|^2\right] \leq \frac{6\gamma^2 G^2}{\alpha^2}.$$

Combining this bound with (66), we have

$$\frac{1}{T}\sum_{t=0}^{T-1}\mathbb{E}\left[\left\|\nabla f(x^t)\right\|^2\right] \le \frac{4\delta_0}{\gamma T} + \frac{2\gamma L\sigma^2}{n} + \frac{12L^2\gamma^2 G^2}{\alpha^2}.$$

The step-size choice $\gamma = \min\left\{\frac{1}{L}, \left(\frac{\delta_0\alpha^2}{TL^2\sigma^2}\right)^{1/3}, \left(\frac{n\delta_0}{TL\sigma^2}\right)^{1/2}\right\}$, allows us to bound the RHS by $\frac{12\delta_0}{\gamma T}$, and guarantees

$$\mathbb{E}\left[\left\|\nabla f(\hat{x}^T)\right\|^2\right] = \mathcal{O}\left(\frac{L\delta_0}{T} + \left(\frac{L\delta_0 G}{\alpha T}\right)^{2/3} + \left(\frac{L\delta_0}{nT}\right)^{1/2}\right),$$

or, equivalently, $T = \mathcal{O}\left(\frac{L\delta_0}{\varepsilon^2} + \frac{L\delta_0 G}{\alpha\varepsilon^3} + \frac{L\delta_0}{n\varepsilon^4}\right)$ sample complexity to find a stationary point. This analysis using BG assumption and derived sample complexity is essentially a simplified version of the one by Koloskova et al. [2020].[23]

**BGS asssumption.** If we assume bounded gradient similarity (BGS), i.e., $\frac{1}{n}\sum_{i=1}^{n}\mathbb{E}\left[\left\|\nabla f_i(x) - \nabla f(x)\right\|^2\right] \le G^2$, we can slightly modify the derivation in (67) as follows

$$
\begin{aligned}
\mathbb{E}\left[\left\|e_i^{t+1}\right\|^2\right] &\le (1-\alpha)\mathbb{E}\left[\left\|e_i^t + \gamma\nabla f_i(x^t,\xi_i^t)\right\|^2\right] \\
&= (1-\alpha)\mathbb{E}\left[\left\|e_i^t + \gamma\nabla f_i(x^t)\right\|^2\right] + (1-\alpha)\gamma^2\mathbb{E}\left[\left\|\nabla f_i(x^t,\xi_i^t) - \nabla f_i(x^t)\right\|^2\right] \\
&\le (1-\alpha)\left(1+\frac{\alpha}{2}\right)\mathbb{E}\left[\left\|e_i^t\right\|^2\right] + \left(1+\frac{2}{\alpha}\right)\mathbb{E}\left[\left\|\gamma\nabla f_i(x^t)\right\|^2\right] + \gamma^2\sigma^2 \\
&\le \left(1-\frac{\alpha}{2}\right)\mathbb{E}\left[\left\|e_i^t\right\|^2\right] + \frac{3\gamma^2}{\alpha}\mathbb{E}\left[\left\|\nabla f_i(x^t)\right\|^2\right] + \gamma^2\sigma^2. \quad (68)
\end{aligned}
$$

Averaging the above inequalities over $i = 1,\dots,n$ and using BGS assumption, i.e., $\frac{1}{n}\sum_{i=1}^{n}\mathbb{E}\left[\left\|\nabla f_i(x)\right\|^2\right] \le \left\|\nabla f(x)\right\|^2 + G^2$, we can derive via averaging over $t = 0,\dots,T-1$

$$\frac{1}{T}\sum_{t=0}^{T-1}\mathbb{E}\left[\left\|e^t\right\|^2\right] \le \frac{6\gamma^2}{\alpha^2}\frac{1}{T}\sum_{t=0}^{T-1}\mathbb{E}\left[\left\|\nabla f(x^t)\right\|^2\right] + \frac{6\gamma^2 G^2}{\alpha^2} + \frac{2\gamma^2\sigma^2}{\alpha},$$

Combining the above inequality with (66), we have

$$\left(1-\frac{12L^2\gamma^2}{\alpha^2}\right)\frac{1}{T}\sum_{t=0}^{T-1}\mathbb{E}\left[\left\|\nabla f(x^t)\right\|^2\right] \le \frac{4\delta_0}{\gamma T} + \frac{2\gamma L\sigma^2}{n} + \frac{12\gamma^2 L^2 G^2}{\alpha^2} + \frac{4\gamma^2 L^2\sigma^2}{\alpha}.$$

By setting $\gamma = \min\left\{\frac{\alpha}{4L}, \left(\frac{n\delta_0}{L\sigma^2 T}\right)^{1/2}, \left(\frac{\alpha^2\delta_0}{L^2 G^2 T}\right)^{1/3}, \left(\frac{\alpha\delta_0}{L^2\sigma^2 T}\right)^{1/3}\right\}$, we have $\left(1-\frac{12L^2\gamma^2}{\alpha^2}\right) \ge \frac{1}{4}$, and the RHS is at most $\frac{16\delta_0}{\gamma T}$. Therefore,

$$\mathbb{E}\left[\left\|\nabla f(\hat{x}^T)\right\|^2\right] = \mathcal{O}\left(\frac{L\delta_0}{\alpha T} + \left(\frac{L\delta_0 G}{\alpha T}\right)^{2/3} + \left(\frac{L\delta_0\sigma}{\sqrt{\alpha}T}\right)^{2/3} + \left(\frac{L\delta_0}{nT}\right)^{1/2}\right),$$

or, equivalently, $T = \mathcal{O}\left(\frac{L\delta_0}{\alpha\varepsilon^2} + \frac{L\delta_0 G}{\alpha\varepsilon^3} + \frac{L\delta_0\sigma}{\sqrt{\alpha}\varepsilon^3} + \frac{L\delta_0}{n\varepsilon^4}\right)$ sample complexity. Notice that in the single node case ($n = 1$), we have $G = 0$, and by Young's inequality $\left(\frac{L\delta_0\sigma}{\sqrt{\alpha}T}\right)^{2/3} \le \frac{1}{3}\frac{L\delta_0}{\alpha T} + \frac{2}{3}\left(\frac{L\delta_0\sigma^2}{T}\right)^{1/2}$. Therefore, the above rate recovers the one by Stich and Karimireddy [2021] in the single node setting.

---

[23]Up to a smoothness constant and the fact that Koloskova et al. [2020] works in a more general decentralized setting.

