# OpenReview forum: "Momentum Provably Improves Error Feedback!"
_NeurIPS.cc/2023/Conference — NeurIPS 2023 poster_

### Official Review · Reviewer_naHZ · 2023-07-05

**Soundness:** 3 good
**Presentation:** 3 good
**Contribution:** 3 good
**Rating:** 6
**Confidence:** 4

**Summary:**

This paper introduces a modification to the EF21-SGD algorithm by incorporating momentum, resulting in a new algorithm named EF21-SGDM. The innovative analysis accompanying this new method successfully addresses the challenges associated with EF21-SGD, reducing the sample complexity from $\Omega(\sigma^2/\epsilon^2)$ to $\mathcal{O}(\sigma^2/(L\delta_0)$. Importantly, EF21-SGDM operates without requiring an assumption of bounded gradient.

**Strengths:**

The paper proposes a new algorithm for distributed settings with compressed gradient. The authors also provide an original analysis that delivers improved results, contributing substantially to the existing work.

**Weaknesses:**

Although the paper's sample complexity in each iteration number is independent of $\varepsilon$, it still depends on the variance term, $\sigma$. This dependence should be explicitly stated to ensure a comprehensive understanding of the algorithm. Furthermore, Algorithm 1 uses $B_{init}$, but the batch size remains consistent throughout the iterations. It may be more appropriate to avoid this term, considering the batch size does not vary.

**Questions:**

The theoretical results suggest that $\eta=\mathcal{O}(1/T)$, but the experiment utilizes $\eta=0.1$. Could the authors clarify the reasoning behind this discrepancy between theoretical and experimental parameters?

**Limitations:**

No.

---

> ### Author Rebuttal · Authors · 2023-08-06
>
> > Although the paper's sample complexity in each iteration number is independent of $\varepsilon$ , it still depends on the variance term, $\sigma$. This dependence should be explicitly stated to ensure a comprehensive understanding of the algorithm.
>
> The sample complexity of each iteration of our algorithms (EF21-SGDM and EF21-SGD2M) is $1$, since using one stochastic gradient is sufficient at each iteration at every node, see Theorems 2, 3 and 5. Our total sample complexities (see Corollaries 2 and 3) naturally depend on both $\varepsilon$ and $\sigma$. We report this dependence in all theorems, corollaries, tables and carefully compare the dependence on each quantity to the previous work, see lines 281-288 after Corollary 2.
>
> > Furthermore, Algorithm 1 uses $B_{init}$, but the batch size remains consistent throughout the iterations. It may be more appropriate to avoid this term, considering the batch size does not vary.
>
> Our equation (9) in the main Theorem 3 holds for any initial batch-size including $B_{init} = 1$, thus large initial batch size is not necessary for convergence. However, to make a fair comparison in the total sample complexities of EF21-SGD, EF14-SGD, EF21-SGDM and EF21-SGD2M, we distinguish between $\delta_0$ and $\Lambda_0$ (it could be that $\delta_0 << \Lambda_0$) and use a large initial batch size to make them of the same order.
>
> > *Question*. The theoretical results suggest that $\eta = 1/T$, but the experiment utilizes $\eta = 0.1$. Could the authors clarify the reasoning behind this discrepancy between theoretical and experimental parameters?
>
> According to Theorem 3, the best theoretical choice of $\eta$ depends not only on the number of iterations $T$, but also on some unknown parameters such as $\sigma$, $L$ and $\delta_0$. Therefore, we resort to tuning the value of $\eta$. We finetuned $\eta$ from the set {$ 0.01, 0.1 $} on the independent dataset (w8a) before all our main experiments. We describe this procedure in Lines 312-313.  We also include an additional discussion regarding this in Appendix J.
>
>
> ---
>
> We believe we addressed all criticism raised. As we have shown, some of it was just based on a misunderstanding. We hope this might lead to a better score; thanks! We are ready to answer any further questions!

---

> > ### Comment · Reviewer_naHZ · 2023-08-18
> >
> > Thank you for the rebuttals. I do not have more questions.

---

### Official Review · Reviewer_5FHY · 2023-07-06

**Soundness:** 3 good
**Presentation:** 3 good
**Contribution:** 3 good
**Rating:** 7
**Confidence:** 4

**Summary:**

This paper proves that momentum helps EF21-SGD. It makes several non-trivial contributions. First, it theoretically shows that EF21-SGD cannot converge when batch-size is small. Second, it proposes a simple remedy to this issue, i.e., incorporating momentum to EF21-SGD. Third, it proves that EF21-SGDM can converge well even if with small batch-size. Fourth, it extends EF21-SGDM can achieve linear speedup in distributed non-convex optimization without any assumptions on bounded gradient similarity. All these contributions are solid and novel.

**Strengths:**

- It theoretically shows that EF21-SGD cannot converge when batch-size is small.

- It proposes a simple remedy to this issue, i.e., incorporating momentum to EF21-SGD.

- It proves that EF21-SGDM can converge well even if with small batch-size.

- It extends EF21-SGDM can achieve linear speedup in distributed non-convex optimization without any assumptions on bounded gradient similarity.

- The paper is well-written and easy to follow.

**Weaknesses:**

The paper is well-written. I have a few minor questions.

1. In table 1, the authors claim that NEOLITHIC uses a large mini-batch, which may not be correct. While NEOLITHIC uses R times larger batch-size than EF21-SGD per iteration, it runs R times fewer iterations than EF21-SGD (see the NEOLITHIC algorithm in Huang et. al., 2022). On average, NEOLITHIC uses a normal O(1) mini-batch as EF21-SGD.

2. Typo: In line 176, "such" should be "such as".

3. In line 182, the authors claim that the proof technique can help establish linear speedup for Scaffold without relying on data similarity assumption. But Scaffold is not relying on this assumption, right?

4. In Figure 2, NEOLITHIC is far slower than EF21-SGD, which somehow contradicts with the results shown in Huang et. al., 2022. Can the authors provide more details on the experimental settings on NEOLITHIC? How many iterations do NEOLITHIC run? Is accumulated gradient used? How does the total communication cost be counted in NEOLITHIC? Does NEOLITHIC converge slower than the other algorithms in terms of iterations (not in communication cost)?

**Questions:**

See above.

---

> ### Author Rebuttal · Authors · 2023-08-06
>
> > *Question 1.* In table 1, the authors claim that NEOLITHIC uses a large mini-batch, which may not be correct. While NEOLITHIC uses R times larger batch-size than EF21-SGD per iteration, it runs R times fewer iterations than EF21-SGD (see the NEOLITHIC algorithm in Huang et. al., 2022). On average, NEOLITHIC uses a normal O(1) mini-batch as EF21-SGD.
>
> According to Theorem 3 in [Huang et al, 2022], NEOLITHIC requires at each iteration a batch-size of order $\frac{1}{\alpha} \log(G/\varepsilon)$ (even in the deterministic case) in our notations. While the dependence on $G$ and $1/\varepsilon$ are logarithmic, the dependence on $\alpha = K / d$ can still result in a very large batch-size, especially when $K$ is small and the dimension $d$ is large. We are not aware if the analysis of NEOLITHIC can achieve $\mathcal O \left(\frac{1}{\alpha \varepsilon^2} \log(\frac{G}{\varepsilon}) + \frac{\sigma^2}{n \varepsilon^4}\right\)$ sample complexity using the batch-size equal to one at each iteration (i.e., $R = 1$).
>
> >  *Question 3.* In line 182, the authors claim that the proof technique can help establish linear speedup for Scaffold without relying on data similarity assumption. But Scaffold is not relying on this assumption, right?
>
> We thank the reviewer for pointing this out and admit that this sentence might be confusing. In fact, we meant to mention only Scaffnew/ProxSkip algorithm by Mishchenko et al, 2022 here. This remark relates to the corresponding limitation of Scaffnew in the stochastic setting, which is mentioned in the end of Section 5.1 of their work. That is although Scaffnew is provably faster than Scaffold in the deterministic case, their sample complexity in the stochastic setting does not have the linear speedup. We will edit this sentence accordingly in the revision.
>
> > *Question 4.* In Figure 2, NEOLITHIC is far slower than EF21-SGD, which somehow contradicts with the results shown in Huang et. al., 2022. Can the authors provide more details on the experimental settings on NEOLITHIC? How many iterations do NEOLITHIC run? Is accumulated gradient used? How does the total communication cost be counted in NEOLITHIC? Does NEOLITHIC converge slower than the other algorithms in terms of iterations (not in communication cost)?
>
> It does not contradict the results from [Huang et. al., 2022] because they take the parameter $R = 4$ in their experiments. While we follow the theory from [Huang et. al., 2022, Theorem 3] and take $R \approx 1 / \alpha$ because the choice of $R = 4$ is not well justified. Although we admit that $R = 4$ can be a good practical choice, we want to avoid heuristics as much as possible.
>
> All methods (including Neolithic) run the same number of iterations. In all methods, we calculate the number of sent bits in each iteration. This procedure is independent of the methods.
>
> We have checked the convergence of Neolithic in terms of iterations. Neolithic converges **not** slower than other methods and has good performance w.r.t. # of iterations. This is the expected result because Neolithic sends $R \approx 1 / \alpha$ compressed vectors in each iteration.

---

> > ### Comment · Reviewer_5FHY · 2023-08-21
> > **Thanks for the rebuttal**
> >
> > I thank the authors for the rebuttal. They have addressed my concerns. I will keep my current rating and vote to accept the paper in the discussion. Please clarify in the camera-ready version that your experimental results are not contradictory with [Huang et. al. 2022] because of the choices of the $R$ value.

---

### Official Review · Reviewer_hi3u · 2023-07-06

**Soundness:** 3 good
**Presentation:** 3 good
**Contribution:** 3 good
**Rating:** 5
**Confidence:** 3

**Summary:**

The authors present a new method called EF21-SGDM by combining EF21 and Polyak's momentum SGD. The theoretical contribution is that , it improves the communication and sample complexities of previous error feedback algorithms under standard smoothness and bounded variance assumptions. They also propose a double momentum variant to further improve the complexity. The experiments are conducted with non-convex logistic regression problems.

**Strengths:**

- only standard smoothness and bounded variance assumptions are needed.
- identify an issue that EF21 with stochastic gradients has weak sample complexity guarantees, and fix it by leveraging our new Lyapunov function construction and new analysis.
- sample complexity is free of $\alpha$ and batch-free, the best from table 1.
- The theoretical analysis is comprehensive and seems solid.

**Weaknesses:**

- Compared with theoretical contribution, the algorithm itself is straightforward, i.e., combining two existing methods EF21 and SGDM.
- The experiments are validated on only logistic regression, where large-scale distributed training is not as crucial as in larger models.
- EF21-SGD2M is not implemented in experiments.

**Questions:**

Does EF21-SGD2M has practical benefits over EF21-SGDM? If not, how do you verify the complexity is improved over EF21-SGDM?

**Limitations:**

The authors did not discuss societal impact. I believe this is a theoretical paper and do not see much negative societal impact.

---

> ### Author Rebuttal · Authors · 2023-08-06
>
> > 1. Compared with theoretical contribution, the algorithm itself is straightforward, i.e., combining two existing methods EF21 and SGDM.
>
> In our opinion, the simplicity of the algorithms should be viewed as a strength rather than weakness of our work.
>
> **Multiple ways to combine EF21 and SGDM.** When our work is put into the context of the literature on compressed gradient methods, one can notice that several attempts were made before to combine EF with momentum. There are a number of ways one can combine EF with momentum, and it is not clear a priori which combination "works", i.e., gives any provable benefit over the non-momentum variant. For instance, [Xie et al., 2020] analyze the combination of EF14-SGD with Nesterov's momentum (M-CSER) and derive the convergence rate, which matches the one for EF14-SGD without any improvement (!). They also make a strong assumption on bounded gradients (BG) as for EF14-SGD.
>
> Another approach to combine EF with momentum was proposed in [Fatkhullin et al., 2021]. In their work, it is suggested to look at the following scheme (EF21-HB):
>
> $$
> \text{Master: } \quad x^{t+1} = x^t - \gamma v^t ,
> $$
>
> $$
> \text{Nodes: } \quad  g_i^{t+1} = g_i^t + \mathcal{C} (\nabla f_i(x^t) - g_i^t ) ,
> $$
>
> $$
> \text{Master: } \quad g^{t+1} = \frac{1}{n} \sum_{i=1}^n g_i^{t+1}, \qquad v^{t+1} = v^{t} + \eta g^{t+1} .
> $$
>
> As our EF21-SGDM, the above method is also a combination of EF21 and Polyak's momentum. However, the algorithm is very different: first EF21 mechanism is applied, the gradient estimators are aggregated by the Master, and finally the momentum is applied at the server level. In contrast, our EF21-SGDM applies momentum at each node followed by EF21 mechanism. Convergence analysis of EF21-HB is shown in the deterministic case and again only matches the rate of non-momentum variant. It is unclear if such variant would even converge when stochastic gradients (without mini batch) are used.
>
> In summary, there are many ways to combine EF and momentum, and it is unclear if any other combination can show the provable benefit. A few previous works tried, but failed. That is why we believe it is *not straightforward* to find the combination that *works*. Our work is the first to demonstrate the provable advantage of our combination over all non-momentum variants of EF in the non-convex case. We manage to do so by proposing a new Lyapunov function analysis, which appears to be novel even when EF and the compressor ($\alpha = 1$) are removed from our method.
>
> **Differences even in case of no compression.** We would like to point out that even in case of no compression ($\alpha = 1$) the choice of the *step-size* and *momentum* parameters (in our Theorems 2, 3 and 8) are completely different from those proposed in the earlier works on SGDM, e.g., by Liu et al (2020). Due to a different order of momentum parameters, our proof technique is also completely different from [Liu et al., 2020]. Namely, we use the Lyapunov function presented in the equation (8), while the analysis of Liu et al. [2020] and the majority of other works on momentum (including EF21-HB in [Fatkhullin et al., 2021]) relies on (11), which has a completely different interpretation. We elaborate more on these differences in Appendix A (momentum).
>
> From the algorithmic side, this analysis with different momentum parameters can be viewed as an encouragement to use smaller momentum parameters (of order $\eta_t = 1/\sqrt{T}$ instead of constant) or even time varying momentum ($\eta_t = \eta/\sqrt{t+1}$) as we describe in Appendix J.
>
> **New double momentum variant.** Additionally, we analyze a double momentum variant of EF21-SGDM (Section 3.4 and Appendix G), which further improves the sample complexity of EF21-SGDM. We are not aware if such an algorithm was proposed or analyzed before in the literature even in case of no compression ($\alpha = 1$).
>
> > 2. The experiments are validated on only logistic regression, where large-scale distributed training is not as crucial as in larger models.
>
> As suggested by the reviewer, we include additional experiments with a larger model (ResNet-18 deep neural network with CIFAR10 image dataset), see attached PDF file above in the general response (Author Rebuttal). In summary, our previous observations based on non-convex logistic regression (with MNIST dataset) translate into this large scale experiment.
>
> > 3. EF21-SGD2M is not implemented in experiments.
>
> In the above PDF file, we also provide experiments with the double momentum variant of our algorithm (EF21-SGD2M). Our simulations show that the performance of EF21-SGD2M is comparable to that of EF21-SGDM and also improves convergence of the previously known algorithms under this setting.
>
> > Question: Does EF21-SGD2M has practical benefits over EF21-SGDM? If not, how do you verify the complexity is improved over EF21-SGDM?
>
> By "EF21-SGD2M further improves the sample complexity of EF21-SGDM", we mean the *theoretical improvement* when comparing the upper bounds in Corollaries 2 and 3. Notice that the iteration/sample complexity reported in Corollary 3 (for EF21-SGD2M) is better than the one in Corollary 2 (for EF21-SGDM) because the additive term with $1/\varepsilon^3$ disappears for the double momentum variant. We provide some intuition behind this in the beginning of Appendix G. In our work, we do not claim any practical benefits of EF21-SGD2M over EF21-SGDM.
>
> ---
>
> We believe we addressed all criticism raised. We hope this might lead to a better score; thanks in advance! We are ready to answer any further questions!

---

### Official Review · Reviewer_Pi6q · 2023-07-07

**Soundness:** 2 fair
**Presentation:** 3 good
**Contribution:** 2 fair
**Rating:** 4
**Confidence:** 3

**Summary:**

The authors propose a new version of EF-SGD which uses momentum. The authors show that under standard assumptions the proposed method has a better convergence rate. The authors claim that in several cases it is hard to perform large batch sampling like when performing RL training. To overcome this problem they propose a momentum based method, which can overcome the small batch issue. The perform additional experiments on Mnist and real sim datasets and compare there effectiveness.

**Strengths:**

1. The proof indeed looks novel the assumptions are reasonable and the improvement in rates for small batch size is very encouraging.

2. The problem is well motivated.

3. The authors perform experiments and compare with different version of EF-SGD.


**Weaknesses:**


1. EF21 SGD although theoretically motivated is not a practical due to only 1 way compression and often highoverhead, authors should comment on the real world implication of EF21-SGD and EF21-SGDM.

2. The authors are motivating their problem using examples from Medical Literature and Federated RL, but the actual experiments are performed on MNIST. In 2023, work like https://www.mosaicml.com/blog/mosaic-resnet trains Imagenet in 27 minutes using just 8 GPU, shouldn’t the paper at least have experiments using Cifar.

3. The only comparison is with other EF21-SGD variants, can you please have additional comparison with other communication efficient methods. And the comparisons should be on wall clock time rather than bits communicated.

4. Authors seems to be not accounting the memory consumption needed because of momentum, it would be great to have a discussion on that.


**Questions:**

Please comment on the concerns raised in weakness sections.

**Limitations:**

The authors in the opinion of this reviewer have not addressed limitations of their methods and their applicability to problems.

---

> ### Author Rebuttal · Authors · 2023-08-06
>
> > 1. EF21 SGD although theoretically motivated is not a practical due to only 1 way compression and often high overhead, authors should comment on the real world implication of EF21-SGD and EF21-SGDM.
>
> In our work, we specifically focus on the uplink communication (from clients to the server) since it is often the key bottleneck in distributed systems with many clients (many clients are trying to the server simultaneously), and it is crucial to tackle this problem before considering the downlink compression. It is common in this line of work to focus on the uplink compression [Stich et al., 2018], [Beznosikov et al., 2020], [Richtarik et al., 2021]. On the other hand, we agree with the reviewer that the downlink communication can be also important. One way to tackle this problem is to use compression with error feedback for both uplink and downlink communications. In this direction, Fatkhullin et al., 2021 propose a modification of the EF21 algorithm, which supports the bidirectional compression (Algorithm 5, EF21-BC), and analyze this method in a deterministic setting. Our momentum variant can be combined with EF21-BC to achieve (batch-free) convergence of the combined algorithm with stochastic gradients. Due to the space limitation, we describe the pseudocode of such combination above in the "global" response (Author Rebuttal).
>
> > 2. The authors are motivating their problem using examples from Medical Literature and Federated RL, but the actual experiments are performed on MNIST. In 2023, work like https://www.mosaicml.com/blog/mosaic-resnet trains Imagenet in 27 minutes using just 8 GPU, shouldn’t the paper at least have experiments using Cifar.
>
> As requested by the reviewer, we additionally test the algorithms on image recognition task CIFAR10 with ResNet-18 deep neural network, see the attached PDF file. In summary, our observations based on non-convex logistic regression (with MNIST dataset) translate into this large scale experiment.
>
> > 3. The only comparison is with other EF21-SGD variants, can you please have additional comparison with other communication efficient methods. And the comparisons should be on wall clock time rather than bits communicated.
>
> We respectfully disagree that the only comparison is with other EF21-SGD variants. Table 1 shows *theoretical comparisons* with Neolithic and EF14-SGD, which are based on the classical EF14 (EF) mechanism (Seide et al. [2014]). These two methods are not related to the EF21 mechanism. In experiments, we also compare our new method with Neolithic and EF14-SGD. The methods from Table 1 provide the current SOTA theoretical guarantees in our setting, which is why we only consider them.
>
> Our paper is not a systems/software work, where comparison using run time would be appropriate and expected. Instead, we address a specific algorithmic and theory issue  present in existing SGD methods with error feedback (the methods require large minibatches both in theory and practice), and thus our contributions are in designing an algorithmic fix and associated theory which proves that the fix indeed works. A theorem is worth a thousand experiments. Our experiments are meant to illustrate that our method solves this issue, and that our theory has predictive power. We believe that our experiments do exactly that. Combined with our theory, we believe this is conclusive evidence. Note that to show what we set out to show we do not need to rely on any particular computer system - indeed, our aim is to underline the system/architecture independent nature of our improvements.
>
> For this reason, we aim to capture the dependence between the number of sent bits and convergence rates in experiments; these are system/architecture/runtime independent quantities. This way of comparing methods is standard in the literature on communication compression [Gorbunov et al, 2021], [Richtarik et al, 2021], [Huang et al, 2022], [Zhao et al, 2022], since such a measure is independent of specific implementation and computing system / architecture. It is also a way of comparison which ages much more gracefully - systems change quickly, but our plots are independent of these changes.
>
> Notice that compared algorithms calculate the same number of stochastic gradients and communicate the same number of bits in each communication round, so the comparison of methods will not change if we plot the convergence on the number of epochs. To support our argument, we re-run the experiment from Figure 3 (c) and measure the wall-clock time to get $\varepsilon$-solution with $\varepsilon = 0.1.$ One can see that the wall-clock times are strongly correlated with the communication complexity results from Figure 3 (c).
> | Alg.      | Wall-Clock Time |
> | ----------- | ----------- |
> | EF14-SGD | 1179.96 sec.  |
> | EF21-SGD   | 682.28 sec.  |
> | EF21-SGDM  | 314.47 sec.   |
>
> > 4. Authors seems to be not accounting the memory consumption needed because of momentum, it would be great to have a discussion on that.
>
> Compared to EF21-SGD, our momentum variant requires to store one more vector at each node $v_i^t$, thus the memory requirement of the proposed algorithm is indeed larger than that of EF21-SGD by a small numerical constant (< 2). It seems to be a relatively small price to pay considering that EF21-SGD may fail to converge (at least without batch size). On the other hand, compared to EF14-SGD, our EF21-SGDM algorithm requires to store the same number of vectors ($e_i^t <--> v_i^t$), which means that our improvement over EF14-SGD in the sample complexity and the strength of the assumptions comes without resorting to additional memory. We will include a brief discussion about this in the next revision; this is a good suggestion.
>
> ---
>
> We believe we addressed all criticism raised, which seems very minor to us. We hope this might lead to a better score. Thanks in advance! We are ready to answer any further questions!

---

> > ### Comment · Reviewer_Pi6q · 2023-08-13
> > **Additional Clarifications**
> >
> > 1. I am curious why did the authors turn off all the optimizations to achieve SOTA accuracies ? These optimizations like LR decay, data augmentation are standard and are widely used.
> >
> > 2. "A theorem is worth a thousand experiments" and "Our paper is not a systems/software work, where comparison using run time would be appropriate and expected" unfortunately I as a reviewer strongly disagree with these statements. However, I understand the authors  have a different opinion. I personally believe to have a meaningful impact both theoretical and experimental validation should be provided on real metrics. For example - Work like Powersgd (Vogels et al. ) which end up having meaningful impact on improving accuracy had extensive experiments. I understand from the statements that this might not be goal of reviewers, but for a top tier conference like Neurips it would be expected to provide real world impact. Especially in the case where previous methods have been actually compared on runtime.
> >
> > Given these issues I can not convincingly recommend for the acceptance of the paper.

---

> > > ### Author Response · Authors · 2023-08-14
> > > **Re: Additional Clarifications**
> > >
> > > > I am curious why did the authors turn off all the optimizations to achieve SOTA accuracies ? These optimizations like LR decay, data augmentation are standard and are widely used.
> > >
> > > We do this for several reasons:
> > > - First, our paper's aim is *not* to compete with methods/papers/systems whose goal is to achieve SOTA generalization accuracies on selected benchmarks - we agree that in such work such optimizations should be used. Our work is of a completely different variety : our paper is not about generalization at all. Our work isolates an open theoretical question (can error feedback provably work with small minibatches?) and proposes an algorithmic fix (use of momentum), and theory which conclusively shows that this trick works as advertised, in the class of smooth nonconvex functions.
> > > - Second, our experiments are designed to test the predictive power of our theory. Heuristics such as data augmentation and LR decay are orthogonal considerations which are entirely irrelevant in our study. They are important as far as actual generalization performance of various optimizers is concerned, but as explained above, this is not the subject of our paper. For this reason, if we included these heuristics in our experiments, it would actually make the experiments and conclusions one can draw from them more confusing.
> > >
> > > > "A theorem is worth a thousand experiments" and "Our paper is not a systems/software work, where comparison using run time would be appropriate and expected". Unfortunately I as a reviewer strongly disagree with these statements. However, I understand the authors have a different opinion.
> > >
> > > Yes, we are of a different opinion. We strongly believe that theory and empirics have equal value in ML research. One feeds into the other and vice versa. We believe that the ML field needs to stand on both its feet (theory and empirics) to advance and to be truly useful. State of the art empirics typically stands on the shoulders of strong theory, and uses additional tricks, heuristics and ideas to push things further. Such tricks are then studied by theoreticians, uncovering their robustness or brittleness, improving and modifying them, or replacing them with more theoretically well grounded tricks. The actual interplay between theory and practice is much more complicated and intricate than this, of course.
> > >
> > > Once you have the belief that theory and empirics have equal intrinsic value (and we actually believe this is what the community ideally/hopefully *should* believe), then it becomes clear that the community should be able to equally appreciate strong theory and empirical works. Fortunately, as the record of papers accepted to NeurIPS in the past clearly shows, this is the case. We believe it is in fact dangerous to use a double standard. For example,
> > > - We believe that a strong practical/empirical work should stand on its own, and be perfectly acceptable to NeurIPS without the requirement that it contains any theory whatsoever. Of course, some theory is welcome, and can make that paper even stronger (say an 8/9/10), but it should not be a requirement for acceptance.
> > > - Likewise, a strong theory work should stand on its own, and be perfectly acceptable to NeurIPS without the requirement that it contains any experiments whatsoever. Of course, some empirics is welcome, and can make that paper even stronger (say an 8/9/10), but it should not be a requirement for acceptance.
> > >
> > > It seems to us you do not subscribe to this philosophy.
> > >
> > > > I personally believe to have a meaningful impact both theoretical and experimental validation should be provided on real metrics.
> > >
> > > This can't possibly be the way to evaluate NeurIPS papers since otherwise no pure empirical and pure theory work would ever get published - and there are many examples of immensely influential works in these categories.
> > >
> > > > For example - Work like Powersgd (Vogels et al. ) which end up having meaningful impact on improving accuracy had extensive experiments.
> > >
> > > This paper is not a theory paper - it does not include a single theorem in the main body of the paper. The one theorem in the appendix is minor, and not central to the paper. This is a very good example of an empirical work which we believe should be accepted to a top venue. In the same manner, pure theory papers with perhaps just one experiment in the appendix should also be perfectly acceptable to a top conference. As an example, consider the 2012 NeurIPS paper by Nicolas Roux, Mark Schmidt and Francis Bach: A Stochastic Gradient Method with an Exponential Convergence Rate for Finite Training Sets. This work is of a theoretical nature just like ours, isolating an important theory problem and proposing a solution. The experiments are designed to test the theory. Yet, this work won the Lagrange Prize in Continuous Optimization, and had an enormous impact.

---

> > > ### Author Response · Authors · 2023-08-14
> > > **Re: Additional Clarifications (part 2)**
> > >
> > > > I understand from the statements that this might not be goal of reviewers, but for a top tier conference like Neurips it would be expected to provide real world impact. Especially in the case where previous methods have been actually compared on runtime.
> > >
> > > We disagree.
> > >
> > > We believe that each work needs to be judged based on its own merits, and by the standards of the subfield/field it belongs to. Theory works need to judged based on the theoretical breakthroughs and contributions, works that build systems should be judged on the real world efficiency of the systems, network architecture works on the benefits the architecture brings, and so on. If we use a single parameter to judge all works (e.g., SOTA generalization performance), we would be doing a massive disservice to the community, and would effectively narrow down the scope of the field to the detriment of of everybody.
> > >
> > > NeurIPS papers are like the olympics. Different fields have different quality standards. We can't judge all sports by the standard of one. We can't evaluate a marathon runner by the standards of 100m sprint. If we did so, we would disqualify marathon as a discipline, and would not be able to appreciate even a world-record-breaking marathon run.

---

### Author Rebuttal · Authors · 2023-08-06


We thank the reviewers for their feedback and the overall positive evaluation of our work. We are glad that the reviewers appreciate that our studied problem is “**well motivated**”, the paper is “**well-written and easy to follow**”, the analysis is “**novel, comprehensive, solid**”, the assumptions are “**reasonable and standard**”, the improvement in rates is “**very encouraging and contributes substantially to the existing work**”. Reviewers hi3u and 5FHY also appreciate our **lower bound construction** for EF21-SGD and the **linear speedup** property of the proposed algorithms.

At the same time, we took all the criticism seriously, and will soon upload a detailed response to each comment. As requested by several reviewers, we conducted additional experiments with deep neural networks and also tested the performance of the EF21-SGD2M method. We attach these results as a separate one page PDF file below.

---
Due to the space limitation in the response to Reviewer Pi6q, we include here the description of the combination of our EF21-SGDM with EF21-BC algorithm (from [Fatkhullin et al., 2021]) to achieve sparse communication in both directions: from the server (master) to clients (nodes) and from clients (nodes) to the server (master). We denote by $\mathcal C_W$ and $\mathcal C_M$ contractive compressors at the clients and the master respectively.

*Nodes:*
$$
 x^{t+1} = x^t - \gamma g^t ,
$$
$$
v_i^{t+1} = (1-\eta) v_i^{t} + \eta \nabla f_i(x^{t+1}, \xi_i^{t+1}) ,
$$
$$
c_i^{t+1} = \mathcal C_W ( v_i^{t+1} - \widetilde g_i^{t} ) , \quad \text{send } c_i^{t+1} \text{ to master, }
$$
$$
\widetilde g_i^{t+1} = \widetilde g_i^{t} + c_i^{t+1} ,
$$

*Master:*
$$
\widetilde g^{t+1} = \widetilde g^{t} + \frac{1}{n} \sum_{i=1}^{n} c_i^{t+1} ,
$$
$$
b^{t+1} = \mathcal C_M (\widetilde g^{t+1} - g^t ) ,  \quad \text{send } b^{t+1} \text{ to nodes, }
$$

*Master and Nodes:*
$$
g^{t+1} = g^t + b^{t+1} .
$$

One can extend the convergence analysis of our EF21-SGDM variant to the above bidirectional scheme. For this, one should combine Lemma 7 and 8 from [Fatkhullin et al., 2021] with Lemma 2 and 3 in our work.

---

### Decision · Program_Chairs · 2023-09-21

**Decision:**

Accept (poster)

**Comment:**

The paper provides good theoretical contribution and the techniques are of sufficient interest. I recommend acceptance of the paper, though I also believe a more thorough experiments will further increase the strength of the paper.